# An evaluation of a physics-based and a semi-empirical firn model across the Greenland Ice Sheet (1980–2020)

Megan Thompson-Munson[1,2], Nander Wever[1,3], C. Max Stevens[4], Jan T. M. Lenaerts[1], and Brooke Medley[4,5]

[1]Department of Atmospheric and Oceanic Sciences, University of Colorado Boulder, Boulder, CO, USA
[2]Cooperative Institute for Research in Environmental Sciences (CIRES), University of Colorado Boulder, Boulder, CO, USA
[3]WSL Institute for Snow and Avalanche Research SLF, Davos, Switzerland
[4]Earth System Science Interdisciplinary Center, University of Maryland College Park, College Park, MD, USA
[5]NASA Goddard Space Flight Center, Greenbelt, MD, USA

**Correspondence:** Megan Thompson-Munson (megan.thompson-munson@colorado.edu)

**Abstract.** The Greenland Ice Sheet's (GrIS) firn layer buffers the ice sheet's contribution to sea level rise by storing meltwater in its pore space. However, available pore space and meltwater retention capability is lost due to ablation of the firn layer and refreezing of meltwater as near-surface ice slabs in the firn. Understanding how firn properties respond to climate is important for constraining the GrIS's future contribution to sea level rise in a warming climate. Observations of firn density provide detailed information about firn properties, but they are spatially and temporally limited. Here we use two firn models, the physics-based SNOWPACK model and the Community Firn Model configured with a semi-empirical densification equation (CFM-GSFC), to quantify firn properties across the GrIS from 1980 through 2020. We use an identical forcing (MERRA-2 atmospheric reanalysis) for SNOWPACK and the CFM-GSFC in order to isolate firn model differences. To evaluate the models, we compare simulated firn properties, including firn air content (FAC), to measurements from the SUMup dataset of snow and firn density. Both models perform well (mean absolute percentage errors of 14 % in SNOWPACK and 16 % in the CFM-GSFC), though their performance is hindered by the spatial resolution of the atmospheric forcing. In the ice-sheet-wide simulations, the 1980–1995 average spatially-integrated FAC (i.e., air volume in the firn) for the upper 100 m is 34,645 km$^3$ from SNOWPACK and 28,581 km$^3$ from the CFM-GSFC. The discrepancy in the magnitude of the modeled FAC stems from differences in densification with depth and variations in the sensitivity of the models to atmospheric forcing. In more recent years (2005–2020), both models simulate substantial depletion of pore space. During this period, the spatially-integrated FAC across the entire GrIS decreases by 3.2 % (-66.6 km$^3$ y$^{-1}$) in SNOWPACK and 1.5 % (-17.4 km$^3$ y$^{-1}$) in the CFM-GSFC. These differing magnitudes demonstrate how model differences propagate throughout the FAC record. Over the full modeled record (1980–2020), SNOWPACK simulates a loss of pore space equivalent to 3 mm of sea level rise buffering, while the CFM-GSFC simulates a loss of 1 mm. The greatest depletion in FAC is along the margins, and especially along the western margin where observations and models show the formation of near-surface, low-permeability ice slabs that may inhibit meltwater storage.

# 1 Introduction

Most of the Greenland Ice Sheet (GrIS) is covered by a thick, porous layer of partially compacted snow known as firn. The density of firn varies with depth and across the ice sheet and is sensitive to surface mass balance (SMB) processes like accumulation and melt, which cause firn density to also vary on several different temporal scales (e.g., daily, seasonal, annual). Since firn is porous, it is capable of storing meltwater in its pore space in a solid or liquid form, which can buffer the contribution of increased melt rates to sea-level rise (Harper et al., 2012). The mechanisms for meltwater entering into and remaining stored in firn are complex and varied. Meltwater can remain in a liquid form and remain near the surface in weathered ice crusts (Cooper et al., 2018), pool into subsurface lakes (Dunmire et al., 2021), or percolate into the snowpack and remain in a liquid form in firn aquifers (e.g., Forster et al., 2014). Additionally, meltwater can percolate into the snowpack where it refreezes deeper in the firn layer or is stored in the firn's pore space (Pfeffer et al., 1991; Harper et al., 2012). Refreezing of meltwater within the firn frequently occurs and creates ice lenses (<0.1 m thick) and layers (0.1-1 m thick) that can accumulate into low-permeability ice slabs (>1 m thick) (MacFerrin et al., 2022). These ice slabs make deeper pore space potentially inaccessible to meltwater produced at the surface, which reduces the buffering capacity of the firn (Machguth et al., 2016) and increases the elevation below which meltwater runs off (Tedstone and Machguth, 2022). Rapid depletion of the pore space on the GrIS is expected to accelerate mass loss in the 21st century and increase the ice sheet's contribution to sea level rise (van Angelen et al., 2013). Approximately half of GrIS mass loss is due to an increase in meltwater runoff (van den Broeke et al., 2009; Enderlin et al., 2014), making the firn's buffering capacity increasingly important to the GrIS contribution to sea-level rise.

In recent decades, Greenland's firn layer has begun to show evidence of climate change. The 2012 extreme melt season produced expansive ice slabs that persisted for several years and reduced permeability (Culberg et al., 2021). At lower elevations, where significant melt occurs, the meltwater storage capacity of firn has abruptly decreased (Machguth et al., 2016). In higher-elevation areas where less meltwater is generated, the firn structure has still changed through enhanced densification from warmer temperatures and the presence of liquid water (Machguth et al., 2016), which causes ice-sheet surface height lowering (de la Peña et al., 2015). Changes in the amount of air-filled pore space within the firn, known as the firn air content (FAC), have been investigated in both observations (e.g., Vandecrux et al., 2019; Benson, 1996; Braithwaite et al., 1994; Sørensen et al., 2011; Kuipers Munneke et al., 2015) and models (e.g., Medley et al., 2022). Although regeneration of this pore space is slow (Harper et al., 2012), consecutive years of average or below-average melt have shown to temporarily pause FAC depletion (Rennermalm et al., 2021). These complex interactions between melt and pore space depletion, including firn evolving on multi-year timescales, motivate the use of detailed firn models to capture these processes and to enhance our understanding of the changes occurring in the firn.

Modeling firn has become important for estimating mass balance (MB) from satellite altimetry, since this method relies on firn models to interpret the causes of surface height changes (e.g., Li and Zwally, 2011; Arthern and Wingham, 1998; Morris and Wingham, 2015). Changes in surface height cannot be attributed to ice mass or firn density changes without additional information from firn models (Smith et al., 2022). Accurate estimates of ice sheet firn density over time and space are necessary to constrain the uncertainty in MB assessments relying on surface height changes measured from satellite altimetry.

Additionally, understanding the limits and deficiencies in firn models is essential for quantifying uncertainties in altimetry-based MB estimates (Morris and Wingham, 2014; Verjans et al., 2021).

Firn models can also be used to fill in gaps in firn density observations and study how firn properties vary on larger spatial and temporal scales, especially since in-situ and remotely sensed observations can only provide snapshots of firn properties in space and time. Semi-empirical firn models have been used to simulate firn properties on both Greenland (e.g., Medley et al., 2022) and Antarctica (e.g., Ligtenberg et al., 2011). These models use empirical relationships between densification, accumulation, and temperature, and they are often tuned to observations (e.g., Ligtenberg et al., 2011; Medley et al., 2022; Li and Zwally, 2011; Herron and Langway, 1980; Arthern et al., 2010; Simonsen et al., 2013; Verjans et al., 2020). Semi-empirical models are beneficial because they can simulate more accurate depth-density profiles by calibration, which removes the uncertainties introduced by poorly understood densification processes in firn. On the other hand, the observations these models rely on for calibration may not be representative of future firn properties in a warming climate (Ligtenberg et al., 2018). Additionally, these models are often tuned using depth-density profiles, which requires a steady-state assumption and thereby lends uncertainty in their ability to simulate firn changes in a transient climate (Lundin et al., 2017). The alternatives to semi-empirical models are physics-based models that use the material properties of snow and firn to simulate densification based constitutive relations between stress and strain (e.g., Vionnet et al., 2012; Bartelt and Lehning, 2002; Lehning et al., 2002b, a). While physics-based models do not rely on observations for tuning, they generally need more detailed meteorological forcing data and are limited by uncertainties in the representation of physical processes. The wealth of snow physics studies allowed for the development of more complex, physics-based, seasonal snow models like SNOWPACK. Both semi-empirical as well as physics-based firn models have been used in Greenland (e.g., Vandecrux et al., 2020a; Dunmire et al., 2020; Medley et al., 2022; Sørensen et al., 2011; Kuipers Munneke et al., 2015).

The Community Firn Model (CFM) and SNOWPACK firn model have seen significant development in polar regions in recent years. In SNOWPACK, there have been modifications to the settling and microstructure schemes (Groot Zwaaftink et al., 2013; Steger et al., 2017), inclusion of drifting snow impacts on near surface density (Groot Zwaaftink et al., 2013; Keenan et al., 2021; Wever et al., 2022), and optimizations for computational efficiency by improving the layer merging scheme (Steger et al., 2017). The CFM has recently been used over both ice sheets using the GSFC-FDMv1.2.1 densification equation (Medley et al., 2022), which we will subsequently refer to as CFM-GSFC. At an ice-sheet scale, few comparisons of semi-empirical and physics-based models exist (e.g. Steger et al., 2017). In many studies investigating firn models, only a single atmospheric forcing is used with a single model (e.g., Medley et al., 2022). Reported results are often not directly comparable between studies since statistics are calculated over different spatial extents or time periods. This approach makes it difficult to attribute differences between modeled and observed firn properties to model forcing (i.e., boundary conditions) or the model itself (i.e., the representation of physical processes by the model).

Here we use the CFM-GSFC and SNOWPACK to simulate firn properties on the GrIS. Importantly, we use the same atmospheric forcing for both models in order to identify model differences that are independent from the forcing data. We compare the simulated firn properties to point observations and then extend the model domain to the entire ice sheet and simulate evolving firn properties across Greenland from 1980 through 2020. In Section 2, we describe the methods used and include

descriptions of the firn models, atmospheric forcing, and observational dataset. Section 3 reports the results, which we partition into the model evaluation (Section 3.1), a description of firn properties in a steady state climate (Section 3.2), and a description of firn properties in a changing climate (Section 3.3).

## 2 Methods and data

### 2.1 MERRA-2 atmospheric forcing

The Modern-Era Retrospective Analysis for Research and Applications, version 2 (MERRA-2) is a global atmospheric re-analysis spanning the period from 1980 to present day (Gelaro et al., 2017). We use MERRA-2's hourly 2-m air temperature, relative humidity, 10-m wind speed, incoming shortwave radiation, incoming longwave radiation, and precipitation rate for the 1980–2020 period (Global Modeling and Assimilation Office (GMAO), 2015a, b, c, d) to force both firn models. MERRA-2 is a gridded product with a horizontal resolution of 0.5° latitude by 0.625° longitude. Due to convergence of meridians toward the poles, the MERRA-2 grid becomes more tightly spaced at higher latitudes. To account for this, we weight model grid cells by the cosine of the latitude when calculating ice-sheet-wide or basin-wide averages of firn and atmospheric properties. We choose to use MERRA-2 since it is publicly available, regularly updated and released, and spans a temporal window that captures recent climate change (Fig. A2). A different reanalysis product or regional climate model could also be used here, though the exact choice of forcing dataset is less relevant for the firn model intercomparison since we aim to compare the output from the two firn models forced with identical input.

In order to evaluate and compare the firn models, we first run them using forcing data from only the grid cells nearest to firn density observations. We use 177 MERRA-2 grid cells for this first step. For the ice-sheet-wide simulations, we mask out grid cells with an ice coverage of less than 50 % in the MERRA-2 ice cover map, which leaves us with a model domain of 1784 MERRA-2 grid cells. The total area of the model domain is 1.81 million km$^2$, which is greater than the actual ice-sheet area of 1.71 million km$^2$ (Rignot and Mouginot, 2012).

### 2.2 SNOWPACK firn model

SNOWPACK is a single-column, physics-based, multi-layer snow and firn model originally designed for avalanche warning in alpine environments (Bartelt and Lehning, 2002; Lehning et al., 2002b, a). In recent years, this model has been applied over the GrIS (e.g., Van Tricht et al., 2016; Steger et al., 2017; Izeboud et al., 2020; Dunmire et al., 2021), as well as the Antarctic Ice Sheet (Dunmire et al., 2020; Keenan et al., 2021). It uses a Lagrangian framework and adds layers when snowfall occurs with a new-snow density based on atmospheric conditions (e.g., Keenan et al., 2021), including explicit treatment of the wind compaction under drifting and blowing snow conditions (Wever et al., 2022). Firn densification is calculated using a constitutive relationship between stress and strain in snow (Bartelt and Lehning, 2002; Lehning et al., 2002b). For calculating firn temperatures, the upper boundary condition is determined from a surface energy balance scheme. The model describes snow microstructure based on four parameters: grain radius, bond radius, sphericity, and dendricity. These evolve

over time primarily based on temperature, temperature gradients, and liquid water content of the snow layers (Lehning et al., 2002b). SNOWPACK uses the MeteoIO library (Bavay and Egger, 2014) for preparing the meteorological forcing data for the simulations. The library reads the meteorological forcing from the MERRA-2 grids and provides data to SNOWPACK for each grid cell, at each of the SNOWPACK time steps. SNOWPACK is run at smaller times steps than MERRA-2 data is available, and nearest neighbor interpolation (for wind speed) and linear interpolations (for all other variables) are used to provide meteorological forcing at higher frequency than provided by MERRA-2.

We run SNOWPACK with half-hourly time steps. To conserve computational expenses and reduce model output sizes, detailed model output for all firn layers is stored with 7-day temporal resolution. SNOWPACK's simulated firn layers have variable thicknesses, impacted by the layer-merging scheme used to reduce computational cost. As described in more detail in Steger et al. (2017), depending on depth below the surface and similarity of snow properties in adjacent layers, the layers may be merged to reduce computational costs. If those merged layers come closer to the surface, they can be split again to maintain sufficient spatial resolution to capture the steep gradients near the surface. The higher vertical resolution of a few centimeters per layer is required in the near-surface layers where the firn is more sensitive to short-term atmospheric fluctuations. We set the surface roughness to 0.002 m for calculating turbulent energy fluxes when solving the energy balance. Here, we account for atmospheric stability using the Michlmayr et al. (2008) stability correction when a stable boundary layer is diagnosed. For unstable boundaries, which happen rather infrequently (Schlögl et al., 2017), Eq. 8 in Stearns and Weidner (1993) is used. We apply a bucket scheme to represent vertical water percolation in SNOWPACK.

Since the thickness of the firn layer varies across the GrIS and we aim to simulate processes for the full firn column depth, we spin up the model to build a firn layer until a thickness of at least 150 m is reached, or when the bottom 3 m of the simulated firn consist of solid ice with a total thickness of at least 10 m (whichever condition is reached first). To perform the spin-up, we run the model using forcing data from a reference climate interval (RCI) and repeat the model runs until the desired thickness is reached. Once reached, we perform a final model run using the full-length record (1980–2020). We choose an RCI of 1 January 1980 through 31 December 1995 (Fig. A2), which is the same period used in Medley et al. (2022). We make the assumption that this period is representative of the longer-term Greenland climate.

## 2.3 The Community Firn Model (CFM)

The Community Firn Model (CFM, Stevens et al., 2020) is an open-source model framework that simulates physical processes in firn. Its modularity allows users to choose which processes to simulate and which parameterizations to use (e.g., thermal conductivity, densification rate) in a given model run. As the CFM provides a high degree of flexibility, it is important to specific how the CFM has been configured for a particular run. Any pertinent parameterizations used for the model runs in this paper can be found in Medley et al. (2022). As previously noted, in this paper we refer to our particular CFM configuration as "CFM-GSFC" to highlight that we are using the CFM with the semi-empirical GSFC-FDMv1.2.1 firn densification equation. This equation is based on the firn densification equation proposed by Arthern et al. (2010), but optimized using firn-core data from 226 sites across the Antarctic and Greenland Ice Sheets.

Like SNOWPACK, the CFM-GSFC uses a Lagrangian numerical framework; each accumulation event adds a new layer to the grid and one is removed from the bottom. The CFM-GSFC uses a layer-merging scheme at 5- and 10-m depth to reduce computational demands. The CFM-GSFC is coded so that each model time step adds a new layer. As such, daily time stepping generates many thin layers. To reduce computational demands, we use the CFM-GSFC's layer merging scheme. For this study's simulations, we merge 30 of the high-resolution (daily) layers at 5-m depth into mid-resolution (approximately monthly) layers.

At 10-m depth, 12 layers are merged into coarser (approximately annual) layers. Model outputs are interpolated onto a 0.25-m regular grid to reduce output file size.

    For the present work, we run the CFM-GSFC over the GrIS at daily resolution. The CFM-GSFC's required forcing inputs for these model runs are surface temperature, precipitation (rain and snow), sublimation, and melt. However, MERRA-2 does not explicitly provide a melt flux. As such, we force the CFM-GSFC with outputs from SNOWPACK's surface energy balance

scheme, including skin temperature, melt flux, and sublimation. This method ensures that the surface boundary conditions (i.e., mass and energy fluxes) for the CFM-GSFC and SNOWPACK model runs are consistent. We use a constant surface density of $350 \text{ kg m}^{-3}$ for the CFM-GSFC runs, which is a reasonable estimate when compared to observed and SNOWPACK-modeled surface density (Fig. A1). We use a bucket scheme with an enthalpy-based heat flow module to handle meltwater percolation and refreezing. The simulations for each grid cell are initialized with a depth/density profile predicted by the Herron and

Langway (1980) steady state model. We design the CFM-GSFC spin-up to repeat the RCI of 1980–1995 until the entire initial firn column is refreshed.

## 2.4   SUMup observations

    The Surface Mass Balance and Snow on Sea Ice Working Group (SUMup) dataset is a compilation of Arctic and Antarctic observations of SMB components and includes in-situ observations of firn density on both ice sheets (Montgomery et al.,

2018). The 2022 SUMup release contains data from 845 locations in Greenland (Thompson-Munson et al., 2022) that we use to compare with modeled firn properties. The measurements have been taken over the past several decades and the depths of the cores range from 0.03 m to 334.53 m. Of the 845 measurements, 78 are single point measurements of surface density only. For calculating depth-integrated properties, we only use the 767 observations that contain data points from at least two different depths.

## 2.5   Firn air content (FAC) calculation

    To compare the observations with the model results, we select the modeled density profile from the closest location and date to when the measurement was taken. The CFM-GSFC produces daily output, which means the observed and model date are the same. SNOWPACK output for the model evaluation is weekly, which means that the date of the modeled profile can differ from the date of the observed profile by as much as 3 days.

From the observed and modeled density profiles, we calculate the FAC, which has units of meters. The FAC is a quantification of the air-filled pore space in the firn. For any depth interval from $z_j$ (upper bound) to $z_k$ (lower bound) where $z = 0$ m represents

the surface and is increasing downwards, the FAC is calculated as

$$FAC(z_k - z_j) = \int_{z_j}^{z_k} \frac{\rho_{ice} - \rho(z)}{\rho_{ice}} \, dz \tag{1}$$

where $\rho_{ice}$ is the density of ice (917 kg m$^{-3}$) and $\rho(z)$ is the layer density at a given depth.

To evaluate the firn models and ensure a direct comparison with the observations, we first calculate the FAC only over the depths represented by the observations (Section 3.1). For these comparisons, $z_j$ is the uppermost observation depth and $z_k$ is the lowermost observation depth. When examining ice-sheet-wide firn properties (Sections 3.2 and 3.3), we calculate FAC from the surface to a depth of 100 m ($z_j = 0$ m, $z_k = 100$ m), and for 10-m-thick vertical intervals from 0 m down to 50 m ($z_j$ = 0, 10, 20, 30, 40 m; $z_k$ = 10, 20, 30, 40, 50 m).

## 195   2.6   Metrics of model evaluation

We use several metrics to compare the modeled density and FAC to the observations in order to evaluate model performance. The Nash-Sutcliffe efficiency (NSE) coefficient is a measure of a model's goodness of fit and is typically employed in hydrological modeling (NSE; Nash and Sutcliffe, 1970). A value of 1 indicates perfect model performance, whereas a value of 0 indicates that the model's predictive ability is the same as using the observations' means. It is calculated as

$$NSE = 1 - \frac{\sum_{i=1}^{n}(O_i - M_i)^2}{\sum_{i=1}^{n}(O_i - \overline{O})^2} \tag{2}$$

where $O_i$ is the observation value, $M_i$ is the model value, $\overline{O}$ is the observation mean, and $i$ iterates over the $n$ number of values. For quantifying the accuracy of the models, we use the mean absolute percentage error (MAPE), which is calculated as

$$MAPE = \frac{1}{n} \sum_{i=1}^{n} \left| \frac{O_i - M_i}{O_i} \right| \times 100\% \tag{3}$$

Finally, we use the relative bias to understand the fractional degree of under- or over-estimation of the models. At each obser-
vation location, we calculate relative bias as

$$Relative\ bias_i = \frac{M_i - O_i}{O_i} \times 100\% \tag{4}$$

and for the full set of locations, we calculate a single, bulk bias value as

$$Bulk\ relative\ bias = \frac{\sum_{i=1}^{n}(M_i - O_i)}{\sum_{i=1}^{n}(O_i)} \times 100\% \tag{5}$$

## 3   Results

## 210   3.1   Model evaluation

Since we use two models in this study, we first discuss the shared and distinct features in simulated density profiles for two example observations from the SUMup database (Wilhelms, 2000; Machguth et al., 2016). Figure 1a compares a 16.3-m-deep

core with several ice layers collected in southwest Greenland on 12 May 2013 (gray, Machguth et al., 2016) to model results from SNOWPACK (blue) and the CFM-GSFC (green). In this example, both SNOWPACK and the CFM-GSFC simulate high-density layers $\sim$1 m below the surface that formed as a result of high melt in 2012. The observed profile also reaches high densities (>700 kg m$^{-3}$) near the surface. However, neither model captures the even higher-density observed ice layer (>800 kg m$^{-3}$) at $\sim$5 m depth. The models show higher agreement with one another near the surface and begin to diverge with depth. SNOWPACK simulates more variability between layers compared to the CFM-GSFC. This partly results from the fixed surface density of 350 kg m$^{-3}$ set for the CFM-GSFC, while the surface density in SNOWPACK varies based on atmospheric conditions, and partially because the CFM-GSFC outputs are interpolated onto a grid. Figure 1b compares a 102.4-m-deep core from the high-elevation interior measured with gamma-ray attenuation (Wilhelms, 2000) with outputs from both models. The modeled profile shapes generally match the observed profile with an NSE of 0.96 for both SNOWPACK and the CFM-GSFC (Fig. 1b). SNOWPACK's inter-layer variability is present in the upper $\sim$25 m and matches the degree of variability seen in the observations. SNOWPACK's density variability stems from the effect of microstructure on the settling, which disappears at depth due to the model's layer merging algorithm.

Since most observations are from shallow cores (median depth = 2.0 m; Fig. 2a) the observed FAC values are relatively low (median FAC = 1.3 m; Fig. 2b) and do not represent the FAC of the full firn column. In these shallow cores where densification has little impact on FAC, the model performance is a reflection of the models' representations of the surface density. In SNOWPACK, the surface density is modeled from the atmospheric input, while in the CFM-GSFC the surface density is fixed at 350 kg m$^{-3}$. We compare observed and modeled FAC for all 767 points (Fig. 3a, b), but we also partition the dataset into bins based on core depth (Fig. 3c-j) to evaluate model performance in terms of both the surface density parameters (shallower cores) and the densification schemes (deeper cores). We use the following core depth thresholds for binning the data: 0 to 1 m ($n = 253$), 1 to 2 m ($n = 112$), 2 to 10 m ($n = 242$), and >10 m ($n = 160$).

In the comparison of all 767 points, modeled FAC from both SNOWPACK and the CFM-GSFC agree very well with the calculated FAC from the observed density profiles (Fig. 3a, b). The NSE is 0.90 for SNOWPACK and 0.94 for the CFM-GSFC, and the MAPE is 14 % for SNOWPACK and 16 % for CFM-GSFC (Fig. 3a, b). The bulk relative biases of +7.9 % for SNOWPACK and +0.2 % for the CFM-GSFC show that both models are overestimating FAC on average (Fig. 2c, d), though this overestimation is not statistically significant ($p = 0.13$ for SNOWPACK, $p = 0.49$ for the CFM-GSFC). For both models, the relative bias calculated at each observation site is less than 10 % for the majority of points (69 % of SNOWPACK points and 64 % of the CFM-GSFC points). Only 2 % of points in SNOWPACK and 1 % of points in the CFM-GSFC have relative biases greater than 100 %. Many of the large biases occur along the margins of the ice sheet, particularly in the southeast and southwest (Fig. 2c, d).

When the evaluation is performed for the four bins of core depths, we find varying performance in each bin (Fig. 3c-j). SNOWPACK performs best in the shallowest cores (NSE = 0.84, MAPE = 9 %) where the FAC is tightly coupled to the surface density scheme (Fig. 3c). As densification becomes more important with depth, the SNOWPACK model performance decreases, as expressed by both the NSE and MAPE (Fig. 3d–f). For the CFM-GSFC, the FAC in the shallower bins (Fig. 3g, h) is impacted by the fixed surface density and vertical interpolation that together prevent the fine resolution necessary for

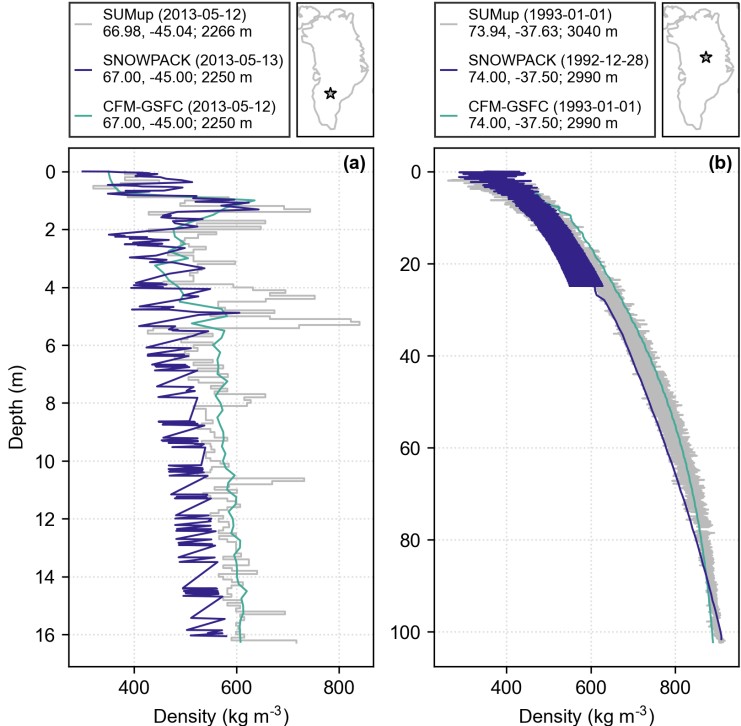

**Figure 1.** Modeled and observed density profiles at two example locations from (a) Machguth et al. (2016) and (b) Wilhelms (2000). The locations of the observations are shown as stars on the map and the profile dates, latitudes, longitudes, and elevations for the observations and models are reported in the legends. Observations are shown in gray, SNOWPACK results are in blue, and the CFM-GSFC results are in green.

comparisons with observations. At depth, the CFM-GSFC generally performs well and has NSE and MAPE values comparable to SNOWPACK (Fig. 3i, j).

For both models, five of the highest absolute biases occur in five cores located in the same MERRA-2 grid cell (Fig. 4). This grid cell is located in an area with observed firn aquifers (Miller et al., 2018), and it contains large topographic gradients. Here, the models use the same MERRA-2 climatology to simulate firn properties for five observations. Both SNOWPACK and the CFM-GSFC overestimate FAC; absolute biases exceed 5 m and relative biases exceed 100 % in all but one of these cases (Fig. 3, 4). Within this grid cell, the observations substantially differ from the models, especially as the distance between

the center of the MERRA-2 grid cell and the observation location increases. The biases are highest furthest from the grid cell center and where the topography is steepest (Fig. 4).

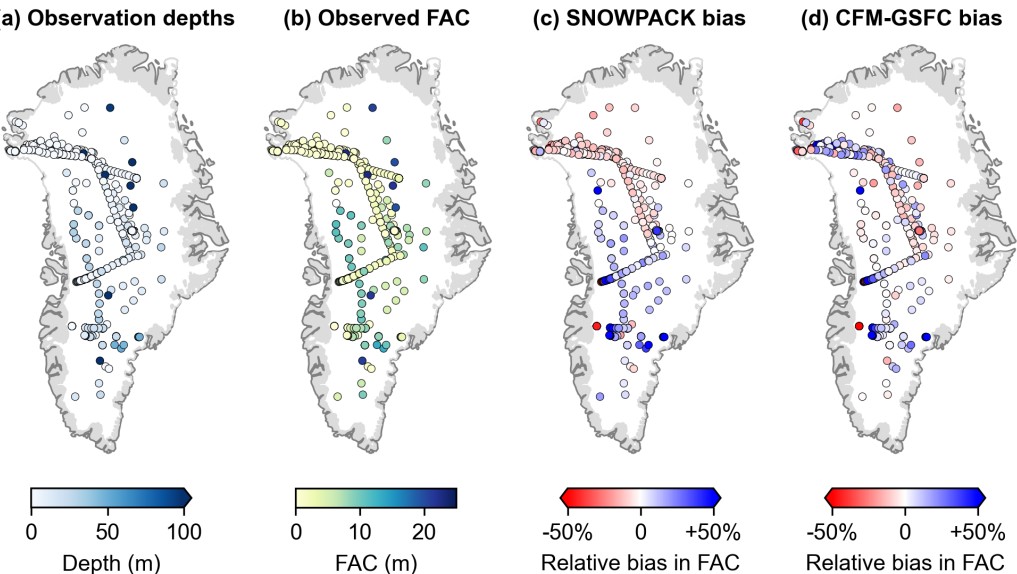

**Figure 2.** (a) Locations of the 767 SUMup cores used for analysis with shading indicating the depth of the core. (b) Firn air content (FAC) calculated from the observed densities in SUMup. (c) Relative bias between SNOWPACK-modeled FAC and observed FAC. (d) Relative bias between the CFM-GSFC-modeled FAC and observed FAC. Positive bias indicates an overestimation of FAC by the model, while negative bias indicates an underestimation of FAC by the model.

## 3.2 Firn properties in a steady-state climate

In this section, we compare the modeled results over the entire GrIS during the RCI. The RCI used for model spin-up spans 1980 through 1995, and since we assume that this period represents a relatively steady-state, long-term Greenland climate

(Fig. A2), the modeled firn is considered to be in steady state as well. This RCI has been used previously by Medley et al. (2022). Since the forcing data from SNOWPACK and the CFM-GSFC are identical, differences in the simulated firn properties can be directly attributed to the differences between models. We calculate FAC for the upper 100 m of the firn column in both SNOWPACK and the CFM-GSFC to reduce any possible biases arising from inconsistent maximum depths between the two models. Unless otherwise stated, all FAC values reported in the remainder of the manuscript have been calculated for the upper

100 m.

We use the liquid-to-solid ratio (LTSR) averaged over the RCI to investigate how both models respond to liquid water input from snowmelt and rain, and solid input from accumulation in a relatively steady state climate. The LTSR is calculated as

$$LTSR = \frac{melt + rain}{snow} \tag{6}$$

and is the same for SNOWPACK and the CFM-GSFC since they use the same forcing data. We examine the extent to which

LTSR predicts FAC, and find a higher $r^2$ value for the CFM-GSFC ($r^2$ = 0.89) compared to SNOWPACK ($r^2$ = 0.77). Overall, the range of SNOWPACK FAC is greater than the range of the CFM-GSFC FAC (Fig. 5a). At very low LTSR values (<0.10)

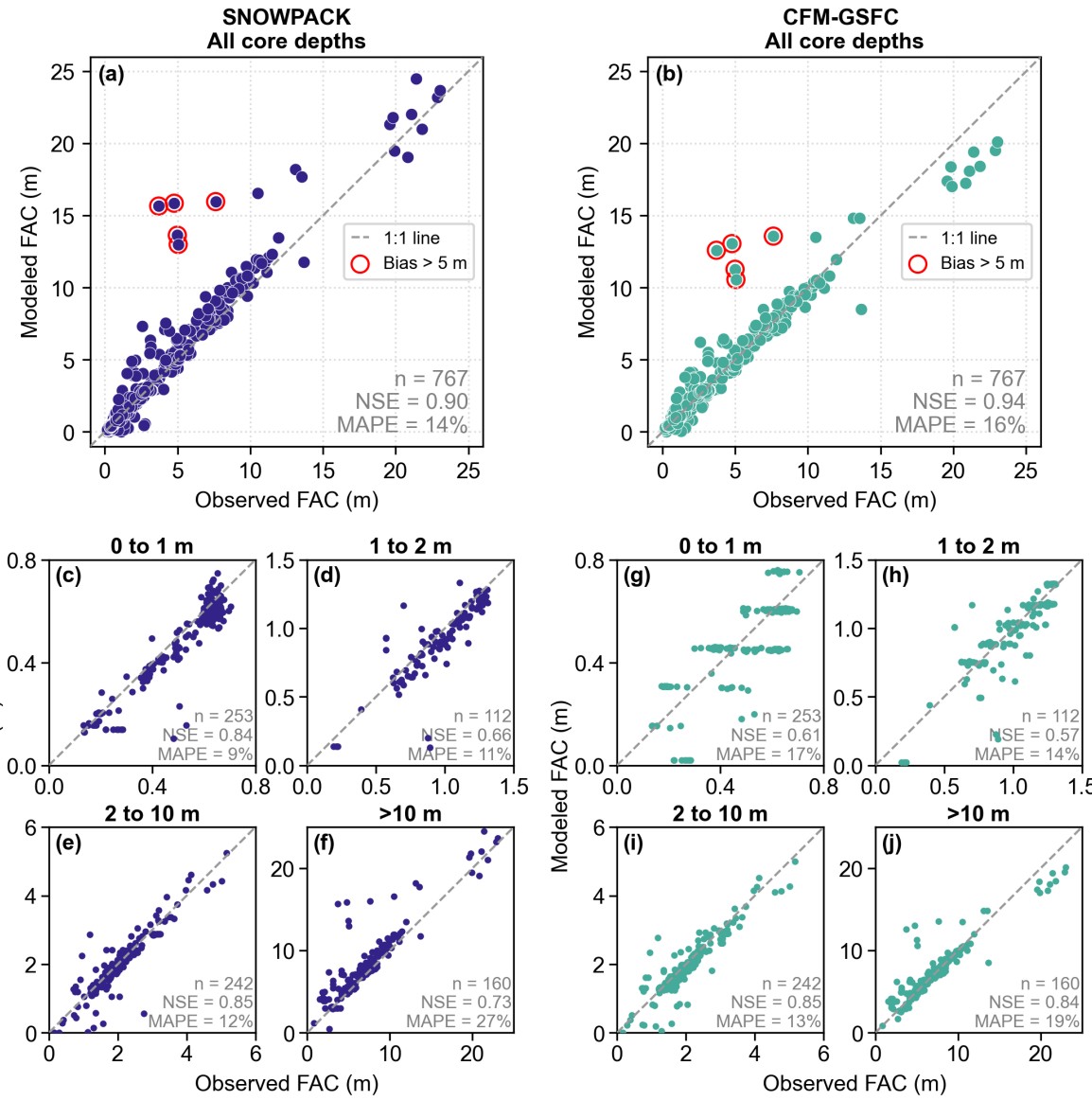

**Figure 3.** Observed versus modeled firn air content (FAC) for all core depths for (a) SNOWPACK and (b) the CFM-GSFC. The smaller panels Panels c-j show the same comparison but for the four bins of observations partitioned by core depth for (c-f) SNOWPACK and (g-j) the CFM-GSFC. The core depth bins are shown in bold above each panel. The number of points ($n$), the Nash-Sutcliffe Efficiency (NSE), and the mean absolute percentage error (MAPE) are reported for each model in the lower right. The gray dashed line is a 1:1 line. Points with biases greater than 5 m are circled in red and correspond to the density profiles shown in Figure 4.

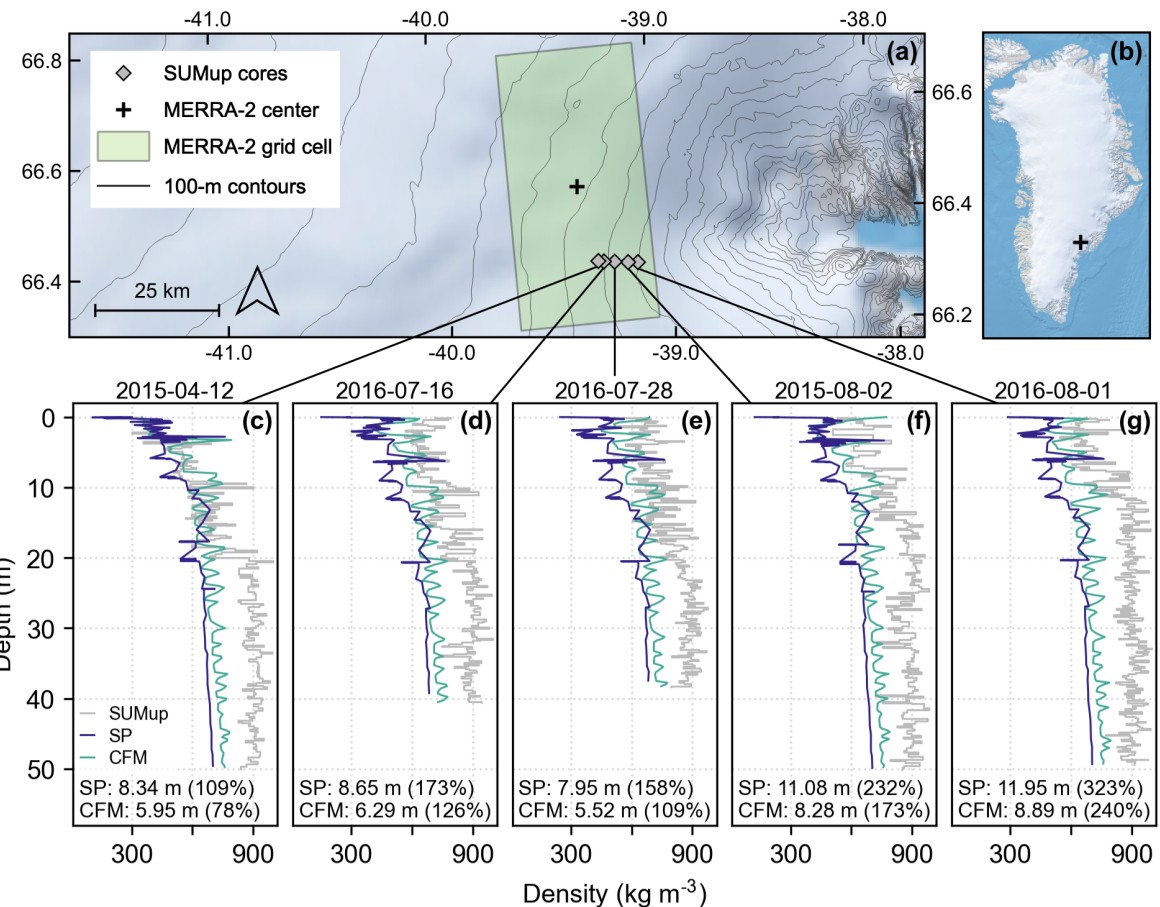

**Figure 4.** Density profiles and locations of high model biases from Fig. 3. (a) Map showing the five SUMup cores plotted in (c-g), the MERRA-2 domain used for SNOWPACK and the CFM-GSFC simulations shown in (c-g), and 100-m surface elevation contours from the BedMachine dataset (Morlighem et al., 2017). (b) Location map of Greenland with the black plus sign indicating the region shown in (a). Maps in panels (a) and (b) were created with QGreenland v2.0.0 (Moon et al., 2022). (c-g) Observed and modeled density profiles for five locations in southeast Greenland where both SNOWPACK and the CFM-GSFC overestimate firn air content by at least 5 m. The modeled profiles come from the same simulation for the MERRA-2 grid cell closest to the firn cores, but differ in date and time to match the observation date as closely as possible. In each panel, we report the FAC absolute bias (modeled minus observed) and the relative bias (in parentheses) for SNOWPACK ("SP") and the CFM-GSFC ("CFM").

where snowfall is the dominant component of surface mass fluxes, FAC in SNOWPACK ranges from 9.7 to 32.1 m, whereas FAC in the CFM-GSFC ranges from 9.9 to only 23.6 m. As the LTSR increases and liquid input dominates, the FAC in both models approaches zero. However, SNOWPACK FAC decreases more rapidly and reaches lower FAC values than the CFM-GSFC at high LTSR values. The CFM-GSFC FAC decreases more gradually with increasing LTSR. In SNOWPACK, 127 points have a FAC less than 1 m, and in the CFM-GSFC only 2 points are as low (Fig. 5a). We also examine how FAC responds to the summer (June, July, and August) 2-m air temperature during the RCI (Fig. 5b). For summer temperatures below ∼-4°C, both models consistently simulate FAC values exceeding 10 m. For these low temperatures, the range of FAC is greater in SNOWPACK (9.9 to 32.1 m) compared to the CFM-GSFC (10.7 to 23.5 m). Both models show a rapid decline in FAC as summer temperatures exceed -4°C, and the modeled FAC is similar at these warmer temperatures.

During the RCI, SNOWPACK and the CFM-GSFC produce similar spatial patterns in FAC, with higher FAC (>10 m) in the ice-sheet interior and lower FAC (<10 m) in the margins (Fig. 6). However, on average, FAC is greater in SNOWPACK than in the CFM-GSFC. SNOWPACK simulates a FAC of 19.1±8.0 m (mean±standard deviation) and the CFM-GSFC simulates a FAC of 15.8±6.0 m, which constitutes a 19 % difference (Fig. 6). The root-mean-squared deviation (RMSD), which represents the averaged difference in FAC between the two models, is 4.0 m. The spatially-integrated FAC, which is the total air volume within the firn layer, is 34,645 km$^3$ for SNOWPACK and 28,581 km$^3$ for the CFM-GSFC. These values represent all modeled grid cells, meaning they include some areas outside of the six basins defined by Rignot and Mouginot (2012). A few areas of missing data exist and are due to one or both of the firn models encountering an issue in the simulation (Fig. 6). On very few occasions, the simulations were unsuccessful because of numerical instabilities and are treated as missing data. For example, if a grid cell is located in the ablation zone and does not receive enough accumulation to build up a firn layer, we treat that grid point as missing data.

We also calculate FAC for vertical segments in steps of 10-m for the ice sheet from a depth of 0 m (surface) down to 50 m (Fig. 7). In both models, the FAC is highest in the shallowest segment (closest to the surface) and lowest in the deepest segment, consistent with densification. In each vertical segment, the SNOWPACK FAC is on average greater than the CFM-GSFC FAC, but the percent difference between the models increases with depth. The models produce similar FAC values between 0 and 10 m, with a difference of 7 %. However, from 40 to 50 m depth, there is a 29 % difference between SNOWPACK and the CFM-GSFC (Fig. 7).

To better understand spatial patterns represented by both models, we compare FAC in each basin of the GrIS as defined by Rignot and Mouginot (2012). The six basins and their abbreviations used in subsequent figures and tables are the northwest (NW), central west (CW), southwest (SW), north (NO), northeast (NE), and southeast (SE) (e.g., Fig. 6). These basins have distinct climatological features captured by MERRA-2 (Table A1). The southeast and southwest are the warmest and wettest basins as they have the highest temperatures, melt, and precipitation. The precipitation in the southeast is substantially higher (>37 %) than in any other basin. The coldest and driest basins are the north and northeast, which are characterized by very low temperatures, melt, and precipitation. Lastly, the central west and northwest basins have moderate amounts of precipitation and melt but relatively low temperatures (Table A1).

The FAC means for each basin are shown for both models in Table 1. The central west has the highest basin-mean FAC in both SNOWPACK (24.8±4.6 m) and the CFM-GSFC (19.5±3.7 m). In both models, the southeast and northwest have similarly high basin-mean FAC values. The lowest basin-mean FAC occurs in the southwest in SNOWPACK (17.4±7.6 m) and the CFM-GSFC (13.9±5.8 m). The best model agreement is in the northeast where the modeled FAC differs by 13 % and the RMSD is 2.7 m, and in the north where the difference is 15 % and the RMSD is 3.0 m (Table 1). The 1980–1995 trends in the spatially-integrated FAC are shown in Table 2 for each basin. The trends are very small (maximum = $3.1\pm0.3$ km$^3$y$^{-1}$, which confirms that no substantial change in FAC occurs over the RCI.

We examine temporal patterns in FAC spatially integrated across the full ice sheet (Fig. 8) and over each basin (Fig. 9). During the 1980–1995 RCI, there is no strong trend in the full ice sheet's air content in either model (Fig. 8), which is consistent with the design of the spin-up. Both SNOWPACK and the CFM-GSFC simulate a short-term increase in FAC from 1982 until 1987, followed by a decrease in FAC until 1990 (Fig. 8). Patterns of short-term (∼1–5 years) variability are more prevalent in each basin (Fig. 9). The same increasing then decreasing pattern in Fig. 8 is evident in the basin-averaged FAC in the northeast, southeast, and southwest (Fig. 9). Only in the north basin are the short-term trends absent.

To examine how the models represent the seasonal cycle in spatially-integrated FAC during the RCI, we subtract the annual means from each year to isolate the seasonal signal. We then fit a sine wave to the data (Fig. A3) and quantify a seasonal signal from the amplitude following methods from Ligtenberg et al. (2012). The amplitudes of the seasonal signals in the spatially-integrated FAC for each basin are reported in Table 3. In SNOWPACK, the strongest signal during the RCI is in the southeast, and the weakest is in the northeast. In the CFM-GSFC, the strongest signal is also in the southeast, whereas the weakest is in the north where the seasonality was undetectable by the chosen methods (i.e., some basins contain too much intra-annual variability for the sine fitting function to detect a seasonal cycle). Integrated across the full ice sheet, the seasonal signal is about two times greater in the CFM-GSFC (129 km$^3$) compared to SNOWPACK (61 km$^3$) (Table 3; Figs. 8, A3).

### 3.3 Firn properties in a changing climate

We now turn our attention to a period where the GrIS was undergoing relatively more change. In the 2005–2020 period, the ice-sheet-wide mean FAC is slightly less than that of the 1980–1995 period for both models, but the difference is not statistically significant ($p = 0.08$ for SNOWPACK, $p = 0.30$ for the CFM-GSFC) (Table 1). Between the two periods, the basin-averaged FAC values are not significantly different ($p > 0.05$ in all basins). In both models, the highest basin-averaged FAC is still in the central west, and the lowest is still in the southwest. Additionally, the best model agreement (13 % difference) still occurs in the northeast (Table 1).

After the 1980–1995 RCI, trends in spatially-averaged FAC begin to emerge in the full ice sheet signal (Fig. 8). SNOWPACK models a decreasing trend of -66.6 km$^3$ y$^{-1}$ during the 2005–2020 period, which is significantly greater in magnitude than the trend of 1.7 km$^3$ y$^{-1}$ throughout the RCI ($p < 0.05$; Table 2). Two extreme FAC depletion events are captured in 2012 and 2019 (Fig. 8a). Between 1980 and 2020, SNOWPACK simulates a loss of 1043 km$^3$ of firn pore space, which could store liquid water equivalent to 2.9 mm of sea level rise. The pattern is similar but dampened in the CFM-GSFC record (Fig. 8b). The 2005–2020 trend of -17.4 km$^3$ y$^{-1}$ is less than that of SNOWPACK ($p < 0.05$) but is still statistically different from the

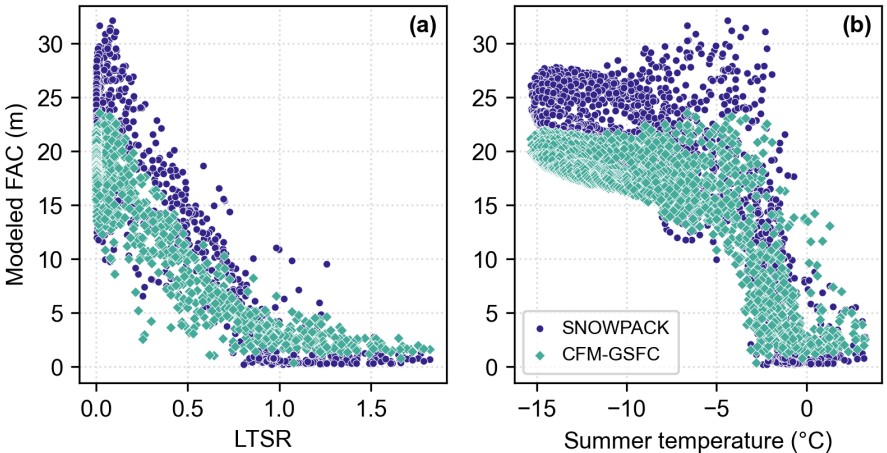

**Figure 5.** Modeled firn air content (FAC) in SNOWPACK (blue) and the CFM-GSFC (green) as a function of (a) the liquid-to-solid ratio (LTSR, Eq. 6), and (b) the summer 2-m air temperature, all calculated for 1980 through 1995.

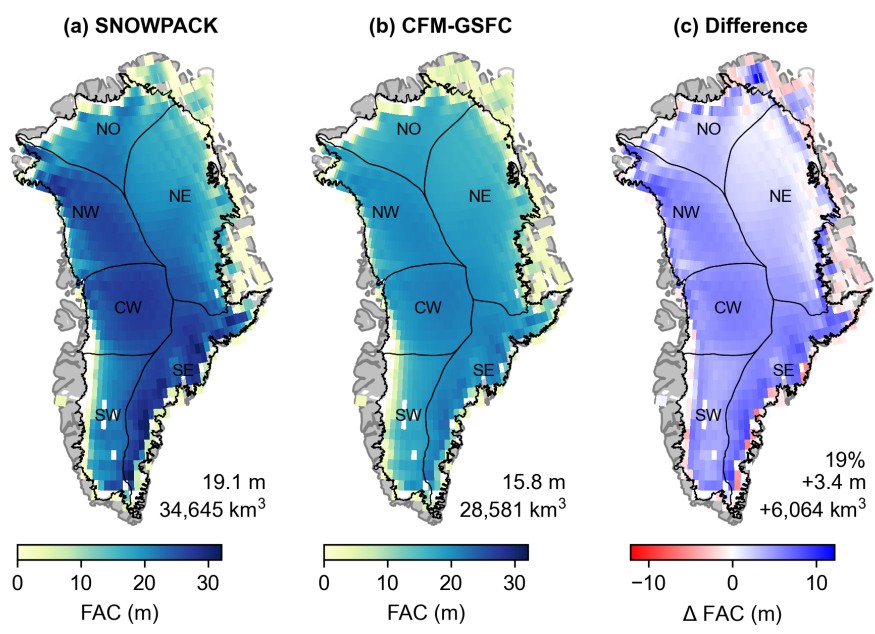

**Figure 6.** Mean firn air content (FAC) calculated over the upper 100 m of the firn column from (a) SNOWPACK and (b) the CFM-GSFC for the reference climate interval (RCI, 1980–1995). Panel (c) shows the difference between the modeled FAC values (SNOWPACK minus CFM-GSFC). The values in the bottom right of each panel are the mean FAC and spatially-integrated FAC. Panel (c) also includes the percent difference. Areas where one or both of the models have missing data are shown in white. Black outlines show the six basins defined by Rignot and Mouginot (2012).

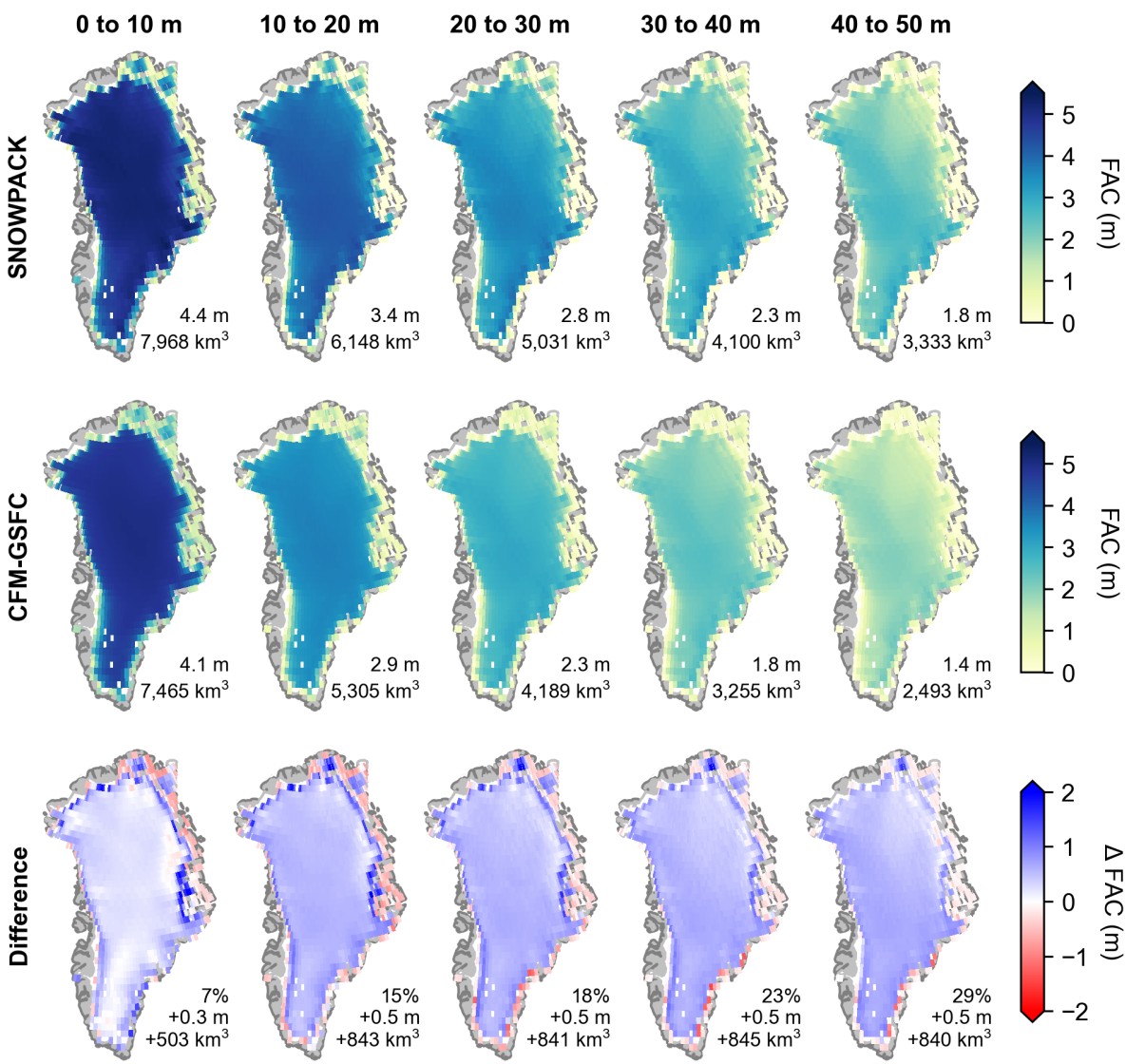

**Figure 7.** Modeled reference climate interval (RCI, 1980–1995) mean firn air content (FAC) calculated for 10-m-thick vertical segments over the GrIS. Top row: SNOWPACK, middle row: the CFM-GSFC, bottom row: SNOWPACK minus the CFM-GSFC. The values in the bottom right of each panel are the mean FAC and spatially-integrated FAC. The bottom row also includes the percent difference averaged over the GrIS.

**Table 1.** Mean modeled firn air content (FAC) for the 1980–1995 reference climate interval (RCI) and for the 2005–2020 period, averaged across each of the six basins shown in Fig. 6. FAC is reported as mean±standard deviation, and the average percent difference and root-mean-square deviation (RMSD) between SNOWPACK and the CFM-GSFC are also shown. The last row shows the statistics for the full GrIS.

| | 1980–1995 | | | | 2005–2020 | | | |
|---|---|---|---|---|---|---|---|---|
| | CFM-GSFC | SNOWPACK | Diff. | RMSD | CFM-GSFC | SNOWPACK | Diff. | RMSD |
| Basin | FAC (m) | FAC (m) | (%) | (m) | FAC (m) | FAC (m) | (%) | (m) |
| NW | 18.5±4.1 | 23.3±5.2 | 23 | 5.1 | 18.5±4.4 | 23.1±5.8 | 22 | 4.9 |
| CW | 19.5±3.7 | 24.8±4.6 | 24 | 5.4 | 19.8±4.0 | 24.8±5.2 | 22 | 5.2 |
| SW | 13.9±5.8 | 17.4±7.6 | 23 | 4.2 | 13.7±6.1 | 16.9±8.0 | 21 | 4.0 |
| NO | 15.5±4.6 | 18.1±5.1 | 15 | 3.0 | 15.2±4.8 | 17.5±5.5 | 14 | 2.7 |
| NE | 16.6±3.5 | 19.0±4.4 | 13 | 2.7 | 16.5±3.8 | 18.7±4.8 | 13 | 2.6 |
| SE | 18.8±4.8 | 23.7±7.7 | 23 | 5.9 | 18.8±5.1 | 23.4±7.9 | 22 | 5.6 |
| GrIS | 15.8±6.0 | 19.1±8.0 | 19 | 4.0 | 15.7±6.3 | 18.8±8.4 | 18 | 3.8 |

**Table 2.** Modeled spatially-integrated firn air content (FAC) trends and standard errors of the trends for each of the six basins (Fig. 6) for the 1980–1995 reference climate interval (RCI) and the 2005–2020 period. The last row shows the trends for the full GrIS.

| | 1980–1995 | | 2005–2020 | |
|---|---|---|---|---|
| | CFM-GSFC | SNOWPACK | CFM-GSFC | SNOWPACK |
| Basin | trend (km$^3$ y$^{-1}$) | trend (km$^3$ y$^{-1}$) | trend (km$^3$ y$^{-1}$) | trend (km$^3$ y$^{-1}$) |
| NW | -0.1±0.1 | -1.1±0.1 | -2.6±0.1 | -12.3±0.2 |
| CW | +0.2±0.0 | +1.5±0.1 | -2.1±0.0 | -10.3±0.1 |
| SW | -1.0±0.1 | -1.2±0.2 | -6.9±0.1 | -16.7±0.2 |
| NO | +0.6±0.0 | +0.4±0.1 | -5.7±0.0 | -11.4±0.1 |
| NE | -1.3±0.1 | -0.8±0.3 | +1.9±0.1 | -2.5±0.2 |
| SE | +3.1±0.1 | +3.1±0.3 | +2.4±0.2 | -4.5±0.4 |
| GrIS | +1.2±1.0 | +1.7±0.9 | -17.4±1.2 | -66.6±1.2 |

**Table 3.** The seasonal signal in each of the six basins (Fig. 6) for the 1980–1995 reference climate interval (RCI) and the 2005–2020 period. The seasonal signal is the amplitude of the best fit sine curve to the spatially-integrated FAC anomalies (Fig. A3). Undetectable signals are reported as "N/A", and the last row shows the values for all basins together.

| Basin | 1980–1995 | | 2005–2020 | |
|---|---|---|---|---|
| | CFM-GSFC signal ($km^3$) | SNOWPACK signal ($km^3$) | CFM-GSFC signal ($km^3$) | SNOWPACK signal ($km^3$) |
| NW | 4 | 9 | 8 | N/A |
| CW | 5 | 6 | 9 | 4 |
| SW | 17 | 5 | 23 | 13 |
| NO | N/A | 5 | 4 | N/A |
| NE | 11 | 4 | 15 | N/A |
| SE | 49 | 25 | 53 | 33 |
| GrIS | 129 | 61 | 165 | 117 |

340    RCI trend of ($p < 0.05$; Table 2). The CFM-GSFC shows less depletion of FAC, with only 356 $km^3$ of pore space lost (1.0 mm sea level rise) by the end of 2020 compared to 1980.

    The marginal areas of the GrIS experience the greatest amount of FAC depletion between 2005 and 2020 (Fig. 10). Both models simulate the same spatial patterns in loss, but the trends vary by basin (Table 2). SNOWPACK simulates a negative trend in spatially-integrated FAC in all basins during this time, with the strongest trend of -16.7$\pm$0.2 $km^3y^{-1}$ in the southwest.

345    The negative trend is weakest in the northeast (-2.5$\pm$0.2 $km^3y^{-1}$) and southeast (-4.5$\pm$0.4 $km^3y^{-1}$), which are also the only two basins where the CFM-GSFC simulates positive trends (1.9$\pm$0.1 and 2.4$\pm$0.2 $km^3y^{-1}$, respectively). The CFM-GSFC also simulates the strongest negative trend in the southwest where the spatially-integrated FAC change is -6.9$\pm$0.1 $km^3y^{-1}$ (Table 2).

    Similarities and differences in short-term trends are modeled across each basin (Fig. 9). In the north and northeast, both

350    models show that FAC generally decreases with time and is lower in the more recent 2005–2020 period compared to the reference climate interval, consistent with FAC depletion due to increased melt and associated water percolation and refreezing processes. The interannual variability is strongest in the southeast where there is no clear increasing or decreasing trend following the RCI. In the southwest, FAC is fairly constant until it rapidly drops in 2012. Following this, FAC continues to show no clear trend. The northwest and central west show similar patterns in FAC with values increasing from 1995 to until 2005.

355    After 2005, SNOWPACK FAC decreases while the CFM-GSFC FAC remains relatively constant (Fig. 9).

    With the exceptions of the northwest and central west, SNOWPACK and the CFM-GSFC generally simulate the same trends in FAC (Fig. 9). However, in all cases, differences arise in seasonal variability and magnitudes of change. In all basins, the two models begin to diverge between 2005 and 2012. Following this divergence, the magnitude of FAC change is greater in SNOWPACK compared to the CFM-GSFC. In most basins, both models capture an extreme drop in FAC in 2012 associated

360    with the extreme melt occurring in that year, followed by an increase in FAC only in the northeast and southeast. The rapid

depletion is greatest in the southwest but less pronounced in the central west and north basins. The magnitude of the 2012 depletion is similar in SNOWPACK and the CFM-GSFC (Fig. 9).

The seasonal signal is stronger in the CFM-GSFC (165 km$^3$) compared to SNOWPACK (117 km$^3$) in the 2005–2020 period (Table 3; Fig. A3). In both models, the southeast basin has the strongest seasonality (52 and 33 km$^3$ in the CFM-GSFC and SNOWPACK, respectively). The weakest signal in the CFM-GSFC is in the north, and in SNOWPACK the signal is too weak to be detected with the chosen methods in the northwest, northeast, and north basins. These three basins also have the lowest annual precipitation (Table A2). Additionally, the amplitude of the seasonal signal is greater in the 2005–2020 period compared to the 1980–1995 RCI for most basins and for the full ice sheet (Table 3; Fig. A3).

Observations have shown that FAC is not completely indicative of available pore space to store meltwater when thick near-surface ice slabs are present in the firn. To evaluate how both firn models reproduce the formation of near-surface ice layers, we identify which grid cells simulate ice slabs in the top 20 m of the ice column in April 2014. We choose this depth and date in order to directly compare to ice slabs detected by IceBridge accumulation radar (MacFerrin et al., 2019). We define an ice slab as a layer with a density of at least 830 kg m$^{-3}$ and a thickness of at least 1 m. However, we do not distinguish between an ice slab that has formed within the firn and any solid ice exposed at the surface of the ice sheet's ablation zone. Our algorithm outputs the depth and thickness of solid ice nearest to the surface, which in some cases could be bare ice at the ablation surface since there is no condition that the ice must be beneath a layer of snow or firn. The distribution of modeled ice slabs and ablation surfaces is largely constrained to the marginal regions of the ice sheet (Fig. 11). Both SNOWPACK and the CFM-GSFC simulate high concentrations of solid ice in the margins and ablation zone. Most of the ice slabs detected by IceBridge accumulation radar in 2014 (MacFerrin et al., 2019) overlap with the simulated solid ice surfaces. SNOWPACK simulates ice slabs and ablation surfaces in 459 grid cells, and the mean depths to those surfaces is 2.5 m. The CFM-GSFC simulates ice slabs and ablation surfaces in 369 grid cells, and the mean depth to those surfaces is 3.8 m.

## 4    Discussion

### 4.1    Model evaluation

In the evaluation of the models with observations, both firn models perform well when evaluated across all SUMup core depths and within each core depth bin. Their overall high NSE coefficients ($\geq 0.90$) and low errors (MAPEs $\leq 16$ %) demonstrate their generally good agreement with observed FAC. Despite the overall good agreement between both models, differences in model performance can also be identified. For example, the model performance is not uniform across all core depths (Fig. 3). In deeper cores of at least 10-m depth, the performance is worse than in the full set of cores, and the MAPE is higher than in any subset for both SNOWPACK (27 %) and the CFM-GSFC (19 %). More deep-firn observations may be needed to better evaluate the models at depth.

The nature of the discrepancies between the modeled and observed properties tend to differ between the two models. The +7.9 % relative bias in SNOWPACK FAC demonstrates that the model tends to overestimate FAC. This bias in SNOWPACK has been shown in both Greenland (Steger et al., 2017) and Antarctica (Keenan et al., 2021). However, Keenan et al. (2021) also

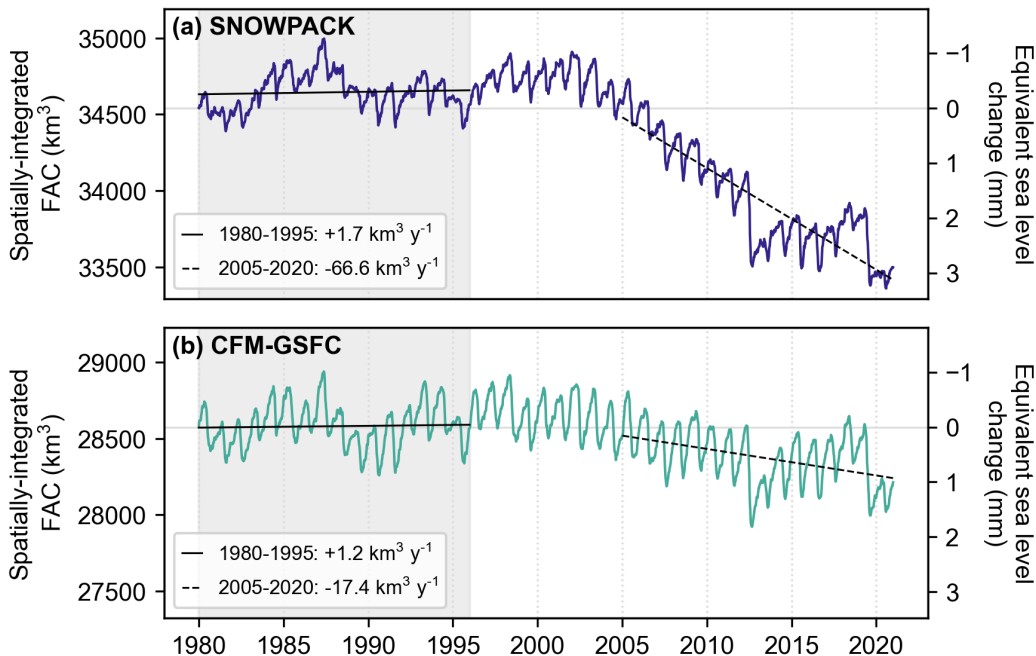

**Figure 8.** Weekly firn air content (FAC) spatially-integrated across the full ice sheet from (a) SNOWPACK and (b) the CFM-GSFC. The left y-axis shows the FAC, and the right y-axis shows the equivalent change in sea level buffering capacity relative to 1980. The gray shading represents the 1980–1995 reference climate interval (RCI). The solid black lines are the trends in spatially-integrated FAC for 1980–1995, and the dashed lines are the trends for 2005–2020.

showed that SNOWPACK outperformed other semi-empirical models in the uppermost 10 m in locations where the models were uncalibrated to the observations, which underscores the utility of SNOWPACK in locations where observations of firn properties are sparse or rapidly changing. In contrast, the CFM-GSFC tends to produce lower FAC values and has a smaller bias of +0.2 %. Below we explore the highest model biases and consider the conditions that cause SNOWPACK and the CFM-GSFC to differ from each other and from the observations.

Some of the highest model biases (>100 %) in SNOWPACK and the CFM-GSFC occur in southeast Greenland and are likely a result of two factors. First, some of the observed density profiles are from cores that were drilled directly into a perennial firn aquifer (Miller et al., 2018). These particular cores approach bulk densities of 1000 kg m$^{-3}$ due to the liquid water contained within the pore space of the firn. In this study, neither model captures the high densities resulting from the firn aquifer because the use of bucket scheme in the models prevents full saturation in the firn. Even though the conditions for firn aquifer development have been previously investigated in models using the bucket scheme by analyzing the presence of layers where water remains liquid throughout winter (Kuipers Munneke et al., 2014), thus far only more advanced water percolation schemes based on Richards equation are able to substantially increase the degree of saturation in firn, congruent with firn aquifer formation (Verjans et al., 2019). A firn hydrology model intercomparison study that included outputs from the two different

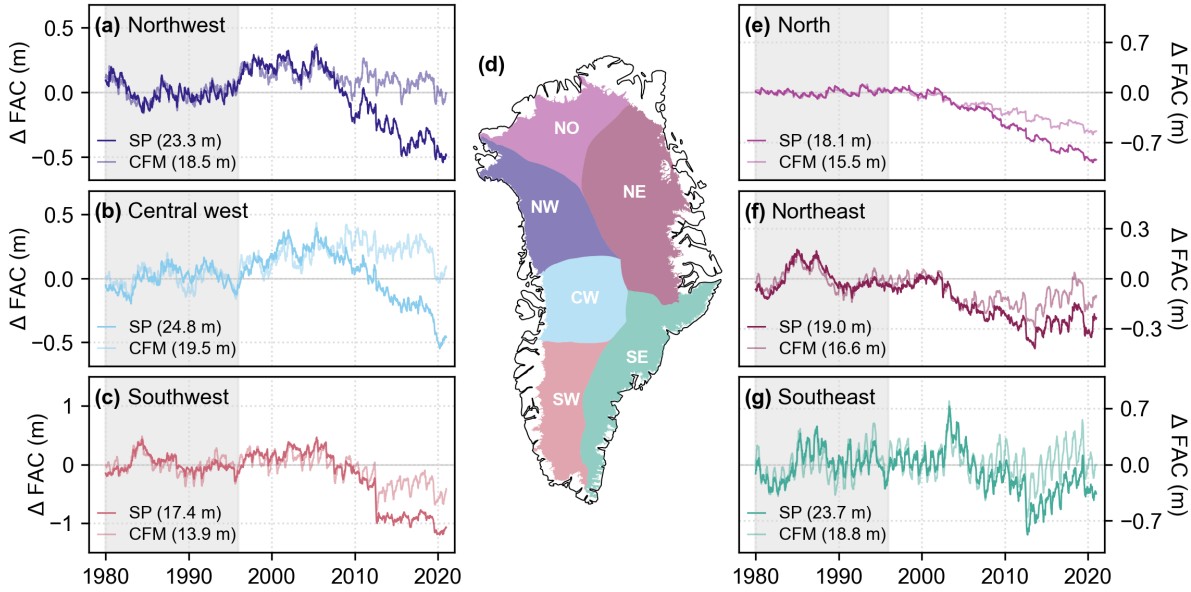

**Figure 9.** (a-c, e-g) Time series of the basin-averaged weekly firn air content (FAC) anomaly from the reference climate interval (RCI, 1980–1995; gray shading) mean, which is shown in the bottom left of each panel. Results from SNOWPACK ("SP") are shown as the darker lines and the results from the CFM-GSFC ("CFM") are shown as the lighter lines. Note the different y-axis scales. (d) Greenland's six basins defined by Rignot and Mouginot (2012) used for calculating trends in firn air content (FAC).

densification schemes in the CFM found that none of the nine models included accurately simulated meltwater infiltration at the four study sites (Vandecrux et al., 2020b). Moreover, Steger et al. (2017) found that the largest model differences between
SNOWPACK and IMAU-FDM (a semi-empirical firn model) occur in the southeast margin of the ice sheet where firn aquifers form. In this study, the high model bias in the five cores in southeast Greenland supports these findings that model differences are highest where liquid water is abundant. This indicates that poor representation of meltwater percolation processes is still a substantial limiting factor in firn model performance (Verjans et al., 2021).

An additional reason why the model bias is high in the southeast is likely the coarseness of the forcing grid in relation to the
steep ice sheet topography in this area. The five locations shown in Figure 4 lie within the same MERRA-2 grid cell and thus share the same atmospheric forcing. However, the cores are located along a transect that spans a steep elevation gradient. For these five cores, the lowest relative bias occurs at the point that is closest to the MERRA-2 grid cell center (i.e., furthest west along the transect) and where the elevation is highest (Fig. 4c). The highest relative bias occurs at the point furthest from the MERRA-2 grid cell, where the elevation is lowest, and where the local vs. MERRA-2 elevation difference is greatest (Fig. 4g).
The steep topography may also lead to strong spatial variations in atmospheric processes such as orographic precipitation, which is not well represented on such a coarse grid (van Kampenhout et al., 2019). This demonstrates the limitations of a

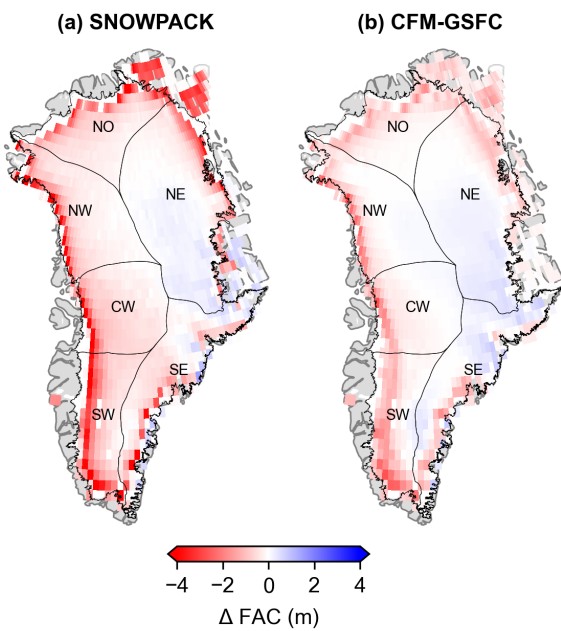

**Figure 10.** Modeled difference in mean firn air content (FAC) between 2005 and 2020 for (a) SNOWPACK and (b) the CFM-GSFC. Black outlines show the six basins defined by Rignot and Mouginot (2012).

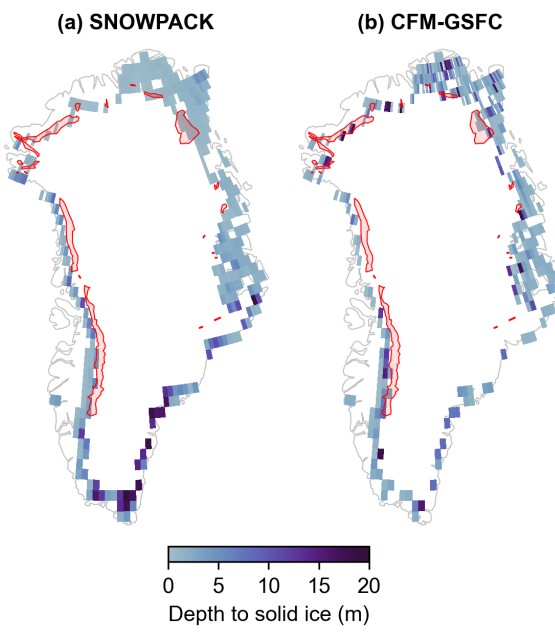

**Figure 11.** Modeled and observed ice slabs in 2014. Red polygons show ice slabs detected by IceBridge accumulation radar (MacFerrin et al., 2019). Modeled ice slabs and ablation surfaces are shaded by the depth to the first ice layer that is at least 1 m thick.

coarsely-gridded forcing, especially in steeply sloped areas where climate is likely to be highly variable within a single grid cell.

Despite the few instances of relatively high disagreement between the models and observations, the overall good performance
of both models in reproducing observations gives confidence in the models' abilities to simulate firn properties across a wide spatial domain. However, most observations are constrained to the accumulation zone, which limits model validation in the hydrologically complex percolation zone. Compared to the dry and flat ice-sheet interior, areas with firn aquifers or steep topography are likely to have higher model uncertainty.

## 4.2 Firn air content response to atmospheric forcing

Since we use identical atmospheric data to force the models, differences in the modeled firn properties are purely due to differences in the firn models themselves. The models simulate complex FAC responses to the forcing variables (Fig. A4), but they are particularly distinct in their response to the LTSR and summer air temperature (Fig. 5). Both models show an inverse, non-linear response between LTSR and FAC (Fig. 5a). However, there is more spread in SNOWPACK FAC compared to the CFM-GSFC FAC, especially for low values of LTSR where snowfall is a larger contributor to surface input. Because
there is less spread in the CFM-GSFC FAC, the LTSR is a stronger predictor of FAC in the CFM-GSFC ($r^2 = 0.89$) compared to SNOWPACK ($r^2 = 0.77$). The large range of possible SNOWPACK-simulated FAC values at low LTSR values is likely due to a combination of (1) the model's sophisticated new-snow density scheme that uses more than only air temperature and accumulation to determine near-surface density, and (2) the densification scheme. The CFM-GSFC uses a fixed new-snow density and its densification scheme is empirically based, which likely explains the differences between the two models. In
the presence of more liquid surface input (i.e., when the LTSR is high), SNOWPACK produces consistently lower FAC values compared to the CFM-GSFC. Our results show that when more liquid water is present compared to snowfall, the CFM-GSFC still simulates available pore space while SNOWPACK's FAC is near-zero in most cases. The response of FAC to the LTSR is important in our consideration of future climate change since we may see the LTSR increase with future warming. In a transition from lower to higher LTSR values, and by proxy, in a transition to a warmer and wetter climate, the CFM-GSFC
shows a more gradual decline in FAC and SNOWPACK shows a more pronounced drop toward near-zero FAC values. These responses to the LTSR can be seen in western Greenland where the CFM-GSFC FAC gradually decreases moving from the interior to the ice sheet margin (Fig. 6b), and SNOWPACK FAC sharply drops off in the same area (Fig. 6a). This is an area vulnerable to modern climate change where increased ice slab formation and decreased firn storage capacity have already been detected (de la Peña et al., 2015).
Additionally, we examine the summer air temperature since it can be directly derived from climate model output unlike the LTSR, which, in the case of MERRA-2, requires more detailed output from a dedicated firn model. We find that a temperature threshold appears to control FAC; at ∼-4°C, FAC in both models rapidly drops (Fig. 5b). The greatest range in FAC and the highest FAC values occur near this temperature for SNOWPACK and the CFM-GSFC. Between ∼-4 and ∼0°C, the models simulate almost the full range of FAC. Therefore, outside of the -4–0°C window, temperature is a relative good predictor of
FAC, especially in the CFM-GSFC where the spread in FAC (0 to ∼24 m) is less than in SNOWPACK (0 to ∼32 m). In both

summer air temperature and the LTSR, the spread in the SNOWPACK values shows that FAC is more than just a function of a single variable or metric, and it points to the complexity of the model. Generally, the FAC simulated by the CFM-GSFC shows less variability for a given LTSR or summer temperature value when compared to SNOWPACK.

### 4.3 Spatial and temporal patterns in firn air content

To place our results in the context of other firn studies, we explore how the modeled FAC compares to existing estimations. We note that this is not a direct comparison since choices of atmospheric forcing, model domain boundaries, and temporal periods will generate differences in FAC that are independent from the firn model choice. Still, we use this comparison to validate the order of magnitude of our results. Our modeled FAC integrated across the full ice sheet for the upper 100 m from the CFM-GSFC (28,581 km$^3$) is similar to a 2010–2017 value (26,800 km$^3$) calculated from observations (Vandecrux et al., 2019).
SNOWPACK's FAC of 34,645 km$^3$ is on the same order but still larger than the observations and the CFM-GSFC-modeled value. SNOWPACK's spatially-integrated FAC in the upper 100 m is close to a regional climate model's (HIRHAM5_MOD) long-term estimate of $\sim$34,000 km$^3$ for this period (Vandecrux et al., 2019).

SNOWPACK and the CFM-GSFC simulate reasonable ice-sheet-wide FAC when compared to other studies, but the two models still differ in magnitude (Fig. 6). In the upper 10 m, the models agree within $\pm$10 %. However, with increasing depth,
the model agreement worsens (Fig. 7). This suggests that the difference in spatially-integrated FAC between the two models arises from the differences in densification with depth. SNOWPACK uses a constitutive relationship between stress and strain in snow to calculate firn densification, while the CFM-GSFC's densification rate is determined using a semi-empirical equation tuned with firn depth-density data. The higher FAC at greater depths predicted by SNOWPACK indicates that its modeled densification rate is slower in the deeper firn than the rate in the CFM-GSFC. This could be related to the fact that SNOWPACK
was developed using data from seasonal alpine snow, which may not be representative of the physical processes driving deep firn densification (Maeno and Ebinuma, 1983; Arnaud et al., 2000). Similarly, Stevens et al. (2020) found that the physically-based snow model Crocus predicted slower densification at Summit, Greenland compared to other firn densification equations. Determining which model performs better at depth is hindered by the paucity of deep firn observations that could provide insight into densification processes.

Further differences in SNOWPACK and the CFM-GSFC are seen in the modeled time series of FAC. Both models show a substantial depletion of FAC from 2005 through 2020, but the CFM-GSFC's response is smaller than that of SNOWPACK. During this time, SNOWPACK's simulated trend of -66.6 km$^3$ y$^{-1}$ corresponds to a -3.2 % change in spatially-integrated FAC. The CFM-GSFC's simulated trend of -17.4 km$^3$ y$^{-1}$ is smaller and the change is also less at -1.5 %. While the magnitudes of trends and FAC changes are greater in SNOWPACK, seasonal signals are greater in the CFM-GSFC. The temporal patterns
in FAC are directly related to the atmospheric forcing. Atmospheric input like accumulation and temperature have strong seasonal patterns, which likely makes them the strongest drivers of FAC seasonality. Both models' FAC seasonal signals are primarily driven by the forcing, but strength of the signal is tied to how each model treats the forcing. SNOWPACK's FAC seasonality is weaker than that of the CFM-GSFC, which points to a more complex treatment of accumulation and temperature in SNOWPACK. In particular, these forcing variables are used in SNOWPACK's new-snow density scheme but not in the

constant new-snow density assumed in the CFM-GSFC. As such, the same new-snow density scheme that leads to the complex relationship between LTSR and FAC may also dampen SNOWPACK's seasonal signal.

Partitioning the record of FAC into climatologically distinct basins reveals further differences in the models and spatial variability. Notably, the signature of the 2012 extreme melt season can be seen as an abrupt drop in FAC in three basins (Fig. 9). In the southeast, northeast, and northwest, both models show that FAC begins to rebound immediately after the 2012

depletion. This replenishing of the firn pore space has also been observed in shallow firn cores following the extreme 2012 melt season (Rennermalm et al., 2021). In the southwest, this rebound is only detected in the CFM-GSFC and not SNOWPACK, which again demonstrates some of the discrepancies between the models. The replenishing of the firn layer is closely tied to high accumulation and low melt. As such, the models' different responses point to the dissimilar treatment of the atmospheric forcing and densification schemes. While they do not agree in every basin, both models are able to capture the ice sheet's

rebuilding of some of the porous firn layer lost during an extreme melt event.

While spatially-integrated FAC describes the total volume of pore space in the GrIS, permeability and access to pore space is important for fully understanding the buffering capacity. Ice slabs, which may render deep pore space inaccessible to meltwater (Machguth et al., 2016), are simulated in the ice sheet's marginal areas where the highest FAC depletion occurs between 2005 and 2020. The strongest negative trend is in the southwest, which has the warmest temperatures and highest melt compared to

other basins during this period (Table A2). These findings agree well with observations that reveal significant FAC depletion in the low-accumulation percolation zone in western Greenland related to increased melt (Vandecrux et al., 2019). Pore space depletion can also be caused by firn densification, which in turn modifies the meltwater refreezing and retention capacities of the firn in a complex manner (Vandecrux et al., 2020b). FAC depletion is found both where ice slabs and ablation surface are simulated and in areas up-glacier from these solid ice surfaces (Figs. 10, 11). This FAC depletion and firn densification may

prime the firn for future migration of ice slabs toward the ice-sheet interior. The differences in the modeled solid ice locations and depths is likely attributed to the overall diverging behavior of the models in wet firn zones, which agrees with findings in the RetMIP firn model intercomparison study (Vandecrux et al., 2020a). SNOWPACK and the CFM-GSFC both rely on the bucket scheme to govern the vertical percolation of meltwater within the pore space, and both have the same applied surface melt from SNOWPACK's surface energy balance scheme. This means that the differences in ice slab and ablation surface locations and

depths are related to the modeled firn structure and temperature, and the models' treatment of atmospheric input. SNOWPACK simulates 24 % more grid cells with ice slabs and ablation surfaces, which is in line with the finding that SNOWPACK's FAC decreases more rapidly (and thus the firn column densifies more rapidly) than the CFM-GSFC's in recent years.

## 4.4 Study limitations

The use of identical forcing data (MERRA-2) allows for direct comparison between the SNOWPACK and CFM-GSFC models

in this work. We also use the same approaches to the spin-up, which uses an RCI of 1 January 1980 through 31 December 1995. Although we assume that the steady-state conditions of the RCI represent the Greenland climate preceding 1980, we acknowledge that they are not representative of true conditions. For example, in the ~100 years before the RCI, significant trends in climate over the GrIS have been found (Hanna et al., 2011). Our steady-state assumption does not allow for such trends

to appear in the spin-up, but the lack of pre-1980 data necessitates such assumptions. Since the focus of this work is to compare firn model results that are independent of the choice of forcing, the RCI assumptions do not impact the intercomparison. When comparing the firn models to the observations, the steady-state assumption may have an impact at depth. Deeper firn in the models is simulated from the repeated 1980–1995 climate, but real firn in the observations is older and may have formed during times when trends in the climate are apparent (e.g., in the pre-1980 20th century; Hanna et al., 2011).

The firn observations themselves are valuable snapshots of firn properties for a specific time and place, but their lack of temporal and spatial continuity limits the extent of this study's evaluation. Density and firn air content from the SUMup observational dataset do not provide sufficient information about how firn evolves through time. Moreover, the timing of when the observations were collected is not uniformly distributed throughout the year, which means there is less information on, for example, winter firn properties versus summer firn properties. However, a key feature of the SNOWPACK and the CFM-GSFC firn models is their ability to simulate the evolution of firn properties on fine temporal scales, even though validation of these model results is very restricted. Similarly, validating the model sensitivities to changes in climate forcing is hampered by the lack of temporal continuity in the available field data.

Additionally, the fact that the field data are spatially limited means that they do not fully capture the variety of potential firn regimes. This is probably most crucial in the percolation zone where few observations exist, yet where meltwater processes are complex and not often well-represented in models (Vandecrux et al., 2020b). For example, the present study relies on the bucket scheme for vertical meltwater transport, which is a simple choice using only the density to control downward water percolation, but it is severely limited in reproducing more complex melt water processes related to firn aquifer and ice slab formation (Verjans et al., 2021). While more sophisticated water percolation schemes exist (particularly those based on Richards equation) and can obtain better results (Verjans et al., 2019), they also show a stronger dependence on firn properties, including for example grain size. The paucity of percolation zone observations limits the opportunities for evaluating model performance, in particular regarding the choice of meltwater percolation scheme.

## 5   Conclusions

An evaluation of the physics-based firn model SNOWPACK, and the CFM-GSFC (the CFM configured with a semi-empirical densification equation) reveals overall high model performance when compared with FAC derived from density observations in the SUMup dataset. Model error is higher in comparisons with deeper cores than in shallower ones, and the highest model biases occur in the marginal areas of the GrIS where conditions controlling FAC (e.g., melt, snowfall, topography) are most variable. Comprehensive model evaluation in this work is hindered by the shortage of deep firn observations, the limited spatial coverage of observations, and the lack of time-varying measurements that could give insight into the evolution of firn properties with time. With the available and chosen evaluation data (i.e., density and FAC), SNOWPACK and the CFM-GSFC differ but still perform well.

The use of identical atmospheric forcing lets us examine how the models respond differently to the same forcing due to structural model differences and parameterization choices. The summer air temperature is a factor for examining the impact

of forcing conditions on the FAC. We find that summer air temperature is related to modeled FAC, which exhibits an abrupt decrease at a summer air temperature threshold of $\sim$-4°C in both models. For average summer air temperatures above $\sim$0°C, both models simulate low FAC values. We also employ the LTSR as a metric summarizing the climatological regime of accumulation and melt. While FAC more gradually decreases in the CFM-GSFC as the LTSR increases, SNOWPACK FAC decreases more rapidly and reaches near-zero FAC values not simulated by the CFM-GSFC. These different responses in air content to increasing liquid water will become important in future warming scenarios. Additionally, the CFM-GSFC's stronger relationship between LTSR and FAC, and the model's stronger seasonal signal point to a more direct impact of forcing variables like accumulation and temperature on the CFM-GSFC compared to SNOWPACK. The models also differ in their response to recent climate change during the 2005–2020 period with SNOWPACK simulating more FAC depletion and more ice slab and ablation area development.

The spatially-integrated FAC during the 1980–1995 RCI is 34,645 km$^3$ from SNOWPACK and 28,581 km$^3$ from the CFM-GSFC, which are both reasonable estimations when compared to other studies (e.g., 26,800 km$^3$ from observations and $\sim$34,000 km$^3$ from a model (Vandecrux et al., 2019)). However inconsistencies in atmospheric forcing data, model domain area, and temporal periods render direct comparisons between studies difficult. Our spin-up is designed such that no significant change occurs in FAC between the start and end of the 16-year period. In a more recent period of the same length (2005–2020), FAC depletes at a rate of 66.6 km$^3$ y$^{-1}$ in SNOWPACK and 17.4 km$^3$ y$^{-1}$ in the CFM-GSFC. The greatest pore space depletion is along the western margins where both models and observations show development of ice slabs and ablation surfaces. This highlights the vulnerability of the firn layer's meltwater storage capacity, especially in these low-elevation and high-melt areas. Notably, the pore space is more depleted in SNOWPACK. Over the full 41-year period, SNOWPACK simulates a loss of pore space equivalent to storing 3 mm of sea level rise, while the CFM-GSFC firn loses only an equivalent of 1 mm.

In the present work, neither model clearly outperforms the other within the scope of the evaluation. Even though we identify disparities between results from both models, we are restricted by the limited availability of the required observational data to draw conclusions about the accuracy of one model over the other. Based on our work, we can draw conclusions about the potential benefits and drawback of each model. The physics-based design of SNOWPACK means that it is not tuned to observations and consequently not biased toward available observational data. This may result in more realistic simulations of firn properties under future climate conditions, whose effects are not capture in existing firn observations. However, firn physics are not fully understood and knowledge gaps limit the accuracy of the model. The CFM's modular design allows for the user to easily choose from several densification schemes. Semi-empirical densification schemes such as the one used in the present work are tuned to observations, which means that realistic densification relationships are built into the model and there is less need to rely on poorly understood physics. Nevertheless, firn models in general are limited by knowledge gaps in firn hydrological processes such as vertical meltwater percolation, lateral flow, and conditions for firn aquifer and ice slab formation. Additional research focusing on obtaining detailed observations of these processes would provide opportunities for important developments in firn modeling.

# Appendix A

**Table A1.** Mean atmospheric forcing variables for the 1980–1995 reference climate interval (RCI), averaged across each of the six basins shown in Fig. 6. Values are reported as mean±basin standard deviation.

| Basin | Temperature ($^\circ$C) | Precip. (mm w.e. yr$^{-1}$) | RH (%) | Wind (m s$^{-1}$) | ISWR (W m$^{-2}$) | ILWR (W m$^{-2}$) | Melt (mm w.e. yr$^{-1}$) |
|---|---|---|---|---|---|---|---|
| NW | -23.5±4.8 | 390.5±204.4 | 92.2±5.5 | 4.6±1.2 | 115.8±6.7 | 169.8±15.6 | 43.1±117.2 |
| CW | -23.4±4.0 | 427.3±132.0 | 92.6±4.8 | 5.7±1.4 | 127.2±4.0 | 171.9±12.1 | 31.5±103.0 |
| SW | -17.0±3.7 | 659.9±368.4 | 87.1±5.3 | 7.9±1.7 | 138.4±7.8 | 191.1±12.8 | 255.2±345.6 |
| NO | -24.3±3.2 | 192.5±63.0 | 91.5±4.8 | 3.9±0.9 | 111.0±5.3 | 164.2±10.2 | 23.7±61.6 |
| NE | -25.4±4.2 | 188.0±89.0 | 90.1±7.7 | 4.0±1.7 | 122.7±5.9 | 158.6±11.6 | 15.9±50.6 |
| SE | -16.8±5.6 | 1049.4±681.3 | 85.4±7.2 | 6.0±2.5 | 132.6±7.4 | 191.4±20.2 | 262.5±427.4 |

**Table A2.** Mean atmospheric forcing variables for the 2005–2020 period, averaged across each of the six basins shown in Fig. 6. Values are reported as mean±basin standard deviation.

| Basin | Temperature ($^\circ$C) | Precip. (mm w.e. yr$^{-1}$) | RH (%) | Wind (m s$^{-1}$) | ISWR (W m$^{-2}$) | ILWR (W m$^{-2}$) | Melt (mm w.e. yr$^{-1}$) |
|---|---|---|---|---|---|---|---|
| NW | -22.1±4.7 | 399.1±212.3 | 91.5±6.2 | 4.7±1.3 | 116.8±6.8 | 173.8±14.9 | 78.2±184.2 |
| CW | -22.0±4.1 | 427.7±137.3 | 92.1±5.1 | 5.8±1.5 | 128.2±4.1 | 176.4±12.5 | 58.0±166.2 |
| SW | -15.5±3.6 | 641.5±363.3 | 86.9±5.4 | 7.7±1.6 | 138.7±7.6 | 196.3±13.0 | 353.3±426.4 |
| NO | -22.8±3.2 | 164.1±46.2 | 90.5±5.9 | 3.7±0.9 | 113.5±4.8 | 167.2±9.5 | 48.0±104.6 |
| NE | -24.0±4.3 | 190.5±95.3 | 89.6±8.3 | 3.9±1.8 | 124.0±5.6 | 162.8±11.6 | 29.8±81.3 |
| SE | -15.7±5.6 | 1079.2±745.9 | 85.1±7.8 | 6.0±2.6 | 133.0±7.2 | 195.9±20.5 | 313.3±477.6 |

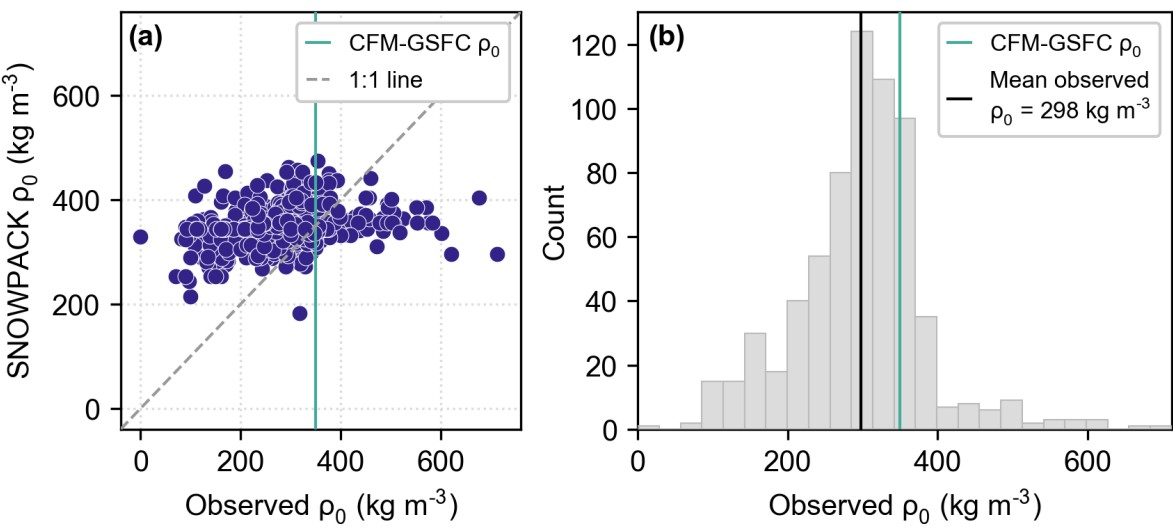

**Figure A1.** (a) Observed surface density ($\rho_0$) from SUMup versus SNOWPACK. Since some observations begin farther below the surface, in this figure, observed $\rho_0$ is defined as the uppermost density measurement that is within 0.1 m from the surface. The SNOWPACK $\rho_0$ is calculated over the same vertical segment as the SUMup observation. The CFM-GSFC uses a prescribed surface density of 350 kg m$^{-3}$ (green vertical line), which falls near many of the observed surface densities. (b) Histogram of observed surface density with the mean represented by the black line. Also plotted is the CFM-GSFC surface density of 350 kg m$^{-3}$.

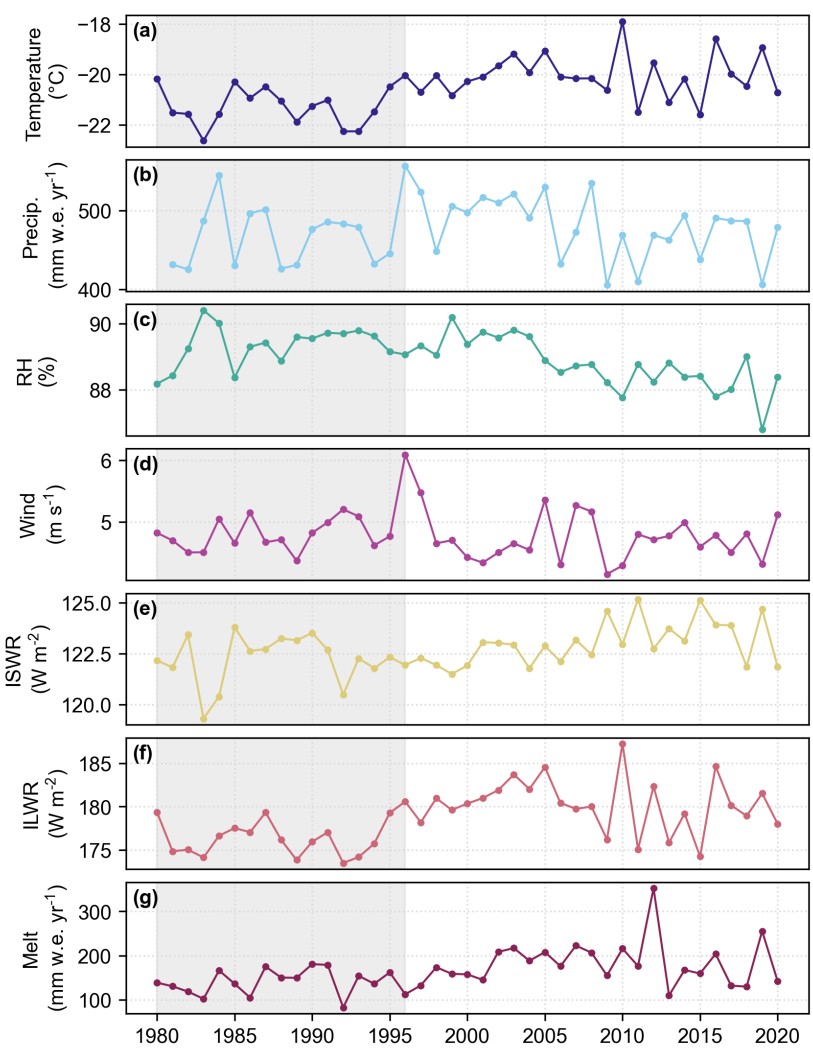

**Figure A2.** Time series of annually-averaged MERRA-2 forcing variables: (a) temperature, (b) precipitation, (c) relative humidity, (d) wind, (e) incoming shortwave radiation (ISWR), and (f) incoming longwave radiation (ILWR) averaged across the GrIS. Also shown is (g) the annually-summed melt output from SNOWPACK used as forcing in the CFM-GSFC. Gray shading represents the reference climate interval from 1980 to 1995.

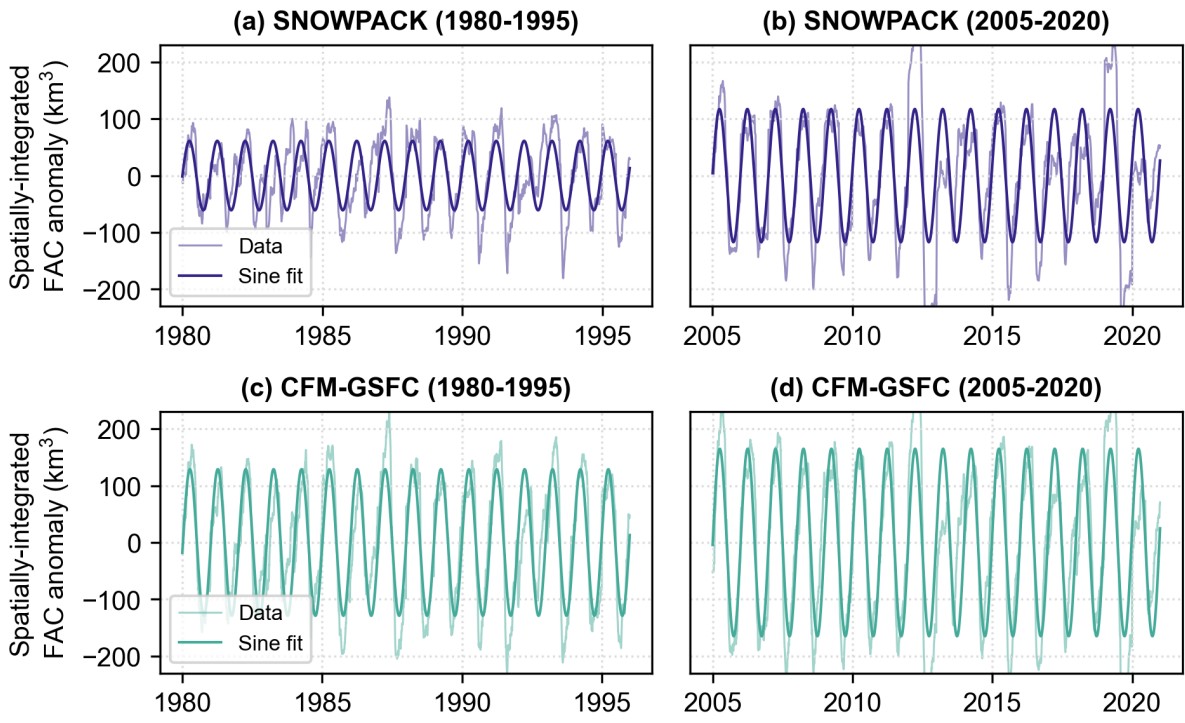

**Figure A3.** The seasonal signal in spatially-integrated weekly firn air content (FAC) for the full ice sheet. (a) SNOWPACK seasonal signal during the 1980–1995 period, (b) SNOWPACK seasonal signal during the 2005–2020 period, (c) the CFM-GSFC seasonal signal during the 1980–1995 period, (d) the CFM-GSFC seasonal signal during the 2005–2020 period. The thinner lines show the anomaly, which is calculated by subtracting each year's mean spatially-integrated FAC from the record. The thicker lines are the best fit sine curve. The amplitudes of the sine curves represent the seasonal signal. Note that the y-axis scales are the same in all four panels.

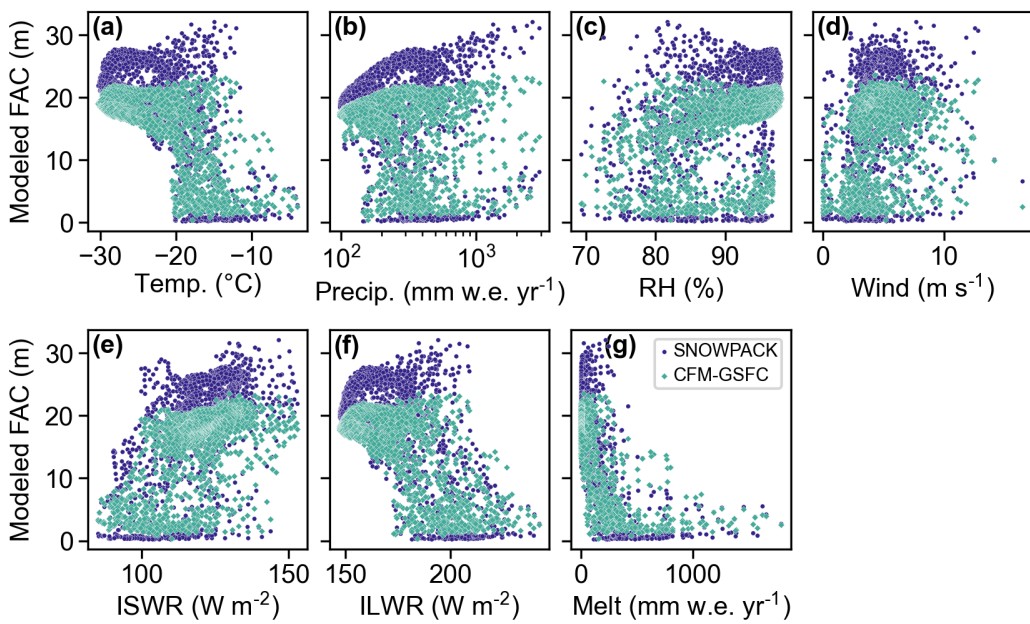

**Figure A4.** Modeled firn air content (FAC) in SNOWPACK (blue) and the CFM-GSFC (green) as a function of the forcing variables: (a) temperature, (b) precipitation, (c) relative humidity, (d) wind, (e) incoming shortwave radiation (ISWR), and (f) incoming longwave radiation (ILWR) all calculated for 1980 through 1995. Also shown is (g) the melt, which is calculated by SNOWPACK's surface energy balance model and used as a forcing in the CFM-GSFC.

*Code and data availability.* The NASA GSFC MERRA-2 data are available at https://disc.gsfc.nasa.gov/. The SUMup snow density sub-dataset can be found at https://arcticdata.io/catalog/view/doi%3A10.18739%2FA2NP1WK6M. Model code is available on GitHub; the code to run the SNOWPACK firn model is available at https://github.com/snowpack-model/snowpack and the code to run the Community Firn Model (CFM-GSFC) is available at https://github.com/UWGlaciology/CommunityFirnModel. Data for the figures can be found on Zenodo at https://doi.org/10.5281/zenodo.7671892.


*Author contributions.* MTM and NW ran the SNOWPACK model, and CMS ran the CFM-GSFC. MTM processed and analyzed observational data as well as output from both models, and led the manuscript writing. JTML and BM led the study design. All authors contributed to the writing of the manuscript.

*Competing interests.* The contact author has declared that none of the authors has any competing interests.

*Acknowledgements.* The authors acknowledge Eric Keenan for his assistance in accessing MERRA-2 data and compiling the SNOWPACK model. This work used the RMACC Summit supercomputer, which is supported by the National Science Foundation (awards ACI-1532235 and ACI-1532236), the University of Colorado Boulder, and Colorado State University. The Summit supercomputer is a joint effort of the University of Colorado Boulder and Colorado State University.

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
