# Peer review of "An evaluation of a physics-based and a semi-empirical firn model across the Greenland Ice Sheet (1980–2020)"

_The Cryosphere, 2022_

## Referee Comment (RC1)

**Summary of study and overall assessment**

In this study, the firn properties of Greenland are simulated for the period 1980 – 2020 with two different models – the semi-empirical Community Firn Model (CFM) and the physically-based SNOWPACK (SP) model. Both models are driven with atmospheric forcing from MERRA-2 reanalysis and applied on the same spatial grid (~0.5°) as the reanalysis data. To equilibrate the firn layer, the models were first spun-up with forcing data from the so-called reference climate interval (RCI), which ranges from 1980 to 1995. Subsequently, the actual simulations with CFM and SP were performed and evaluated with 767 firn cores from the SUMup project. After demonstrating the good overall performance of both models, results are analysed and intercompared with a focus on firn air content (FAC) and its temporal evolution (interval means, inter-/intraannual changes) and spatial distribution on basin scales.

Firn models are important tools, because they allow a spatially comprehensive assessment of ice sheet's average firn porosity. This quantity is essential for estimating the potential of the ice sheet to retain meltwater in the firn layer (→ buffer effect) and thus slowing down global sea level rise. It is therefore crucial to have a good understanding of how well firn models of different complexity simulate firn related processes. This manuscript adds interesting results and findings to previous work. The study is generally well written and structured and the figures are of excellent quality. Find below some suggestions to improve the manuscript – most comments are of minor nature and concern details.

**General comments**

**Conclusion section**
In my opinion, the conclusion section needs some improvement. The structure seems currently a bit chaotic – e.g. the part with the outlook ("This will in turn allow us to better predict the firn's response to future warming.") should rather be at the end of the section. I suggest to rearrange this section in a more logical way. Furthermore, the following points could be included/extended:

- Embed findings in a larger picture (and discuss further implications). For instance, I guess the computational cost of running the physically-based SNOWPACK model is substantially higher (could you state how much approximately?). Does the higher complexity (e.g. explicit consideration of effects like wind compaction under drifting/blowing snow that influence new-snow density) "pay off" (i.e. add some distinctive benefits)?
- State recommendations for future (similar) studies and extend outlook. For instance, which are the most crucial processes in firn model that should be better represented in future models (I have in mind processes like vertical (or even lateral) water flow, reduced permeability of ice layers/slabs, ponding water conditions in firn aquifers, etc.)?

**Point-comments**

**Content-related (text)**
**Line 10:** For which time are these statements valid? 1980, 2020 or averaged over the 40 years?
**L84:** I would call this section "**Methods and data**" (because you also present the SUMup observations)
**L88:** Do you consider both snow- and rainfall data from MERRA-2? Or do you derive precipitation fractions (solid/liquid) with an air temperature threshold?
**L95:** I would explicitly state that MERRA-2 was also considered in Zhang et al. (2021) – this is not obvious from the current statement. Maybe you could also briefly summarise how the model performs with respect to Automatic Weather Stations (AWSs) data.
**L99:** I have a general question (just out of curiosity – no changes regarding this question are required for the current manuscript): SNOWPACK and CFM inherit MERRA-2's spatial grid. However, one could also apply a different (unstructured) grid, which e.g. has a higher spacing close to the ice sheet's margins. With this, one could better capture areas with strong climate gradients and the complex boundary of the ice sheet (which might also reduce the disagreement in total glaciated area). However, such a solution might anyway only be relevant if a generally higher grid spacing than 0.5° is used (also in terms of atmospheric forcing data). What's your option on this idea for future firn model applications to the GrIS?
**Section 2.2:** Could you specify which scheme for vertical water percolation is applied in SNOWPACK?

**L125:** It might be useful to refer to Fig. A2 here (time series in the grey-shaded areas show no (strong) temporal trends, which supports the definition of the RCI period)

**L135:** Why do you perform the vertical interpolation only for CFM output (and not for SNOWPACK – which also has a fine grid spacing)?

**L150:** Why do you apply different spin-up conditions for SNOWPACK and CFM? Is it due to computational constraints (i.e. that SNOWPACK is more expensive to run)?

**L169:** Here, you neglect any liquid water in the firn – right? Compare e.g. to Eq. (6) and (7) in Kuipers Munneke et al. (2015).

**L174:** Why do you use 100 m as a lower limit (and not e.g. 150 m – the spin-up depth of SNOWPACK)?

**L181:** I would briefly explain what the NSE range (<0, 1, etc.) means for the model (because most readers are probably unfamiliar with this metric)

**L269:** "no change" might be a bit too restrictive. Maybe better "only negligible changes"

**L289:** I would shift this first paragraph (maybe to the end of this section?). For me, these first lines suggest that it is not interesting to look at trends because there is no significant change in FAC between the two periods. However, looking e.g. at Fig. 8, there seems to be a clear trend during the latter period which is definitely worthwhile to discuss. Anyway, I have to admit that I'm not an expert on statistical methods, so there might be a reason why you start with comparing the two periods statistically…

**L323:** I think it would be more robust to look at linear trends here. Computing the difference between two (somehow arbitrary selected years) is prone to noise introduced by interannual variability...

**L445:** I'm not able to follow this sentence. Do you mean "intensified firn densification"? And why does that increase the firn's cold content?

**Typos, phrasing and stylistic comments**

**Line 6:** …Community Firn Model (CFM)**,** to quantify…

**L15:** This sentence reads odd somehow. It might be better to add the negative rates to the previous sentence and then state: "The reduction in spatially-integrated FAC in SNOWPACK and CFM demonstrate how model differences propagate throughout the FAC record."

**L117:** "scheme use to" → "scheme use**d** to"

**L195:** "form**ed**"

**L197:** "in the surface" → "close to the surface"?

**L364:** I would change this to something like: "The five locations shown in Figure 4 lie all within the same MERRA-2 grid cell and thus share the same atmospheric forcing data for the models."

**L366:** change "MERRA-2 grid point" to "MERRA-2 grid cell" (also later in the text)

**L374:** "in simulating observations" → "in reproducing observations"

**L375:** I would rephrase this sentence.

**L443:** I would rephrase this sentence.

**L461:** I would rephrase this to something like: "For both models, the summer air temperature seems to be a good proxy for the abrupt drop in FAC, which happens at temperatures between approximately -4 to 0° Celsius."

**Figures and Tables**

**Figure 1:** Adding degree symbols and N/E to the latitude/longitude coordinates would help the reader.

**Figure 2:** I would state relative biases in percentages like specified in Eq. (4)

**Figure 4:** I would change "MERRA-2 domain" to "MERRA-2 grid cell" and in the caption: "MERRA-2 grid point" → "MERRA-2 grid cell"

**Figure 6:** caption → what caused the missing data?

**Table 3:** How did you distinguish between detectable and undetectable signals?

**Fig. A1:** caption: this means you only consider SUMup observations for this analysis in which the upmost density measurement covers the topmost 0.1 m or less – right? Furthermore, I would change the following sentence slightly: "The CFM uses a prescribed surface density of 350 kg m$^{-3}$ **(green vertical line)**, which falls near many of the observed surface densities."

**Fig. A3:** caption: I'm not able to follow the anomaly calculation. Wouldn't subtracting each year's mean from the record lead to discontinuities in the time series? And wouldn't it be easier to simply detrend the time series? Because this part is methodological a bit more complex (see also my

comment to Table 3), it might even be worth to move this part to a separate section in *2. Methods and data*.

**New references**

Kuipers Munneke, P., Ligtenberg, S. R. M., Noël, B. P. Y., Howat, I. M., Box, J. E., Mosley-Thompson, E., McConnell, J. R., Steffen, K., Harper, J. T., Das, S. B., and van den Broeke, M. R (2015).: Elevation change of the Greenland Ice Sheet due to surface mass balance and firn processes, 1960–2014, The Cryosphere, 9, 2009–2025, https://doi.org/10.5194/tc-9-2009-2015

---

## Referee Comment (RC2)

Review of Thompson-Munson et al.: *Observed and modeled Greenland firn properties (1980–2020)*
by Vincent Verjans.

This study applies two state-of-the-art firn models at the scale of the Greenland Ice Sheet (GrIS). The Community Firn Model (CFM) is used with the semi-empirical firn densification formulation NASA GSFC-FDMv1 of Medley et al. (2022). SNOWPACK is a more physically-detailed snow compaction model. The goal of this study is to compare results from these two different approaches to firn modeling over the GrIS. The authors also perform a comparison of model output against in-situ firn core observations.

I believe that this study demonstrates a comprehensive modeling effort, and the results are a valuable large-scale comparison of firn model behavior. The modeling experiments are rigorous, well-explained, and are undoubtedly a great contribution to the firn modeling community. The use of a same atmospheric forcing allows to identify differences only related to firn model and to parameterization choices. The figures are of good quality. And the authors perform a thorough evaluation by using an extensive dataset of 766 firn cores. However, I believe that there is a problem in the methodology of the evaluation, as I explain in this review. I appreciate the modeling effort and the results of this study, but I have some reservations concerning the interpretation and the lack of in-depth investigation of the structural differences between the two models. In other words, the results are excellent, but what can we conclude from this study? What is the main message for the firn modeling community? I believe that with a little more analysis, this study can be much more than simply providing model output from two models at the GrIS scale. This work is in the scope of The Cryosphere, and I welcome its publication pending some revisions.

I have separated my review in Major comments that require a re-evaluation of some steps of the study, Minor comments that require more clarity in the manuscript and/or small changes, and Specific comments, which focus on specific aspects, and are mostly of technical nature. Despite my numerous comments, I strongly encourage the authors to re-submit the manuscript after the revisions have been made.

Major comments
1) Problems of the evaluation
The use of 766 firn cores in the evaluation process is noteworthy. However, as pointed out by the authors (l.207-208):
"*Since most observations are from shallow cores (median depth = 2.0 m; Fig. 2a) the observed FAC values are relatively low (median FAC = 1.3 m; Fig. 2b) and do not represent the FAC of the full firn column.*"
This skewed distribution of the observations make the evaluation very biased and difficult to interpret. In any modeled firn profile, FAC in the upper two meters is essentially dictated by the surface density ($\rho 0$). In the CFM, $\rho 0$ is fixed to 350 kg m$^{-3}$. This is purely a choice of the authors, as any other constant value or parameterization of $\rho 0$ would be equally valid (e.g., Kuipers Munneke et al., 2015; Fausto et al., 2018; Medley et al., 2022). Thus, the statistics of the evaluation (NSE, relative bias) essentially reflect how well the choice of $\rho 0$ fixed to 350 kg m$^{-3}$ fits the SUMup surface densities, rather than showing the performance of the CFM GSFC-FDMv1 densification scheme. Similarly for SNOWPACK, the evaluation reflects the performance of the surface density scheme compared to the SUMup surface densities, and not its performance in densification physics.
Another issue with using FAC as an evaluation metric is that the best predictor of FAC is the core depth. However, this does not bear any information about model performance. For example, a pair of modeled and observed FAC values over 2m depth will (almost) always be close to each other, and a pair of modeled and observed FAC values over 20m depth will also (almost) always be relatively close to each other. The good correlation between modeled and observed FAC values is due to cores being compared over a same depth. This problem arises because the authors have decided to use all the SUMup cores in the evaluation, and not to restrict their analysis to cores with a minimum depth threshold.
In order to alleviate these two problems, I encourage the authors to make an evaluation by binning cores based on their depth. A separate evaluation for each depth bin should be performed. For example, all the cores can be separated in groups of depth<2m, 2m<depth<5m, 5m<depth<15m, depth>15m or something similar. The binning should be made appropriately in order to have sufficient cores in each bin, but also meaningful evaluation

statistics at the same time. Furthermore, I would like the authors to highlight more clearly that the CFM results at low FAC values are mostly determined by the ρ0 choice and not by the GSFC-FDMv1 densification scheme. In the current version of the manuscript, this is not clear for readers less familiar with firn modeling. Finally, the authors point out (l197-200): "*SNOWPACK simulates more variability between layers compared to the CFM. This partly results from the fixed surface density of 350 kg m⁻³ set for the CFM, while the surface density in SNOWPACK varies based on atmospheric conditions, and partially because the CFM outputs are interpolated onto a grid.*" This is important and needs to be quantified. How much of this low-variability error is due to the ρ0 assumption? And how much is due to the interpolation? I strongly recommend to run the CFM at some firn core locations with ρ0 set to the SNOWPACK ρ0 time series, and without the interpolation scheme. This would bring better insights into the impact of these aspects.

2) Interpretation of the results
This study is a model intercomparison. I believe that this warrants more discussion of why the models diverge, and what conditions make them more prone to diverge/agree.
This firstly necessitates a better description of the model physics. The governing equations (densification, heat conduction, etc.) should be provided in the manuscript or in an Appendix. Based on these equations and on their results, the authors should provide some explanation on the different sensitivities of CFM-GSFC-FDM and SNOWPACK to temperature, accumulation, melt, wind forcing, etc. For example, I found Figure 9 very interesting. But the analysis does not tell why GSFC-FDM and SNOWPACK agree well in the Northeast and Southeast, but show strong discrepancies in the Central West and Norhwest. As another example, from Figure 5, why does SNOWPACK simulate much larger FAC at low summer temperatures than CFM-GSFC-FDM? Throughout the manuscript, I have been somewhat frustrated by the dichotomy between impressive results but lack of in-depth explanations.
Finally, the authors have related the sensitivity of both models to climatic conditions (LTSR and summer temperature) in the steady-state climate configuration. It would be interesting to expand such an analysis to the transient climate configuration. This would involve quantifying the sensitivity of FAC loss/gain to changes in atmospheric forcing.
As a final note concerning this Major comment 2, I should emphasize that addressing the sort of questions that I raise is not an absolute necessity for publication. The study is already a thorough modeling effort, with a good quantitative evaluation of the results. I simply believe that a thorough analysis of model behavior would bring this study to the next level.

Minor comments
1) References
I find that this study does not sufficiently recognize previous work from the firn science community. I provide here some examples, but I also encourage the authors to proceed to a more in-depth literature review, and to cite other previous relevant studies in their manuscript.
- l42-43 "*Changes in the amount of air-filled pore space within the firn, known as the firn air content (FAC), have been investigated in both observations (e.g., Vandecrux et al., 2019) and models (e.g., Medley et al., 2022).*": please cite Benson (1962); Braithwaite et al. (1994); Sorensen et al. (2011); Kuipers Munneke et al. (2015); etc.
- l48-49 "*Modeling firn has become important for estimating mass balance (MB) from satellite altimetry, since this method relies on firn models to interpret the causes of surface height changes (e.g., Li and Zwally, 2011).*": please cite Arthern and Wingham (1998); Morris and Wingham (2014).
- l52-53 "*Additionally, understanding the limits and deficiencies in firn models is essential for quantifying uncertainties in altimetry-based MB estimates.*": please cite Morris and Wingham (2015); Verjans et al. (2021).
- l57-59 "T*hese models use empirical relationships between densification, accumulation, and temperature, and they are often tuned to observations (e.g., Ligtenberg et al., 2011; Medley et al., 2022; Li and Zwally, 2011).*": please cite Herron and Langway (1980); Arthern et al. (2010); Simonsen et al. (2013); Verjans et al. (2020)
- l68-69 "*Still, both semi-empirical as well as physics-based firn models have been successfully used in Greenland (e.g., Vandecrux et al., 2020b; Dunmire et al., 2020; Medley et al., 2022).*": please cite Sorensen et al. (2011); Kuipers Munneke et al. (2015).

l71-72 "*At an ice-sheet scale, few comparisons of semi-empirical and physics-based models exist*": please recognize the work of Steger et al. (2017) here.

-l356-357 "*Neither model captures the high densities resulting from the firn aquifer because the use of bucket scheme in the models prevents full saturation in the firn.*": when discussing this aspect, please note that firn aquifer formation has been modeled by Verjans et al., (2019) and that conditions for aquifer development have been investigated by Kuipers Munneke et al. (2014).

- l361-362 "*Our results agree with these findings that model differences are highest where liquid water is present, indicating that poor representation of meltwater percolation processes is still a substantial limiting factor in firn model performance.*": please cite Verjans et al. (2019).

- l370-372 "*This demonstrates the limitations of a coarsely-gridded forcing, especially in steeply sloped areas where climate is likely to be highly variable within a single grid cell*.": please mention the downscaling work of Noël et al. (2016).

- l419-420 "*This could be related to the fact that SNOWPACK was developed using data from seasonal alpine snow which may not be representative of the physical processes driving deep firn densifcation.*": please cite Maeno and Einuma (1983); Arnaud et al. (2000).

2) The Reference Climate Interval (RCI)

The use of an RCI is necessary for the spin-up of firn models. However, this implies assumptions which must be properly understood, explicitly stated, and discussed. Here, the authors state (l126-127): "*We make the assumption that this period is representative of the longer-term Greenland climate.*" And further, they state (l225-227): "*The RCI used for model spin-up spans 1980 through 1995, and since we assume that this period represents a relatively steady-state, long-term Greenland climate (Fig. A2)*". I believe that the message conveyed to the readers about the RCI is misleading. The RCI is used to develop the initial model firn column, from which transient experiments over the period of interest (1980-2020) start. As such, ideally, the initialization should be computed with the true climate forcing of the decades and centuries preceding 1980. This is true regardless of whether the long-term conditions were in steady-state (i.e., without trends) or not. In other words, the "perfect" RCI should not represent steady-state conditions if the true conditions were not in steady-state prior to 1980. However, we have only incomplete knowledge of the true climate, especially prior to 1980. In light of this incomplete knowledge, using steady-state conditions over the RCI is a reasonable simplification, but not a necessary condition for a valid firn model initialization procedure. In the manuscript, it should be clearer that steady-state conditions over the RCI are used in order to isolate effects of climatic deviations from the RCI on firn column changes. But such steady-state conditions are not representative of true conditions, and the true changes in firn thickness are influenced by the unsteady nature of past climate conditions. I would like the authors to mention these points in the Discussion, as well as other studies that have shown that the pre-1980 climate of GrIS was not in steady-state (e.g., Hanna et al., 2011).

3) Evaluation of sensitivity to climatic changes

The evaluation of model performances is performed with firn core data. Because firn cores only provide a snapshot of firn density in time (as pointed out by the authors), they cannot serve to evaluate the sensitivity of firn models to changes in climatic forcing. For example, good performance of a model when evaluated against firn cores does not imply that the model would accurately capture FAC changes under a +1°C change in mean surface temperature. This is particularly important to keep in mind when firn models are used to compute FAC change estimates in time, as done in this study. Evaluation with firn core data is legitimate given the scarcity of transient in-situ firn data, however I think that this limitation of the evaluation process deserves a paragraph in the Discussion section.

4) Clarification about the "CFM"

The authors repeatedly use the name "CFM" for one of the two firn models used. However, as far as I am aware of, the CFM allows to choose among various different firn densification formulations. Thus, I would find it more appropriate to call the model CFM- GSFC-FDM (or something similar). This is important because much of the FAC patterns are due to the use of the GSFC-FDM densification scheme, and not to the CFM itself, which is

simply a numerical tool. Furthermore, the authors state that (l144-145) "*The densification rate is determined with the NASA GSFC-FDMv1 firn densification equation (Medley et al., 2022)*". If this is the case, I suggest that they specify GSFC-FDMv1.2.1 to avoid any possible confusion with other versions of GSFC-FDMv1. Also, the CFM allows for a range of thermal conductivity parameterizations, the choice of which likely influences the results. I ask the authors to specify the thermal conductivity parameterization used. Finally, it should be clearer in the manuscript that the CFM itself is a numerical tool, and that the CFM output thus largely depends on the parameterization of the CFM (densification scheme, ρ0, thermal conductivity), and not on the CFM itself. This is important for readers less familiar with the CFM.

Specific comments
Title
This study does not bring any new observations about Greenland firn properties. For this reason, I find the use of the word "*Observed*" in the title inadequate.
l8
Specify: "isolate firn model differences".
l9
"*Both models perform well*": this needs to be quantified.
l10
"*is hindered by meltwater percolation*": this not really evaluated in this study. The authors only analyze in details the performance at firn aquifer sites, but do not compare performance in dry areas versus percolation areas more generally.
l10
Change "*the full ice-sheet*" to "ice-sheet-wide".
l11
Please move "*(i.e., air volume in the firn)*" to the line where FAC is used for the first time.
l13
Change "*the models' treatment of atmospheric input*" to "the sensitivity of the models to atmospheric forcing".
l15-16
Specify "spatially-integrated FAC decrease of".
l22
Change "*in a thick*" to "by a thick".
l23
Change "*density of firn varies across the ice sheet*" to "density of firn varies in depth and across the ice sheet".
l24
Remove "*in time and*".
l25
Specify: "can buffer the contribution of increased melt rates to sea-level rise".
l26-27
I find this sentence confusing, as it is not specifically about firn. Please consider rephrasing.
l31
To my knowledge, we are not sure whether ice slabs make deeper pore space completely inaccessible. I recommend using "potentially inaccessible".
l39
Change "*firn has lost its capacity to store meltwater*" to "the meltwater storage capacity of firn has abruptly decreased".
l46
Change "*the memory effect of changes to the firn from previous years*" to "firn changes evolving on multi-year timescales".
l52
Change "*measured in satellite altimetry*" to "measured from satellite altimetry".
l59

"*Semi-empirical models are beneficial because they do not rely on the physics of firn densification*": this statement is too crude and needs more nuance (see Arthern et al., 2010 for example).

l63-64

What is "*the constitutive relationship*"? As far as I know, even formulations linking stresses to firn strain rates rely on some form of parameterization, and there is no single universal constitutive relationship.

l65

Change "*observations from tuning*" to "observations for tuning".

l67-68

"*since snow physics have been more-thoroughly studied*": this requires one more line of explanation, and also remove the hyphen.

l68

"*have been successfully used*": what do the authors mean by "*successfully*"?

l70-71

"*have seen significant development for polar regions in recent years*": can the authors please list some of these developments?

l78

Remove "*completely*".

l94-95

"*Regional climate models are not always widely available or regularly updated, and no single reanalysis clearly outperforms others over the GrIS (Zhang et al., 2021).*": either provide more details, or simply say that the method could equally well be applied with a regional climate model or with another reanalysis product.

l95-97

I think that these two sentences should be rephrased as they do not read very well.

l99

Replace "*full ice-sheet*" by "ice-sheet-wide".

l99

Typo: "*gird*".

l105

What do the authors mean by "*successfully*"?

l109-110

"*the constitutive relationship*": see comment above.

l113-114

"*SNOWPACK uses the MeteoIO library (Bavay and Egger, 2014) for preparing the meteorological forcing data for the simulations.*": please explain.

l115-116

I don't think that storing output every 7 days conserves "*computational expenses*", but only reduces storage size. Please correct this statement, or provide explanations if I am wrong.

l117

"*impacted by the layer-merging scheme*": please explain.

l117

Typo: "*use*" should be "used".

l119-120

"*We set the surface roughness to 0.002 m for solving the energy balance with the Michlmayr et al. (2008) stability correction when a stable boundary layer is diagnosed.*": please explain more.

l121

Specify: "varies across the GrIS".

l134-135

"*The CFM uses a layer-merging scheme at 5- and 10-m depth to reduce computational demands.*": please explain more.

l149

"*was chosen to be near the depth at which the firn reaches the ice density*": please be more specific.

l151

Remove "*For example, if the firn needs 1000 years to spin up, the RCI would repeat 63 times.*".

l166

Remove "*(also known as "depth-integrated porosity (DIP)")*".

l168

Specify "where z = 0 m represents the surface, and is increasing downwards".

Section 3.1

Please re-order this section. The authors start with specific results at two individual cores and then provide the general results at the GrIS-scale. I recommend starting from the general results, and then focusing on specific results.

l193

"*collected in southwest Greenland*": please specify the core date also in the main text.

l197

Change "*in the surface*" to "near the surface".

l200-201

"*For polar regions in particular, temporal variations in wind and the presence of drifting snow translate into vertical density variations with increasing accumulation.*": I do not understand "*with increasing accumulation.*". Also, this statement requires some references.

l211

Specify: "are overestimating FAC on average".

Section 3.2

In general, when comparing SNOWPACK and CFM results, I strongly recommend using the Root Mean Squared Deviation metric. This would provide more quantitative information in the analysis.

l232-233

Remove this sentence as this is repetitive information.

Equation (6)

Does "*snow*" account for sublimation and blowing snow? And are these fluxes identical for SNOWPACK and CFM?

Figure 1b

Notice that SNOWPACK does not underestimate density at high depth here. This is interesting to me, and warrants more analysis about the model physics (see Major comment 2).

l242

Here and everywhere in the manuscript, change "*an FAC*" to "a FAC".

l244

Remove "*the*" in "*the FAC values*".

l246

Does "*response*" refer to FAC or to something else? Please avoid any confusion.

l248

"*generally similar spatial patterns in FAC across the ice sheet*": please quantify.

Figure 2

Please use log color scale in Fig 2c and Fig 2d. With the current color scale, almost all the data points simply appear white.

Figure 3

Because the equations of the linear regressions are not discussed in the main text, I recommend removing the black solid lines and the regression equations from Figure 3.

l253

Specify: "is on average greater than".

l253-254

"*difference between the models increases with depth*": please quantify.

l257

Refer to an appropriate Figure when introducing the basins.

l262

"fall in the middle": please rephrase.

l271

Change "*between the two years*" to "over the RCI".

l271-272

Change "*which verifies the steady-state assumption of the RCI and the design of the spin-up (Table 2).*" to "which is a consequence of our choice of RCI and of the design of the spin-up (Table 2)."

l274-277

Much of this information is repetitive information with respect to the paragraph above. Furthermore, please be more careful about the impacts of the RCI assumptions when discussing these results (see Minor comment 2).

l276

Change "*Greenland's sea level rise buffering capacity*" to "the sea-level rise buffering capacity of the Greenland firn layer".

l280

"*somewhat*": please quantify.

l282

Throughout their manuscript, the authors use the term "*seasonal breathing*" to designate the seasonal fluctuations in FAC. I know that this term has been used in some previous studies. However, I personally dislike this term. In my view, it is scientifically incorrect: firn does not breathe. I would appreciate if the authors replace this term by another one. For example: the FAC seasonal amplitude. I thank the authors for their understanding.

l285-286

"*the seasonality was undetectable by the chosen methods*": please explain in Figure A3 why the seasonality was undetectable in some cases.

Caption of Figure 5

Add comma: "air temperature, all calculated".

Figure 6

Why are is there missing data from one or both models in some areas? Please explain in the main text.

Caption of Figure 6

Please rephrase the first sentence of the caption to make it more intelligible grammatically.

Caption of Figure 7

Specify: "The bottom row also includes the percent difference averaged over the GrIS."

Caption of Table 1

Specify: "and the average percent difference between".

Table 1

Provide root mean squared deviation between CFM-GSFC-FDM and SNOWPACK in each basin. And please discuss these values in the main text.

Caption of Table 2

Specify: "The average percent change between".

Table 2

Firstly, I am surprised by the magnitude of changes between 1980 and 1995 (up to 0.5% in magnitude). How is this possible given that the 1980-1995 simulation is simply a repetition of the climatic RCI loop imposed during the spin-up. As far as I understand the spin-up process, repeating the RCI once more should cause only negligibly small changes. Can the authors please explain this?

Secondly, I think that for both periods (1980-1995 and 2005-2020), it would be more relevant to analyze the trend over the period instead of the change between two individual years. Analyzing only two individual years means that conclusions can be influenced by inter-annual variability. This may explain my first point of this comment.

l292

Change "*statistically*" to "significantly".

l297-298

"*A consistent decreasing trend is modeled from ∼2002 and through ∼2011.*": please keep the 2005-2020 as a baseline for analysis. Switching between different periods of analysis makes the messages more confusing.

l305

Change "*associated water percolation processes*" to "associated water percolation and refreezing processes".

l312

"*SNOWPACK simulates greater negative changes in FAC compared to the CFM once the models diverge.*": I find this sentence unclear. Please consider rephrasing.

l314

Please specify: "followed by an increase in FAC only in the northeast and southeast".

l317-329

The differences between CFM-GSFC-FDM and SNOWPACK that are described in these two paragraphs are important. This deserves more detailed investigation into the causes of these differences (see Major comment 2).

l335-336

"*the ice slab nearest to the surface, which in some cases could be bare ice at the surface since there is no condition that the ice slab must be beneath a layer of snow or firn*": I apologize, but I will make another pedantic comment. The term "ice slab" has been used a lot over the last 6 years, and it is now often used inappropriately. Ice slabs are thick layers of ice that develop within a layer of porous firn. They should not be confused with the expansion of the ablation area. In this study, the algorithm of the authors makes no difference between development of ice slabs and ablation area extension. For this reason, I ask the authors to replace their use of "*ice slabs*" by "ice slabs or ablation area extension".

l340

Change "*depth to those slabs*" to "depth of those slabs".

l343-344

Please re-evaluate this following Major comment 1.

l350-351

"*The signature of model biases differs across the ice sheet as climate, topography, and the impact of firn hydrology vary.*": where is this assessed specifically?

Caption of Figure 9

Please add "note the different y-axis scales."

l353

"*Some of the highest model biases in SNOWPACK and the CFM occur in southeast Greenland*": please quantify.

l354

Please clarify: "First, some of the observed density profiles are from cores that were drilled directly into a perennial firn aquifer (Miller et al., 2018)."

l361

"*Our results agree with these findings that model differences are highest where liquid water is present*": please quantify the performances of the models in conditions of high LTSR.

l361

Please specify: "where liquid water is abundant".

l366

"*Within these five cores*": please rephrase.

Figure 10

I am puzzled by the big differences in the northeast ice caps. I wonder how much this influences the results provided in this study. If this influence is significant, the authors should reconsider the presentation of their results. The focus of this study is "*the Greenland ice sheet*" and not "the Greenland ice sheet and its surrounding ice caps".

l368

Change "*observed*" to "local".

l374-375

"*gives confidence in the models' abilities to simulate firn properties across the full ice sheet.*": please nuance this statement as the performances of the models in the percolation zone are limited.

l379

Specify: "in the firn models themselves."

l382-383

"*the LTSR is a stronger predictor of FAC in the CFM compared to SNOWPACK*": please quantify using the coefficient of determination.

l383-384

"*The large range of possible SNOWPACK-simulated FAC values at low LTSR values is likely due to the model's sophisticated new-snow density scheme (…)*": thus, is the difference in LTSR sensitivity between CFM-GSFC-FDM and SNOWPACK mostly caused by the ρ0 assumption for CFM-GSFC-FDM? (see Major comment 1)

l386

Change "*This indicates*" to "Our results show".

l393

Typo: "*has*" should be "have".

l396

"*which requires more detailed output from a dedicated firn model*": please mention that this is only a particularity of MERRA2 because it does not provide melt as an output.

l400

"*where the spread in FAC is less than in SNOWPACK*": please quantify.

l402

Change "*is easier to predict using LTSR*" to "shows less variability for a given LTSR".

l409-410

The total Greenland FAC values provided here do not agree with the values given in Table 2. Please correct this.

l411-412

"*is close to a regional climate model's (HIRHAM5_MOD) estimate for this period (Vandecrux et al., 2019).*": please provide the value.

l416

"*uses the constitutive relationship between stress and strain*": see comment above.

l422-423

Remove "*(and by proxy, whether the physics-based or empirical approach is recommended)*".

l427

Change "*corresponds with*" to "corresponds to".

l429-432

"*The stronger seasonality in the CFM is indicative of the model's more simple treatment of forcing data like accumulation and temperature, which have strong seasonal patterns. SNOWPACK's same sophisticated new-snow density scheme that leads to a complex relationship between LTSR and FAC also results in this smaller seasonal signal.*": it is unclear to me how the authors reach these conclusions from their results. Please explain more.

l434

"*the 2012 extreme melt season can be seen as an abrupt drop in FAC in most basins*": please be more nuanced. There is an abrupt drop only in the southwest, southeast, and to a lesser degree northeast basins.

l441

Please be more nuanced: "Ice slabs, which may render deep pore space inaccessible".

l443-444

Please rephrase this sentence.

l445-447

"*Pore space depletion can also be a sign of firn densification, which has been found to increase cold content in the firn and amplify meltwater freezing and ice slab formation in the near-surface (Vandecrux et al., 2020a).*": this statement is a very crude simplification of a complex process with many interactions and feedbacks. I suggest rephrasing: "Pore space depletion can also be caused by firn densification, which in turn modifies the meltwater refreezing and retention capacities of the firn in a complex manner.*"

l459

"*place our results in a context of uncertainty*": this study does not perform any uncertainty analysis.

l460-461

Change "*lets us isolate the differences in the models themselves and examine how they respond to the same forcing*" to "lets us examine how the models respond differently to the same forcing due to structural model differences and parameterization choice".

l461

Change "*metric*" to "factor".

l464

Change "*While FAC in the CFM more gradually decreases as*" to "While FAC more gradually decreases in the CFM as".

l465

"*reaches near-zero FAC values that the CFM does not capture*": please consider rephrasing. This sentence suggests that SNOWPACK is right and that CFM is wrong.

l466

Change "*even more*" to "on larger areas".

l470

The FAC values given here do not agree with values in Table 2. Please correct this.

l471

"*reasonable estimations when compared to other studies*": please quantify.

l477

Change "*the pore space depletion is more extreme*" to "the pore space is more depleted".

l480

Specify: "the firn layer in the CFM loses only an equivalent of 1 mm."

Caption of Table A1

Does the standard deviation refer to (a) the within-basin standard deviations of the RCI mean values or (b) the annual standard deviation of the basin mean values? If it is (a) specify "mean plus/minus basin standard deviation", and if it is (b) specify "mean plus/minus inter-annual standard deviation".

Caption of Table A2

Same comment as for the caption of Table A1.

Caption of Figure A1

Move "*(green vertical line)*" next to "*350 kg m⁻³*".

Caption of Figure A2

Change "*full ice sheet*" to "Greenland ice sheet".

Section "*Code and data availability*"

I thank the authors for providing the entire model output from both models as an open-source dataset. However, I ask the authors to include a README file in the dataset that explains in details how one can read and extract information from the files.

References used in this review

Arnaud, L., Gay, M., Barnola, J.-M., and Duval, P.: Physical modeling of the densification of snow/firn and ice in the upper part of polar ice sheets, in: Physics of Ice Core Records, edited by: Hondoh, T., Hokkaido University Press, Sapporo, Japan, 285–305, 2000.

Arthern, R. J. and Wingham, D. J.: The Natural Fluctuations of Firn Densification and Their Effect on the Geodetic Determination of Ice Sheet Mass Balance, Clim. Change, 40, 605–624, https://doi.org/10.1023/A:1005320713306, 1998.

Arthern, R. J., Vaughan, D. G., Rankin, A. M., Mulvaney, R., and Thomas, E. R.: In situ measurements of Antarctic snow compaction compared with predictions of models, J. Geophys. Res.-Earth Surf., 115, 1–12, https://doi.org/10.1029/2009JF001306, 2010.

Benson, C. S.: Stratigraphic Studies in the Snow and Firn of the Greenland Ice Sheet, U.S. Army Snow, Ice and Permafrost Research Establishment (SIPRE–CRREL), Research Report 70, reprinted with revisions by CRREL, 1996, 1962.

Braithwaite, R., Laternser, M., and Pfeffer, W. T.: Variation of near-surface firn density in the lower accumulation area of the Greenland ice sheet, Pâkitsoq, West Greenland, J. Glaciol., 40, 477–485, https://doi.org/10.3189/S002214300001234X, 1994.

Fausto, R. S., Box, J. E., Vandecrux, B., van As, D., Steffen, K., MacFerrin, M., Machguth H., Colgan W., Koenig L. S., McGrath D., Charalampidis, C., and Braithwaite, R. J.: A Snow Density Dataset for Improving Surface Boundary Conditions in Greenland Ice Sheet Firn Modeling, Front. Earth Sci., 6, 51 pp., https://doi.org/10.3389/feart.2018.00051, 2018

Hanna, E., Huybrechts, P., Cappelen, J., Steffen, K., Bales, R. C., Burgess, E., McConnell, J. R., Peder Steffensen, J., Van den Broeke, M., Wake, L., Bigg, G., Griffiths, M., and Savas, D.: Greenland Ice Sheet surface mass balance 1870 to 2010 based on Twentieth Century Reanalysis, and links with global climate forcing, J. Geophys. Res.-Atmos., 116, D24121, https://doi.org/10.1029/2011JD016387, 2011.

Herron, M. and Langway, C.: Firn densification: an empirical model, J. Glaciol., 25, 373–385, https://doi.org/10.3189/S0022143000015239, 1980.

Kuipers Munneke, P., Ligtenberg, S. R. M., van den Broeke, M. R., van Angelen, J. H., and Forster, R. R.: Explaining the presence of perennial liquid water bodies in the firn of the Greenland Ice Sheet, Geophys. Res. Lett., 41, 476–483, https://doi.org/10.1002/2013GL058389, 2014.

Kuipers Munneke, P., Ligtenberg, S. R. M., Noël, B. P. Y., Howat, I. M., Box, J. E., Mosley-Thompson, E., McConnell, J. R., Steffen, K., Harper, J. T., Das, S. B., and van den Broeke, M. R.: Elevation change of the Greenland Ice Sheet due to surface mass balance and firn processes, 1960–2014, The Cryosphere, 9, 2009–2025, https://doi.org/10.5194/tc-9-2009-2015, 2015.

Maeno, N. and Ebinuma, T.: Pressure sintering of ice and its implication to the densification of snow at polar glaciers and ice sheets, J. Phys. Chem., 87, 4103–4110, https://doi.org/10.1021/j100244a023, 1983.

Medley, B., Neumann, T. A., Zwally, H. J., Smith, B. E., and Stevens, C. M.: Simulations of firn processes over the Greenland and Antarctic ice sheets: 1980–2021, The Cryosphere, 16, 3971–4011, https://doi.org/10.5194/tc-16-3971-2022, 2022.

Morris, E. M. and Wingham, D. J.: Densification of polar snow: Measurements, modeling, and implications for altimetry, J. Geophys. Res.-Earth Surf., 119, 349–365, https://doi.org/10.1002/2013JF002898, 2014.

Morris, E. M. and Wingham, D. J. : Uncertainty in mass balance trends derived from altimetry; a case study along the EGIG line, Central Greenland. Journal of Glaciology, 61(226), 345–356, https://doi.org/10.3189/2015JoG14J123, 2015.

Noël, B., van de Berg, W. J., Machguth, H., Lhermitte, S., Howat, I., Fettweis, X., and van den Broeke, M. R.: A daily, 1 km resolution data set of downscaled Greenland ice sheet surface mass balance (1958–2015), The Cryosphere, 10, 2361–2377, https://doi.org/10.5194/tc-10-2361-2016, 2016.

Simonsen, S. B., Stenseng, L., Adalgeirsdóttir, G., Fausto, R. S., Hvidberg, C. S., and Lucas-Picher, P.: Assessing a multilayered dynamic firn-compaction model for Greenland with ASIRAS radar measurements, J. Glaciol., 59, 545–558, https://doi.org/10.3189/2013JoG12J158, 2013.

Sørensen, L. S., Simonsen, S. B., Nielsen, K., Lucas-Picher, P., Spada, G., Adalgeirsdottir, G., Forsberg, R., and Hvidberg, C. S.: Mass balance of the Greenland ice sheet (2003–2008) from ICESat data – the impact of interpolation, sampling and firn density, The Cryosphere, 5, 173–186, https://doi.org/10.5194/tc-5-173-2011, 2011.

Steger, C. R., Reijmer, C. H., van den Broeke, M. R., Wever, N., Forster, R. R., Koenig, L. S., Kuipers Munneke, P., Lehning, M., Lhermitte, S., Ligtenberg, S. R. M., Miège, C., and Noël, B. P. Y.: Firn Meltwater Retention on the Greenland Ice Sheet: A Model Comparison, Front. Earth Sci., 5, 3, https://doi.org/10.3389/feart.2017.00003, 2017

Verjans, V., Leeson, A. A., Stevens, C. M., MacFerrin, M., Noël, B., and van den Broeke, M. R.: Development of physically based liquid water schemes for Greenland firn-densification models, The Cryosphere, 13, 1819–1842, https://doi.org/10.5194/tc-13-1819-2019, 2019.

Verjans, V., Leeson, A. A., Nemeth, C., Stevens, C. M., Kuipers Munneke, P., Noël, B., and van Wessem, J. M.: Bayesian calibration of firn densification models, The Cryosphere, 14, 3017–3032, https://doi.org/10.5194/tc-14-3017-2020, 2020.

Verjans, V., Leeson, A., McMillan, M., Stevens, C., van Wessem, J. M., van de Berg, W. J., van den Broeke, M., Kittel, C., Amory, C., Fettweis, X., Hansen, N., Boberg, F., and Mottram, R.: Uncertainty in East Antarctic firn thickness constrained using a model ensemble approach, Geophys. Res. Lett., 48, e2020GL092060, https://doi.org/10.1029/2020GL092060, 2021.

---

## Author Comment (AC1)

**Summary of study and overall assessment**

In this study, the firn properties of Greenland are simulated for the period 1980 – 2020 with two different models – the semi-empirical Community Firn Model (CFM) and the physically-based SNOWPACK (SP) model. Both models are driven with atmospheric forcing from MERRA-2 reanalysis and applied on the same spatial grid (~0.5°) as the reanalysis data. To equilibrate the firn layer, the models were first spun-up with forcing data from the so-called reference climate interval (RCI), which ranges from 1980 to 1995. Subsequently, the actual simulations with CFM and SP were performed and evaluated with 767 firn cores from the SUMup project. After demonstrating the good overall performance of both models, results are analysed and intercompared with a focus on firn air content (FAC) and its temporal evolution (interval means, inter-/intraannual changes) and spatial distribution on basin scales.

Firn models are important tools, because they allow a spatially comprehensive assessment of ice sheet's average firn porosity. This quantity is essential for estimating the potential of the ice sheet to retain meltwater in the firn layer ( buffer effect) and thus slowing down global sea level rise. It is therefore crucial to have a good understanding of how well firn models of different complexity simulate firn related processes. This manuscript adds interesting results and findings to previous work. The study is generally well written and structured and the figures are of excellent quality. Find below some suggestions to improve the manuscript – most comments are of minor nature and concern details.

We thank the reviewer for taking time to read and comment on this manuscript. We are especially grateful for the thoughtful and inspired questions about this work. The updated manuscript has been greatly improved following the reviewer's comments. We would also like to note that we have changed "CFM" to "CFM-GSFC" per another reviewer's suggestion, so the responses below contain the latter use. The reviewer's comments are in black text, the authors' responses are in blue text, original manuscript text that has been removed or modified is *"blue and in quotes and italics"*, and new manuscript text is ***blue and in italics and bold***.

**General comments**

**Conclusion section**

In my opinion, the conclusion section needs some improvement. The structure seems currently a bit chaotic – e.g. the part with the outlook ("This will in turn allow us to better predict the firn's response to future warming.") should rather be at the end of the section. I suggest to rearrange this section in a more logical way.

We thank the reviewer for this suggestion, and we have restructured and refocused the conclusion section. Below we address some of the specific points noted and show how we have updated the manuscript.

Furthermore, the following points could be included/extended:
- Embed findings in a larger picture (and discuss further implications). For instance, I guess the computational cost of running the physically-based SNOWPACK model is substantially higher (could you state how much approximately?). Does the higher complexity (e.g. explicit consideration of effects like wind compaction under drifting/blowing snow that influence new-snow density) "pay off" (i.e. add some distinctive benefits)?

Thank you for this suggestion. The computational costs are difficult to evaluate due to several factors including the fact that the models are being run on different systems, changing parameters in either model can drastically affect runtime, and different climate conditions (e.g., high melt) can affect model runtimes differently. As such, neither model necessarily stands out as being more computationally efficient or inexpensive. The SNOWPACK and CFM-GSFC simulations performed for the current work took about the same amount of time to perform. However, as to your second point, we have summarized some of the potential benefits of each model in the conclusion section. Our results do not support the idea that one model is better than the other, but rather quantify their differences in several contexts (e.g., stable climate, changing climate, melt, etc.). To the conclusion, we have added:

***In the present work, neither model clearly outperforms the other within the scope of the evaluation. Observational limitations prevent the recommendation of one model over the other,***

*but here we report potential benefits and drawbacks of each model. The physics-based design of SNOWPACK means that it is not tuned to observations and consequently not biased toward available observational data. This may mean that it can more realistically simulate firn properties under future climate conditions whose effects are not capture in existing firn observations. However, firn physics are not fully understood and knowledge gaps limit the accuracy of the model. The CFM's modular design allows for the user to easily choose from several densification schemes. Semi-empirical densification schemes such as the one used in the present work are tuned to observations, which means that realistic densification relationships are built into the model and there is less need to rely on poorly understood physics. To recommend one model over another, additional evaluation data and metrics would be necessary.*

- State recommendations for future (similar) studies and extend outlook. For instance, which are the most crucial processes in firn model that should be better represented in future models (I have in mind processes like vertical (or even lateral) water flow, reduced permeability of ice layers/slabs, ponding water conditions in firn aquifers, etc.)?

These are critically important ideas and questions for the firn modeling and observation communities. Capturing these complex meltwater processes in observations is difficult, and the physics are also not well-constrained. This makes it difficult to determine which process is most crucial to model accuracy. It also makes it difficult to accurately represent these processes in models if the real-world examples are limited. As such, we have added the following to the conclusion section in order to share what may be needed in order to more accurately capture certain processes in firn models:

*Moreover, firn models in general are limited by knowledge gaps in firn hydrological processes such as vertical meltwater percolation, lateral flow, and conditions for firn aquifer and ice slab formation. Additional research focusing on obtaining detailed observations of these processes would provide opportunities for important developments in firn modeling.*

**Point-comments**
**Content-related (text)**
**Line 10:** For which time are these statements valid? 1980, 2020 or averaged over the 40 years?
Thank you for noting this. We have added *1980–1995 average* to clarify.

**L84:** I would call this section "**Methods and data**" (because you also present the SUMup observations)
Done.

**L88:** Do you consider both snow- and rainfall data from MERRA-2? Or do you derive precipitation fractions (solid/liquid) with an air temperature threshold?
From MERRA-2, we obtain the precipitation as three variables from the integrated diagnostics of water and energy dataset (GMAO, 2015). These are the convective rainfall, large-scale rainfall, and snowfall. To make this clearer to the reader, we have added *(rainfall and snowfall)* following *"precipitation"* in the list of MERRA-2 variables used.

**L95:** I would explicitly state that MERRA-2 was also considered in Zhang et al. (2021) – this is not obvious from the current statement. Maybe you could also briefly summarise how the model performs with respect to Automatic Weather Stations (AWSs) data.
Thank you for this suggestion. Per another reviewer's suggestion we have removed this sentence and replaced it with a broader and less biased one: *A different reanalysis product or regional climate model could also be used here, though the choice of forcing dataset will not affect the firn model intercomparison since we provide the two firn models with identical input.*

**L99:** I have a general question (just out of curiosity – no changes regarding this question are required for the current manuscript): SNOWPACK and CFM inherit MERRA-2's spatial grid. However, one could also apply a different (unstructured) grid, which e.g. has a higher spacing close to the ice sheet's margins. With this, one could better capture areas with strong climate gradients and the complex boundary of the ice sheet (which might also reduce the disagreement in total glaciated area). However,

such a solution might anyway only be relevant if a generally higher grid spacing than 0.5° is used (also in terms of atmospheric forcing data). What's your option on this idea for future firn model applications to the GrIS?

We thank the reviewer for this insightful question that could inspire a future research project. A finer resolution grid would indeed be a way to capture the fine spatial-scale impacts of climate and topography on firn structure, especially in these marginal areas. The limitation here would be the accuracy of the climate model. The marginal areas are notoriously difficult to represent as they are characterized by steep topography, orographic precipitation, strong temperature gradients, complex winds, and flowing outlet glaciers. As such, the choice of an adequately accurate downscaled or finer scale climate model would be difficult. Still, this is an interesting question that perhaps could be investigated by using a suite of regional climate models and focusing the modeling/analysis on the margins.

**Section 2.2:** Could you specify which scheme for vertical water percolation is applied in SNOWPACK?

Yes; thank you for pointing out that we have forgotten to mention SNOWPACK's percolation scheme. We use the bucket scheme for both models, and we have updated the text to include: ***We apply a bucket scheme to represent vertical water percolation in SNOWPACK.***

**L125:** It might be useful to refer to Fig. A2 here (time series in the grey-shaded areas show no (strong) temporal trends, which supports the definition of the RCI period)

Thank you for this suggestion. We have added ***(Fig. A2)*** to this line.

**L135:** Why do you perform the vertical interpolation only for CFM output (and not for SNOWPACK – which also has a fine grid spacing)?

Here, the different choices in whether or not to vertically interpolate comes down to differences in the way we chose to reduce computational costs. Both models actually have a layer-merging scheme that reduces computational costs. In the CFM-GSFC, we reduced file sizes by interpolating onto a regular grid, and in SNOWPACK, we only save output every 7 days. These were choices made early in the project, and we may reconsider these if we were to run the models again.

**L150:** Why do you apply different spin-up conditions for SNOWPACK and CFM? Is it due to computational constraints (i.e. that SNOWPACK is more expensive to run)?

To the best of our ability, we have made the spin-ups as close to identical as possible. Any differences in the spin-up regimes are just due to the way the models were coded and designed. We have removed *"For example, if the firn needs 1000 years to spin up, the RCI would repeat 63 times. Once the spin up is completed, the main model run (1980–2020) commences."* to reduce any confusion.

**L169:** Here, you neglect any liquid water in the firn – right? Compare e.g. to Eq. (6) and (7) in Kuipers Munneke et al. (2015).

Yes, that is correct. We calculate FAC as in Eq. (7) in Kuipers Munneke et al. (2015).

**L174:** Why do you use 100 m as a lower limit (and not e.g. 150 m – the spin-up depth of SNOWPACK)?

We choose 100 m as a cutoff for two reasons. (1) Most of the simulations reach solid ice by 100 m, meaning that there is negligible FAC below 100 m for many cases. (2) We chose a depth that every simulation in both SNOWPACK and the CFM would reach. In SNOWPACK, the spin-up will be complete when there is either 150 m of firn or when the bottom 3 m is solid ice (see Section 2.2). This means that a simulation may not reach 150 m, but we calculated the minimum thicknesses from all simulations (both models) and found that they all reached at least 100 m depth.

**L181:** I would briefly explain what the NSE range (<0, 1, etc.) means for the model (because most readers are probably unfamiliar with this metric)

We thank the reviewer for this suggestion. We have added ***A value of 1 indicates perfect model performance, whereas a value of 0 indicates that the model's predictive ability is the same as using the observations' means.***

**L269:** "no change" might be a bit too restrictive. Maybe better "only negligible changes"
Thank you for this suggestion. Per another reviewer's suggestion and this reviewer's comment on Line 323, we have chosen to replace Table 2 (and the explanation of its results) with trends during the two compared periods (see below). Thus, we have removed this line.

*Table 2. Modeled spatially-integrated firn air content (FAC) trends and standard errors of the trends for each of the six basins (Fig. 6) for the 1980–1995 reference climate interval (RCI) and the 2005–2020 period. The last row shows the trends for the full GrIS.*

| Basin | 1980–1995 | | 2005–2020 | |
|-------|-----------|---|-----------|---|
| | CFM-GSFC trend (km$^3$ y$^{-1}$) | SNOWPACK trend (km$^3$ y$^{-1}$) | CFM-GSFC trend (km$^3$ y$^{-1}$) | SNOWPACK trend (km$^3$ y$^{-1}$) |
| NW | -0.1±0.1 | -1.1±0.1 | -2.6±0.1 | -12.3±0.2 |
| CW | +0.2±0.0 | +1.5±0.1 | -2.1±0.0 | -10.3±0.1 |
| SW | -1.0±0.1 | -1.2±0.2 | -6.9±0.1 | -16.7±0.2 |
| NO | +0.6±0.0 | +0.4±0.1 | -5.7±0.0 | -11.4±0.1 |
| NE | -1.3±0.1 | -0.8±0.3 | +1.9±0.1 | -2.5±0.2 |
| SE | +3.1±0.1 | +3.1±0.3 | +2.4±0.2 | -4.5±0.4 |
| GrIS | +1.2±1.0 | +1.7±0.9 | -17.4±1.2 | -66.6±1.2 |

**L289:** I would shift this first paragraph (maybe to the end of this section?). For me, these first lines suggest that it is not interesting to look at trends because there is no significant change in FAC between the two periods. However, looking e.g. at Fig. 8, there seems to be a clear trend during the latter period which is definitely worthwhile to discuss. Anyway, I have to admit that I'm not an expert on statistical methods, so there might be a reason why you start with comparing the two periods statistically…
We appreciate this comment and suggestion from the reviewer. We chose to report these results here for the sake of thoroughness and to create a baseline for comparing with the later 2005-2020 period, which as noted by the reviewer, is much more exciting. To make this clearer to the reader, we have changed the first line of this paragraph to: *To examine how the models represent the seasonal cycle in spatially-integrated FAC during the RCI,...*

**L323:** I think it would be more robust to look at linear trends here. Computing the difference between two (somehow arbitrary selected years) is prone to noise introduced by interannual variability…
This is an excellent point and we see the limitations of comparing two single years, especially in relation to the interannual variability. As such, we have shifted the focus from comparing two years to comparing the trends between the two periods (1980–1995 and 2005–2020). We have remade Table 2 (shown above in an earlier response) and replaced this paragraph with:

*The marginal areas of the GrIS experience the greatest amount of FAC depletion between 2005 and 2020 (Fig. 10). Both models simulate the same spatial patterns in loss, but the trends vary by basin (Table 2). SNOWPACK simulates a negative trend in spatially-integrated FAC in all basins during this time, with the strongest trend of -16.7±0.2 km$^3$y$^{-1}$ in the southwest. The negative trend is weakest in the northeast (-2.5±0.2 km$^3$y$^{-1}$) and southeast (-4.5±0.4 km$^3$y$^{-1}$), which are also the only two basins where the CFM-GSFC simulates positive trends (1.9±0.1 and 2.4±0.2 km$^3$y$^{-1}$, respectively). The CFM-GSFC also simulates the strongest negative trend in the southwest where the spatially-integrated FAC change is -6.9±0.1 km$^3$y$^{-1}$ (Table 2).*

**L445:** I'm not able to follow this sentence. Do you mean "intensified firn densification"? And why does that increase the firn's cold content?

Thank you for noting this point of confusion. We have rewritten this sentence for clarity, and also removed the language about cold content since it was tangential to the main point. It now reads: *Pore space depletion can also be caused by firn densification, which in turn modifies the meltwater refreezing and retention capacities of the firn in a complex manner (Vandecrux et al., 2020).*

**Typos, phrasing and stylistic comments**
**Line 6:** …Community Firn Model (CFM)**,** to quantify…
Done.

**L15**: This sentence reads odd somehow. It might be better to add the negative rates to the previous sentence and then state: "The reduction in spatially-integrated FAC in SNOWPACK and CFM demonstrate how model differences propagate throughout the FAC record."
Thank you for this suggestion. We have changed these lines to: *During this period, the spatially-integrated FAC across the entire GrIS decreases by 3.2 % (-66.6 km$^3$ y$^{-1}$) in SNOWPACK and 1.5 % (-17.4 km$^3$ y$^{-1}$) in the CFM-GSFC. These differing magnitudes demonstrate how model differences propagate throughout the FAC record.*

**L117:** "scheme use to" "scheme use**d** to"
Done.

**L195:** "form**ed**"
Done.

**L197:** "in the surface" "close to the surface"?
Changed to *near the surface*. Thank you.

**L364:** I would change this to something like: "The five locations shown in Figure 4 lie all within the same MERRA-2 grid cell and thus share the same atmospheric forcing data for the models."
Done. Thank you for this sentence restructuring; it reads much better now.

**L366:** change "MERRA-2 grid point" to "MERRA-2 grid cell" (also later in the text)
Done. We have made 7 replacements.

**L374:** "in simulating observations" "in reproducing observations"
Done.

**L375:** I would rephrase this sentence.
This has been changed to: *Compared to the dry and flat ice-sheet interior, areas with firn aquifers or steep topography are likely to have higher model uncertainty.*

**L443:** I would rephrase this sentence.
This has been changed to: *The largest percent change is in the southwest, which has the warmest temperatures and highest melt compared to other basins during this period (Table A2).*

**L461:** I would rephrase this to something like: "For both models, the summer air temperature seems to be a good proxy for the abrupt drop in FAC, which happens at temperatures between approximately -4 to 0° Celsius."
We have changed this sentence to: *Modeled FAC abruptly decreases at a summer air temperature threshold of 4°C, and the models remain in agreement in warmer conditions.*

**Figures and Tables**
**Figure 1:** Adding degree symbols and N/E to the latitude/longitude coordinates would help the reader.
Thank you for this suggestion. We have added degree symbols to the coordinates, but there is not much room in the legend for N/E, so we have kept with the positive/negative convention. Updated Figure 1 is below:

[Figure]

**Figure 2:** I would state relative biases in percentages like specified in Eq. (4)
Thank you for this suggestion. We have change the fractional biases to percentages to better reflect Eq. (4). The color scale has also been changed per another reviewer's comment. Please see the updated Figure 2 below.

[Figure]

**Figure 4:** I would change "MERRA-2 domain" to "MERRA-2 grid cell" and in the caption: "MERRA-2 grid point" "MERRA-2 grid cell"
Done. The updated Figure 4 is shown below.

[Figure]

**Figure 6:** caption what caused the missing data?
We have added the following explanation in Section 3.2: ***A few areas of missing data exist and are due to one or both of the firn models encountering an error in the simulation (Fig. 6). For example, if a grid cell is located in the ablation zone and does not receive enough accumulation to model a snow layer, SNOWPACK will stop the simulation and report an error.***

**Table 3:** How did you distinguish between detectable and undetectable signals?
We have added the following text to Section 3.2 to clarify why this is: ***(i.e., some basins contain too much intra-annual variability for the sine fitting function to detect a seasonal cycle)***

**Fig. A1:** caption: this means you only consider SUMup observations for this analysis in which the upmost density measurement covers the topmost 0.1 m or less – right? Furthermore, I would change the following sentence slightly: "The CFM uses a prescribed surface density of 350 kg m⁻³ **(green vertical line)**, which falls near many of the observed surface densities."
To the first point, yes, for this figure we only consider SUMup cores that have a measurement within 0.1 m of the surface. This is not the case for the entire paper's analysis though, so we have added to the caption, ***in this figure*** to clarify. To the second point, thank you. We have restructured that sentence. The full, updated caption now reads:

***Figure A1. (a) Observed surface density ($\rho_0$) from SUMup versus SNOWPACK. Since some observations begin farther below the surface, in this figure, observed $\rho_0$ is defined as the uppermost density measurement that is within 0.1 m from the surface. The SNOWPACK $\rho_0$ is calculated over the same vertical segment as the SUMup observation. The CFM-GSFC uses a prescribed surface density of 350 kg m⁻³ (green vertical line), which falls near many of the observed surface densities. (b) Histogram of observed surface density with the mean represented by the black line. Also plotted is the CFM-GSFC surface density of 350 kg m⁻³.***

**Fig. A3:** caption: I'm not able to follow the anomaly calculation. Wouldn't subtracting each year's mean from the record lead to discontinuities in the time series? And wouldn't it be easier to simply detrend the time series? Because this part is methodological a bit more complex (see also my comment to Table 3), it might even be worth to move this part to a separate section in *2. Methods and data*.

We thank the reviewer for pointing out the confusion in this caption and we have updated the text in Section 3.2 where we first mention the seasonal signal. The reason we choose to subtract each year's annual mean is to account for the differences in magnitude from year to year. Thus, we end up with the seasonal signal (i.e., anomalies that are independent of the annual mean). The means from one year to the next are not so drastically different that they lead to massive discontinuities.

In Section 3.2, we have rewritten the beginning of the seasonal signal paragraph as follows: ***To examine how the models represent the seasonal cycle in spatially-integrated FAC during the RCI, we subtract the annual means from each year to isolate the seasonal signal. We then fit a sine wave to the data (Fig. A3) and quantify a seasonal signal from the amplitude following methods from Ligtenberg et al. (2012).***

**New references**

Kuipers Munneke, P., Ligtenberg, S. R. M., Noël, B. P. Y., Howat, I. M., Box, J. E., Mosley-Thompson, E., McConnell, J. R., Steffen, K., Harper, J. T., Das, S. B., and van den Broeke, M. R (2015).: Elevation change of the Greenland Ice Sheet due to surface mass balance and firn processes, 1960– 2014, The Cryosphere, 9, 2009–2025, https://doi.org/10.5194/tc-9-2009-2015

References used in response

Global Modeling and Assimilation Office (GMAO): MERRA-2 tavg1_2d_int_Nx: 2d,1-Hourly,Time-Averaged,Single-Level,Assimilation,Vertically Integrated Diagnostics V5.12.4, Greenbelt, MD, USA, Goddard Earth Sciences Data and Information Services Center (GES DISC), https://doi.org/10.5067/Q5GVUVUIVGO7, type: dataset, 2015

---

## Author Comment (AC2)

Review of Thompson-Munson et al.: *Observed and modeled Greenland firn properties (1980–2020)* by Vincent Verjans.

This study applies two state-of-the-art firn models at the scale of the Greenland Ice Sheet (GrIS). The Community Firn Model (CFM) is used with the semi-empirical firn densification formulation NASA GSFC- FDMv1 of Medley et al. (2022). SNOWPACK is a more physically-detailed snow compaction model. The goal of this study is to compare results from these two different approaches to firn modeling over the GrIS. The authors also perform a comparison of model output against in-situ firn core observations.

I believe that this study demonstrates a comprehensive modeling effort, and the results are a valuable large-scale comparison of firn model behavior. The modeling experiments are rigorous, well-explained, and are undoubtedly a great contribution to the firn modeling community. The use of a same atmospheric forcing allows to identify differences only related to firn model and to parameterization choices. The figures are of good quality. And the authors perform a thorough evaluation by using an extensive dataset of 766 firn cores. However, I believe that there is a problem in the methodology of the evaluation, as I explain in this review. I appreciate the modeling effort and the results of this study, but I have some reservations concerning the interpretation and the lack of in- depth investigation of the structural differences between the two models. In other words, the results are excellent, but what can we conclude from this study? What is the main message for the firn modeling community? I believe that with a little more analysis, this study can be much more than simply providing model output from two models at the GrIS scale. This work is in the scope of The Cryosphere, and I welcome its publication pending some revisions.

I have separated my review in Major comments that require a re-evaluation of some steps of the study, Minor comments that require more clarity in the manuscript and/or small changes, and Specific comments, which focus on specific aspects, and are mostly of technical nature. Despite my numerous comments, I strongly encourage the authors to re-submit the manuscript after the revisions have been made.

We thank the reviewer for their careful reading of the manuscript, insightful questions and comments, and expert advice regarding both the scientific methodology and the writing itself. The manuscript has been much improved as a direct result of this reviewer's comments. We have responded to the major, minor, and line-by-line comments below. The reviewer's comments are in black text, the authors' responses are in blue text, original manuscript text that has been removed or modified is *"blue and in quotes and italics"*, and new manuscript text is **blue and in italics and bold**.

Major comments
1) Problems of the evaluation
The use of 766 firn cores in the evaluation process is noteworthy. However, as pointed out by the authors (l.207- 208):
"*Since most observations are from shallow cores (median depth = 2.0 m; Fig. 2a) the observed FAC values are relatively low (median FAC = 1.3 m; Fig. 2b) and do not represent the FAC of the full firn column.*"
This skewed distribution of the observations make the evaluation very biased and difficult to interpret. In any modeled firn profile, FAC in the upper two meters is essentially dictated by the surface density ($\rho_0$). In the CFM, $\rho_0$ is fixed to 350 kg m-3. This is purely a choice of the authors, as any other constant value or parameterization of $\rho_0$ would be equally valid (e.g., Kuipers Munneke et al., 2015; Fausto et al., 2018; Medley et al., 2022). Thus, the statistics of the evaluation (NSE, relative bias) essentially reflect how well the choice of $\rho_0$ fixed to 350 kg m-3 fits the SUMup surface densities, rather than showing the performance of the CFM GSFC-FDMv1 densification scheme. Similarly for SNOWPACK, the evaluation reflects the performance of the surface density scheme compared to the SUMup surface densities, and not its performance in densification physics.
Another issue with using FAC as an evaluation metric is that the best predictor of FAC is the core depth. However, this does not bear any information about model performance. For example, a pair of modeled and observed FAC values over 2m depth will (almost) always be close to each other, and a pair of

modeled and observed FAC values over 20m depth will also (almost) always be relatively close to each other. The good correlation between modeled and observed FAC values is due to cores being compared over a same depth. This problem arises because the authors have decided to use all the SUMup cores in the evaluation, and not to restrict their analysis to cores with a minimum depth threshold.

In order to alleviate these two problems, I encourage the authors to make an evaluation by binning cores based on their depth. A separate evaluation for each depth bin should be performed. For example, all the cores can be separated in groups of depth<2m, 2m<depth<5m, 5m<depth<15m, depth>15m or something similar. The binning should be made appropriately in order to have sufficient cores in each bin, but also meaningful evaluation statistics at the same time.

Thank you for this comment and excellent suggestion to improve the evaluation method. We agree that the skewed distribution of the observations' depths makes the analysis less meaningful, and to address this, we have followed this suggestion and performed the analysis for different bins of core depths. We divided the 767 observations into four bins with core depths of (a) 0 to 1 m, (b) 1 to 2 m, (c) 2 to 10 m, and (d) >10 m. These depth thresholds were chosen to ensure that each bin had >100 observations (Figure R1).

[Figure]

**Figure R1.** SUMup observations partitioned into the four bins defined by the core depth.

We then performed the FAC model evaluation for each bin, which included performing a linear regression, calculating relevant statistics (e.g., NSE), and plotting FAC. We revised Figure 3 (see below) to include eight additional panels: observed vs. modeled FAC for each bin, for both SNOWPACK and the CFM-GSFC. We also updated the caption text to reflect these changes (see below). Finally, we made changes to the manuscript's text. In Section 3.1, we added:

*In these shallow cores where densification has little impact on FAC, the model performance is a reflection of the models' representations of the surface density. In SNOWPACK, the surface density is modeled from the atmospheric input, and in the CFM-GSFC the surface density is fixed at 350 kg m$^{-3}$. We compare observed and modeled FAC for all 767 points (Fig. 3a, b), but we also partition the dataset into bins based on core depth (Fig. 3c-j) to evaluate model performance in terms of both the surface density parameters (shallower cores) and the densification schemes (deeper cores). We use the following core depth thresholds for binning the data: 0 to 1 m (n = 253), 1 to 2 m (n = 112), 2 to 10 m (n = 242), and >10~m (n = 160).*

*When the evaluation is performed for the four bins of core depths, the models still agree with the observations but the performance differs in each bin (Fig. 3c-j). SNOWPACK performs best in the shallowest cores (NSE = 0.84, MAPE = 9 %) where the FAC is a reflection of the surface density scheme (Fig. 3c). As densification becomes more important with depth, the model performance decreases but still remains reasonable (Fig. 3d–f). For the CFM-GSFC, the FAC in the shallower bins (Fig. 3g, h) is impacted by the fixed surface density and vertical interpolation that together prevent the fine resolution necessary for comparisons with observations. At depth, the CFM generally performs well and has NSE and MAPE values comparable to SNOWPACK (Fig. 3i, j).*

[Figure]

**Figure 3. Observed versus modeled firn air content (FAC) for all core depths for (a) SNOWPACK and (b) the CFM-GSFC. The smaller panels show the same comparison but for the four bins of observations partitioned by core depth for (c-f) SNOWPACK and (g-j) the CFM-GSFC. The core depth bins are shown in bold above each panel. The number of points (n), the Nash-Sutcliffe Efficiency (NSE), and the mean absolute percentage error (MAPE) are reported for each model in the lower right. The gray dashed line is a 1:1 line. Points with biases greater than 5 m are circled in red and correspond to the density profiles shown in Figure 4.**

Furthermore, I would like the authors to highlight more clearly that the CFM results at low FAC values are mostly determined by the $\rho_0$ choice and not by the GSFC-FDMv1 densification scheme. In the current version of the manuscript, this is not clear for readers less familiar with firn modeling. Finally, the authors point out (l197-200): "*SNOWPACK simulates more variability between layers compared to the CFM. This partly results from the fixed surface density of 350 kg m$^{-3}$ set for the CFM, while the surface density in SNOWPACK varies based on atmospheric conditions, and partially because the CFM outputs are interpolated onto a grid.*" This is important and needs to be quantified. How much of this low-variability error is due to the $\rho_0$ assumption? And how much is due to the interpolation? I strongly recommend to run the CFM at some firn core locations with $\rho_0$ set to the SNOWPACK $\rho_0$ time series, and without the interpolation scheme. This would bring better insights into the impact of these aspects.

This is a very good point and we appreciate the reviewer for writing out such a careful and helpful response. Originally, we ran the CFM-GSFC with (1) $\rho_0$=350 kg m$^{-3}$ as reported in the manuscript, and (2) $\rho_0$=$\rho_{SNOWPACK}$. Essentially, we used the surface density from SNOWPACK as the initial density for the CFM-GSFC, as suggested here. We chose to use $\rho_0$=350 kg m$^{-3}$ in the manuscript because there was no significant improvement in the CFM-GSFC performance. We have shown the results of using $\rho_0$=$\rho_{SNOWPACK}$ in Figures R2 and R3 below, which are Figures 3 and 1 in the manuscript remade with

SNOWPACK's surface density as the CFM-GSFC's initial density. Very little change is apparent between Figure 3 and R2. In Figure R3, the red lines show the density profiles created with $\rho_0 = \rho_{SNOWPACK}$. At depth (Figure R3b), there is very little difference between the two CFM-GSFC simulations. We have not rerun the CFM-GSFC with a different interpolation scheme, but the new Figure 3 shows that the interpolation is likely to matter in shallow cores (<2 m). Therefore, for a study interested in very accurately modeling shallow firn, the interpolation scheme should be carefully considered. However, in this study, we refocus our model evaluation to the deeper cores where uncertainty in initial snow density and the interpolation is negligible.

[Figure]

**Figure R2.** Same as Figure 3 in the manuscript except CFM was run using the new-snow density from SNOWPACK rather than a fixed 350 kg m$^{-3}$.

[Figure]

**Figure R3.** Same as manuscript Figure 1 with the addition of the red line that shows the density profile from CFM-GSFC using the new-snow density from SNOWPACK rather than a fixed 350 kg m⁻³. In panel (b), the two CFM-GSFC runs are almost identical at depth and the red line lies on top of the green.

2) Interpretation of the results
This study is a model intercomparison. I believe that this warrants more discussion of why the models diverge, and what conditions make them more prone to diverge/agree.

Thank you for this suggestion to draw more connections between the forcing and the firn models. We agree that there are opportunities for more thoroughly analyzing model behavior that would elevate the value of this manuscript. Below we outline the ways we have expanded the analysis further, but we also note that the complexity of the models, the variety of forcing variables, and the processes that impact FAC make it difficult to describe model behavior simply and answer some of the questions the reviewer has asked here. Nonetheless, we have made efforts to expand on the analysis and explanations.

This firstly necessitates a better description of the model physics. The governing equations (densification, heat conduction, etc.) should be provided in the manuscript or in an Appendix.

We appreciate the reviewer's comment pertaining to the model physics description. However, the governing equations for both the CFM-GSFC and SNOWPACK are described in detail in their respective papers (Stevens et al., 2020; Medley et al., 2022; Bartelt and Lehning, 2002; Lehning et al., 2002a, b) so we have chosen not to report them here. The models are complex, and especially in the case of SNOWPACK, no single equation or even small subset of equations can be used to fully explain the sensitivities of the models to atmospheric forcing as described in the below comment. In the manuscript, we have provided the relevant model parameterizations (or references to papers with those parameterizations). We would like to note that we do see the value in describing the model physics and including the equations, but since this information already exists elsewhere, we have chosen to simply reference the previous studies instead.

Based on these equations and on their results, the authors should provide some explanation on the different sensitivities of CFM-GSFC-FDM and SNOWPACK to temperature, accumulation, melt, wind forcing, etc. For example, I found Figure 9 very interesting. But the analysis does not tell why GSFC-FDM and SNOWPACK agree well in the Northeast and Southeast, but show strong discrepancies in the Central West and Norhwest. As another example, from Figure 5, why does SNOWPACK simulate much larger FAC at low summer temperatures than CFM-GSFC-FDM?

We decided to report and discuss FAC as function of summer temperature and LTSR (Figure 5) because these two variables seemed to have the clearest impact on FAC. However, we have followed the reviewer's advice and added Figure A4 (below), which expands Figure 5 to the six forcing variables plus melt. The relationship between these variables and FAC is complex and more difficult to explain. This highlights the complexity of the models and demonstrates that no single atmospheric variable can predict FAC and therefore cannot fully explain the spatial patterns.

[Figure]

**Figure A4. Modeled firn air content (FAC) in SNOWPACK (blue) and the CFM (green) as a function of the forcing variables: (a) temperature, (b) precipitation, (c) relative humidity, (d) wind, (e) incoming shortwave radiation (ISWR), and (f) incoming longwave radiation (ILWR) all calculated for 1980 through 1995. Also shown is (g) the melt, which is calculated by SNOWPACK's surface energy balance model and used as a forcing in the CFM.**

In Section 4.2, we have referenced Figure A4 with the following additional text before introducing the LTSR: *The models simulate complex FAC responses to to the forcing variables (Fig. A4)...*

Throughout the manuscript, I have been somewhat frustrated by the dichotomy between impressive results but lack of in-depth explanations.

Thank you for pointing out this shortcoming in the manuscript. We acknowledge that there are opportunities to improve and expand upon the discussions of the results. As such, we have expanded the explanations of many of our results, and we have added to the discussion sections. Examples of this are demonstrated in responses to later comments below.

Finally, the authors have related the sensitivity of both models to climatic conditions (LTSR and summer temperature) in the steady-state climate configuration. It would be interesting to expand such an analysis to the transient climate configuration. This would involve quantifying the sensitivity of FAC loss/gain to changes in atmospheric forcing.

Although we agree that this would be a very interesting result, we feel that this analysis would fall outside of the goals of this particular study. On its own, quantifying the sensitivity of FAC changes to atmospheric forcing changes could actually be a standalone research project. There would be opportunities to use machine learning techniques (e.g., logistic regression) to determine which forcing variables have the strongest effect on FAC loss or gain. Moreover, there may also be lags in the response of FAC to forcing changes, which would require a careful and systematic approach to performing the regression. Finally, one would want to study the effect of changes in each forcing variable individually (e.g., temperature changes but no other variable does) to concretely quantify the sensitivity.

As a final note concerning this Major comment 2, I should emphasize that addressing the sort of questions that I raise is not an absolute necessity for publication. The study is already a thorough modeling effort, with a good quantitative evaluation of the results. I simply believe that a thorough analysis of model behavior would bring this study to the next level.

Thank you for both the suggestions in this comment and this final note. We appreciate the opportunity to further improve this study and make the results more meaningful, and we feel that the reviewer's comments have greatly helped.

Minor comments
1) References
I find that this study does not sufficiently recognize previous work from the firn science community. I provide here some examples, but I also encourage the authors to proceed to a more in-depth literature review, and to cite other previous relevant studies in their manuscript.
We thank the reviewer for both pointing this out and providing an excellent list of references to include. We have added all but one of the citations and we have explained below the reasoning for not citing one of the suggested papers.

- l42-43 "*Changes in the amount of air-filled pore space within the firn, known as the firn air content (FAC), have been investigated in both observations (e.g., Vandecrux et al., 2019) and models (e.g., Medley et al., 2022).*": please cite Benson (1962); Braithwaite et al. (1994); Sorensen et al. (2011); Kuipers Munneke et al. (2015); etc.
Done

- l48-49 "*Modeling firn has become important for estimating mass balance (MB) from satellite altimetry, since this method relies on firn models to interpret the causes of surface height changes (e.g., Li and Zwally, 2011).*": please cite Arthern and Wingham (1998); Morris and Wingham (2014).
Done

- l52-53 "*Additionally, understanding the limits and deficiencies in firn models is essential for quantifying uncertainties in altimetry-based MB estimates.*": please cite Morris and Wingham (2015); Verjans et al. (2021).
Done

- l57-59 "*These models use empirical relationships between densification, accumulation, and temperature, and they are often tuned to observations (e.g., Ligtenberg et al., 2011; Medley et al., 2022; Li and Zwally, 2011).*": please cite Herron and Langway (1980); Arthern et al. (2010); Simonsen et al. (2013); Verjans et al. (2020)
Done

- l68-69 "*Still, both semi-empirical as well as physics-based firn models have been successfully used in Greenland (e.g., Vandecrux et al., 2020b; Dunmire et al., 2020; Medley et al., 2022).*": please cite Sorensen et al. (2011); Kuipers Munneke et al. (2015).
Done

l71-72 "*At an ice-sheet scale, few comparisons of semi-empirical and physics-based models exist*": please recognize the work of Steger et al. (2017) here.
Done

-l356-357 "*Neither model captures the high densities resulting from the firn aquifer because the use of bucket scheme in the models prevents full saturation in the firn.*": when discussing this aspect, please note that firn aquifer formation has been modeled by Verjans et al., (2019) and that conditions for aquifer development have been investigated by Kuipers Munneke et al. (2014).
Thank you for pointing this out and including these references. We have added ***In this study,*** before this sentence to reflect that the models' performances is due to our choices rather than inherent issues in the models. We have also noted the work the reviewer has mentioned and added the following sentence: ***However, the conditions for firn aquifer development have been previously investigated (Kuipers Munneke et al., 2014) and their formation has been previously modeled in another study (Verjans et al., 2019).***

- l361-362 "*Our results agree with these findings that model differences are highest where liquid water is present, indicating that poor representation of meltwater percolation processes is still a substantial limiting factor in firn model performance.*": please cite Verjans et al. (2019).
Done

- l370-372 "*This demonstrates the limitations of a coarsely-gridded forcing, especially in steeply sloped areas where climate is likely to be highly variable within a single grid cell*.": please mention the downscaling work of Noël et al. (2016).
We appreciate the suggestion of including this excellent downscaled product from Noël et al. (2016). However, we feel that mentioning it in this context is not suitable because RACMO2.3 at 1 km is a daily product. The SNOWPACK model set-up requires a finer temporal resolution (e.g., hourly), which means that this particular forcing would not be sufficient.

- l419-420 "*This could be related to the fact that SNOWPACK was developed using data from seasonal alpine snow which may not be representative of the physical processes driving deep firn densifcation.*": please cite Maeno and Einuma (1983); Arnaud et al. (2000).
Done

2) The Reference Climate Interval (RCI)
The use of an RCI is necessary for the spin-up of firn models. However, this implies assumptions which must be properly understood, explicitly stated, and discussed. Here, the authors state (l126-127): "*We make the assumption that this period is representative of the longer-term Greenland climate.*" And further, they state (l225-227): "*The RCI used for model spin-up spans 1980 through 1995, and since we assume that this period represents a relatively steady-state, long-term Greenland climate (Fig. A2)*". I believe that the message conveyed to the readers about the RCI is misleading. The RCI is used to develop the initial model firn column, from which transient experiments over the period of interest (1980-2020) start. As such, ideally, the initialization should be computed with the true climate forcing of the decades and centuries preceding 1980. This is true regardless of whether the long-term conditions were in steady-state (i.e., without trends) or not. In other words, the "perfect" RCI should not represent steady-state conditions if the true conditions were not in steady-state prior to 1980.
However, we have only incomplete knowledge of the true climate, especially prior to 1980. In light of this incomplete knowledge, using steady-state conditions over the RCI is a reasonable simplification, but not a necessary condition for a valid firn model initialization procedure. In the manuscript, it should be clearer that steady-state conditions over the RCI are used in order to isolate effects of climatic deviations from the RCI on firn column changes. But such steady-state conditions are not representative of true conditions, and the true changes in firn thickness are influenced by the unsteady nature of past climate conditions. I would like the authors to mention these points in the Discussion, as well as other studies that have shown that the pre-1980 climate of GrIS was not in steady-state (e.g., Hanna et al., 2011).

We thank the reviewer for this comment about the RCI and we acknowledge the shortcomings of this choice of initialization. Since the design of the study necessitates an RCI, we are required to make

certain assumptions about it. However, we want to emphasize that the focus of our paper is the intercompare model outputs, which means that the assumptions made apply to both firn models and differences in the model output are therefore not due to the choice of forcing data, RCI, and spin-up procedure. We also appreciate the points noted in the reviewer's comment and agree that there are more opportunities to expand on the RCI and its assumptions, and be clearer that the steady-state conditions are not representative of true conditions that have occurred in the past. With this in mind, we have created a new subsection in the Discussion (Section 4.4 Study limitations), to which we have added the following:

*The use of identical forcing data (MERRA-2) allows for direct comparison between the SNOWPACK and CFM-GSFC models in this work. We also use the same approaches to the spin-up, which uses a RCI of 1 January 1980 through 31 December 1995. Although we assume the steady-state conditions of the RCI represent the Greenland climate preceding 1980, we acknowledge that they are not representative of true conditions. For example, in the ~100 years before the RCI, significant trends in climate over the GrIS have been found (Hanna et al., 2011). Our steady-state assumption does not allow for such trends to appear in the spin-up, but the lack of pre-1980 data necessitates such assumptions. Since the focus of this work is to compare firn model results that are independent of the choice of forcing, the RCI assumptions do not impact the intercomparison. When comparing the firn models to the observations, the steady-state assumption may have an impact at depth. Deeper firn in the models is simulated from the repeated 1980–1995 climate, but real firn in the observations is older and may have formed during times when trends in the climate appeared (e.g., in the pre-1980 20th century; Hanna et al., 2011).*

3) Evaluation of sensitivity to climatic changes
The evaluation of model performances is performed with firn core data. Because firn cores only provide a snapshot of firn density in time (as pointed out by the authors), they cannot serve to evaluate the sensitivity of firn models to changes in climatic forcing. For example, good performance of a model when evaluated against firn cores does not imply that the model would accurately capture FAC changes under a +1°C change in mean surface temperature. This is particularly important to keep in mind when firn models are used to compute FAC change estimates in time, as done in this study. Evaluation with firn core data is legitimate given the scarcity of transient in-situ firn data, however I think that this limitation of the evaluation process deserves a paragraph in the Discussion section.

We appreciate this comment and acknowledge the limitations of only using firn core data, which are very limited in both space and time. We have added the following to the Discussion's new subsection (Section 4.4 Study limitations):

*The firn observations themselves are valuable snapshots of firn properties for a specific time and place, but their lack of temporal and spatial continuity limits the extent of this study's evaluation. Density and firn air content from the SUMup observational dataset do not provide sufficient information about how firn evolves through time. Moreover, the timing of when the observations were collected is not uniformly distributed throughout the year, which means there is less information on, for example, winter firn properties versus summer firn properties. However, a key feature of the SNOWPACK and the CFM-GSFC firn models is their ability to simulate the evolution of firn properties on fine temporal scales. The SUMup density measurements cannot be used to evaluate how well the models are able to simulate changing firn properties on fine scales, or how sensitive the models are to changes in the climate forcing.*

4) Clarification about the "CFM"
The authors repeatedly use the name "CFM" for one of the two firn models used. However, as far as I am aware of, the CFM allows to choose among various different firn densification formulations. Thus, I would find it more appropriate to call the model CFM- GSFC-FDM (or something similar). This is important because much of the FAC patterns are due to the use of the GSFC-FDM densification scheme, and not to the CFM itself, which is simply a numerical tool. Furthermore, the authors state that (l144-145) "*The densification rate is determined with the NASA GSFC-FDMv1 firn densification equation*

*(Medley et al., 2022)*". If this is the case, I suggest that they specify GSFC-FDMv1.2.1 to avoid any possible confusion with other versions of GSFC-FDMv1. Also, the CFM allows for a range of thermal conductivity parameterizations, the choice of which likely influences the results. I ask the authors to specify the thermal conductivity parameterization used. Finally, it should be clearer in the manuscript that the CFM itself is a numerical tool, and that the CFM output thus largely depends on the parameterization of the CFM (densification scheme, $\rho_0$, thermal conductivity), and not on the CFM itself. This is important for readers less familiar with the CFM.

Thank you for this comment and for pointing out the need to be more specific with the naming convention. We have chosen to use "CFM-GSFC" to highlight the fact that the CFM is a model framework that allows the user to choose parameterizations. We have replaced all instances of *CFM* with **CFM-GSFC** in both the text, tables, and figures. To clarify how the CFM-GSFC is designed for users, we have modified its description in Section 2.3 to now read:

***The Community Firn Model (CFM, Stevens et al., 2020) is an open-source model framework that simulates physical processes in firn. Its modularity allows users to choose which processes to simulate and which parameterizations to use (e.g., thermal conductivity, densification rate) in a given model run. As the CFM is a numerical tool, it is important to specific how the CFM has been configured for a particular run. The pertinent parameterizations used for the model runs in this paper can be found in Medley et al. (2022). For simplicity, in this paper we refer to our particular CFM configuration as "CFM-GSFC" to highlight that we are using the CFM with the semi-empirical GSFC-FDMv1.2.1 firn densification equation.***

We have also adjusted some of the language surrounding "CFM." In the abstract, we changed *the semi-empirical Community Firn Model (CFM)* to **the Community Firn Model (CFM) configured with a semi-empirical densification equation**. We have also removed any instances of "semi-empirical" when referring to just the CFM rather than the densification equation itself.

Specific comments

Title This study does not bring any new observations about Greenland firn properties. For this reason, I find the use of the word "*Observed*" in the title inadequate.
This is a very good point and we thank the reviewer for this comment. We have change the title of the study to **An evaluation of a physics-based and a semi-empirical firn model across the Greenland Ice Sheet (1980-2020)** to better reflect the focus of the paper.

l8
Specify: "isolate firn model differences".
Done

l9
"*Both models perform well*": this needs to be quantified.
We have added **(mean annual percentage errors of 14 % in SNOWPACK and 16 % in the CFM-GSFC)** after this phrase.

l10
"*is hindered by meltwater percolation*": this not really evaluated in this study. The authors only analyze in details the performance at firn aquifer sites, but do not compare performance in dry areas versus percolation areas more generally.
We thank the reviewer for this comment. We have removed this statement to simple read **…though their performance is hindered by the spatial resolution of the atmospheric forcing.**

l10
Change "*the full ice-sheet*" to "ice-sheet-wide".
Done.

l11
Please move "*(i.e., air volume in the firn)*" to the line where FAC is used for the first time.
We have decided to leave this as is since spatially-integrated FAC is a volume whereas FAC itself is reported in meters. To avoid unit confusion, we have not referred to FAC as a volume.

l13
Change "*the models' treatment of atmospheric input*" to "the sensitivity of the models to atmospheric forcing".
Done.

l15-16
Specify "spatially-integrated FAC decrease of".
Per another reviewer's suggestion, these lines have been written as follows: ***During this period, the spatially-integrated FAC across the entire GrIS decreases by 3.2 % (-66.6 $km^3$ $y^{-1}$) in SNOWPACK and 1.5 % (-17.4 $km^3$ $y^{-1}$) in the CFM-GSFC. These differing magnitudes demonstrate how model differences propagate throughout the FAC record.***

l22
Change "*in a thick*" to "by a thick".
Done.

l23
Change "*density of firn varies across the ice sheet*" to "density of firn varies in depth and across the ice sheet".
We have changed this to ***density of firn varies with depth and across the ice sheet***.

l24
Remove "*in time and*".
Done

l25
Specify: "can buffer the contribution of increased melt rates to sea-level rise".
Done

l26-27
I find this sentence confusing, as it is not specifically about firn. Please consider rephrasing.
Thank you for pointing this out. We have added the following sentence before this one in order to better introduce the purpose of providing these details, which is to highlight the complexity of meltwater flow and storage in the firn: ***The mechanisms for meltwater entering into and remaining stored in the firn are complex and varied.***

l31
To my knowledge, we are not sure whether ice slabs make deeper pore space completely inaccessible. I recommend using "potentially inaccessible".
This is an excellent point and we have made the suggested change.

l39
Change "*firn has lost its capacity to store meltwater*" to "the meltwater storage capacity of firn has abruptly decreased".
Done

l46
Change "*the memory effect of changes to the firn from previous years*" to "firn changes evolving on multi-year timescales".
Done

l52
Change "*measured in satellite altimetry*" to "measured from satellite altimetry".
Done

l59
"*Semi-empirical models are beneficial because they do not rely on the physics of firn densification*": this statement is too crude and needs more nuance (see Arthern et al., 2010 for example).
Thank you for noting this. We have modified the language to be more specific: **Semi-empirical models are beneficial because they can simulate more accurate depth-density profiles by calibration, which removes the uncertainties introduced by poorly understood densification processes in firn.**

l63-64
What is "*the constitutive relationship*"? As far as I know, even formulations linking stresses to firn strain rates rely on some form of parameterization, and there is no single universal constitutive relationship.
Thank you for pointing out the ambiguity in our use of "constitutive relationship." Here, we are are not necessarily pointing to a single, universal equation, but rather to the idea of a constitutive relation that describes the material property of snow and firn that links stress and strain rate (Cuffey and Patterson, 2010, p. 29). In physics-based models like SNOWPACK, the material properties of the snow layer are used to determine the strain rate under the applied stress (Bartelt and Lehning, 2002). To convey the fact that we are not referring to a single equation, we have changed the sentence to: **The alternatives to semi-empirical models are physics-based models that use the material properties of snow and firn to simulate densification based constitutive relations between stress and strain…**

l65
Change "*observations from tuning*" to "observations for tuning".
Done.

l67-68
"*since snow physics have been more-thoroughly studied*": this requires one more line of explanation, and also remove the hyphen.
We have changed this to: **The wealth of snow physics studies allowed for the development of more complex, physics-based, seasonal snow models like SNOWPACK.**

l68
"*have been successfully used*": what do the authors mean by "*successfully*"?
We have removed the word *"successfully"* to avoid any ambiguity or bias in this sentence.

l70-71
"*have seen significant development for polar regions in recent years*": can the authors please list some of these developments?
We have modified the beginning of this paragraph to elaborate on these developments: **The Community Firn Model (CFM) and SNOWPACK firn model have seen significant development in polar regions in recent years. In SNOWPACK, there have been modifications to the settling and microstructure schemes (Groot Zwaaftink et al., 2013; Steger et al., 2017), inclusion of drifting snow impacts on near surface density (Groot Zwaaftink et al., 2013; Keenan et al., 2021; Wever et al., 2022), and optimizations for computational efficiency by improving the layer merging scheme (Steger et al., 2017). The CFM has recently been used over both ice sheets using the GSFC-FDMv1.2.1 densification model (Medley et al., 2022).**

l78
Remove "*completely*".
Done.

l94-95
"*Regional climate models are not always widely available or regularly updated, and no single reanalysis clearly outperforms others over the GrIS (Zhang et al., 2021).*": either provide more details, or simply say

that the method could equally well be applied with a regional climate model or with another reanalysis product.

l95-97

I think that these two sentences should be rephrased as they do not read very well.

We have combined our responses to the two above comments: We have rewritten these sentences to now say: **A different reanalysis product or regional climate model could also be used here, though the choice of forcing dataset will not affect the firn model intercomparison since we provide the two firn models with identical input.**

l99

Replace "*full ice-sheet*" by "ice-sheet-wide".

Done

l99

Typo: "*gird*".

Thank you for seeing this. It has been changed to **grid**.

l105

What do the authors mean by "*successfully*"?

We have removed the word **successfully** since it is unnecessary here.

l109-110

"*the constitutive relationship*": see comment above.

We have change *"the"* to **a** to clarify that we are not referencing a specific equation.

l113-114

"*SNOWPACK uses the MeteoIO library (Bavay and Egger, 2014) for preparing the meteorological forcing data for the simulations.*": please explain.

We have added the following sentences to clarify how MeteoIO works: **The library reads the meteorological forcing from the MERRA-2 grids and provides data to SNOWPACK fro each grid cell, at each of the SNOWPACK time steps. SNOWPACK is run at smaller times steps than MERRA-2 data is available, and nearest neighbor interpolation (for wind speed) and linear interpolations (for all other variables) are used to provide meteorological forcing at higher frequency than provided by MERRA-2.**

l115-116

I don't think that storing output every 7 days conserves "*computational expenses*", but only reduces storage size. Please correct this statement, or provide explanations if I am wrong.

Storing the output at a lower resolution does in fact conserve computational expences and reduce model output sizes. Computational efficiency is impacted when high-frequency output is requested since write speeds to network drives are limited, and congestion occurs when multiple processes write to the drives. Moreover, the additional code and conversions required to write the text files adds to the expense. Therefore, we have chosen to leave this statement as is in the manuscript.

l117

"*impacted by the layer-merging scheme*": please explain.

We have added the following sentences to further explain the layer merging: **As described in more detail in Steger et al. (2017), depending on depth below the surface and similarity of snow properties in adjacent layers, the layers can be merged to reduce computational costs. If those merged layers come closer to the surface, they can be split again to maintain sufficient spatial resolution to capture the steep gradients near the surface.**

l117

Typo: "*use*" should be "used".

Changed. Thank you for finding this.

l119-120
"*We set the surface roughness to 0.002 m for solving the energy balance with the Michlmayr et al. (2008) stability correction when a stable boundary layer is diagnosed.*": please explain more.
We have expanded on this to better explain the stability correction and why it is needed. The text now reads: ***We set the surface roughness to 0.002 m for calculating turbulent energy fluxes when solving the energy balance. Here, we account for atmospheric stability using the Michlmayr et al. (2008) stability correction when a stable boundary layer is diagnosed. For unstable boundaries, which happen rather infrequently (Schlogl et al.), Eq. 8 in Stearns and Weidner 1993 is used.***

l121
Specify: "varies across the GrIS".
Done

l134-135
"*The CFM uses a layer-merging scheme at 5- and 10-m depth to reduce computational demands.*": please explain more.
We have added the following text to explain this further: ***The CFM-GSFC is coded so that each model time step adds a new layer. As such, daily time stepping generates many thin layers. To reduce computational demands, we use the CFM-GSFC's layer merging scheme. For this study's simulations, we merge 30 of the high-resolution (daily) layers at 5-m depth into mid-resolution (approximately monthly) layers. At 10-m depth, 12 layers are merged into coarser (approximately annual) layers.***

l149
"*was chosen to be near the depth at which the firn reaches the ice density*": please be more specific.
We have removed this sentence to avoid any confusion.

l151
Remove "*For example, if the firn needs 1000 years to spin up, the RCI would repeat 63 times.*".
Done.

l166
Remove "*(also known as "depth-integrated porosity (DIP)")*".
Done.

l168
Specify "where z = 0 m represents the surface, and is increasing downwards".
Done.

Section 3.1
Please re-order this section. The authors start with specific results at two individual cores and then provide the general results at the GrIS-scale. I recommend starting from the general results, and then focusing on specific results.
Thank you for your suggestion. We have decided to keep the original order of this section to first introduce the reader to the example density profiles, which are the basis for calculating the FAC that is later mentioned.

l193
"*collected in southwest Greenland*": please specify the core date also in the main text.
Done. We have added ***on 12 May 2013***

l197
Change "*in the surface*" to "near the surface".
Done.

l200-201

"*For polar regions in particular, temporal variations in wind and the presence of drifting snow translate into vertical density variations with increasing accumulation.*": I do not understand "*with increasing accumulation.*". Also, this statement requires some references.

Thank you for pointing this out. We have removed this sentence since it does not add value to this section and is not clear.

l211
Specify: "are overestimating FAC on average".
Done.

Section 3.2
In general, when comparing SNOWPACK and CFM results, I strongly recommend using the Root Mean Squared Deviation metric. This would provide more quantitative information in the analysis.

Thank you for this suggestions; we appreciate the added value of the RMSD statistic. We have added these values to Table 1, updated its caption, and reported the RMSD in the text. The updated table and its caption are below. We also report the changes and additions made to the manuscript.

***Table 1. Mean modeled firn air content (FAC) for the 1980–1995 reference climate interval (RCI) and for the 2005–2020 period, averaged across each of the six basins shown in Fig. 6. FAC is reported as mean±standard deviation, and the average percent difference and root-mean-square deviation (RMSD) between SNOWPACK and the CFM-GSFC are also shown. The last row shows the statistics for the full GrIS.***

| | 1980–1995 | | | | 2005–2020 | | | |
|---|---|---|---|---|---|---|---|---|
| | CFM-GSFC | SNOWPACK | Diff. | RMSD | CFM-GSFC | SNOWPACK | Diff. | RMSD |
| Basin | FAC (m) | FAC (m) | (%) | (m) | FAC (m) | FAC (m) | (%) | (m) |
| NW | 18.5±4.1 | 23.3±5.2 | 23 | 5.1 | 18.5±4.4 | 23.1±5.8 | 22 | 4.9 |
| CW | 19.5±3.7 | 24.8±4.6 | 24 | 5.4 | 19.8±4.0 | 24.8±5.2 | 22 | 5.2 |
| SW | 13.9±5.8 | 17.4±7.6 | 23 | 4.2 | 13.7±6.1 | 16.9±8.0 | 21 | 4.0 |
| NO | 15.5±4.6 | 18.1±5.1 | 15 | 3.0 | 15.2±4.8 | 17.5±5.5 | 14 | 2.7 |
| NE | 16.6±3.5 | 19.0±4.4 | 13 | 2.7 | 16.5±3.8 | 18.7±4.8 | 13 | 2.6 |
| SE | 18.8±4.8 | 23.7±7.7 | 23 | 5.9 | 18.8±5.1 | 23.4±7.9 | 22 | 5.6 |
| GrIS | 15.8±6.0 | 19.1±8.0 | 19 | 4.0 | 15.7±6.3 | 18.8±8.4 | 18 | 3.8 |

We have added the following to Section 3.2: ***The root-mean-squared deviation (RMSD), which represents the average difference in FAC between the two models, is 4.0 m.***

In Section 3.2, we changed *"The best model agreement is in the northeast and north basins where the difference is 13 % and 15 %, respectively (Table 1)"* to ***The best model agreement is in the northeast where the modeled FAC differs by 13 % and the RMSD is 2.7 m, and in the north where the difference is 15 % and the RMSD is 3.0 m (Table 1).***

l232-233
Remove this sentence as this is repetitive information.
Done.

Equation (6)
Does "*snow*" account for sublimation and blowing snow? And are these fluxes identical for SNOWPACK and CFM?

These fluxes are identical for SNOWPACK and the CFM-GSFC. Here, snow is the is solid precipitation.

Figure 1b
Notice that SNOWPACK does not underestimate density at high depth here. This is interesting to me, and warrants more analysis about the model physics (see Major comment 2).
The depth-density relationship is most definitely interesting here, especially since we have so few SUMup cores reaching depths >100 m. At depth, the driver of model differences is the different densification equations–whether they be empirical or physical. Please see the earlier response to Major Comment 2.

l242
Here and everywhere in the manuscript, change "*an FAC*" to "a FAC".
Done.

l244
Remove "*the*" in "*the FAC values*".
Done.

l246
Does "*response*" refer to FAC or to something else? Please avoid any confusion.
Thank you for pointing out the ambiguity. We have changed *"response"* to **FAC** to be more clear and specific.

l248
"*generally similar spatial patterns in FAC across the ice sheet*": please quantify.
We have removed this first sentence of the paragraph since it is too qualitative. We have instead replaced it with: **During the RCI, SNOWPACK and the CFM-GSFC produce similar spatial patterns in FAC, with higher FAC (>10 m) in the ice-sheet interior and lower FAC (<10 m) in the margins (Fig. 6). However, on average, FAC is greater in SNOWPACK than in the CFM-GSFC.**

Figure 2
Please use log color scale in Fig 2c and Fig 2d. With the current color scale, almost all the data points simply appear white.
Our original intention with this color scale was to show how well the models are performing in most cases, and to then highlight where the bias is very high. However, we acknowledge that this does not provide as much information pertaining to the spatial distribution of positive versus negative bias. As such, we have adjusted the color scale and thank the reviewer for this suggestion. Updated Figure 2c and d are below:

[Figure]

Figure 3

Because the equations of the linear regressions are not discussed in the main text, I recommend removing the black solid lines and the regression equations from Figure 3.
We have removed the regression lines and equations from Figure 3, and we have updated the caption to reflect those removals. Please see earlier comment for the updated figure and caption.

l253
Specify: "is on average greater than".
Done.

l253-254
"*difference between the models increases with depth*": please quantify.
The two sentences directly following this one provide quantification of the changes in percent difference with depth, and they report that it increases form 7 to 29 %.

l257
Refer to an appropriate Figure when introducing the basins.
Done.

l262
"fall in the middle": please rephrase.
We have rephrased **fall in the middle** with to simply **have**.

l271
Change "*between the two years*" to "over the RCI".
Done.

l271-272
Change "*which verifies the steady-state assumption of the RCI and the design of the spin-up (Table 2).*" to "which is a consequence of our choice of RCI and of the design of the spin-up (Table 2)."
Done. Thank you for this suggestion.

l274-277
Much of this information is repetitive information with respect to the paragraph above. Furthermore, please be more careful about the impacts of the RCI assumptions when discussing these results (see Minor comment 2).
Thank you for this comment. We have shortened and modified the discussion of the RCI and its impact on trends. We have removed *"By design of the spin-up, there are no substantial changes in the spatially-integrated FAC between the start and end of the RCI, which means the change to the sea-level rise buffering capacity of the Greenland firn layer is also negligible."* We have also changed *"our assignment of that time period as the RCI"* to **the design of the spin-up**.

l276
Change "*Greenland's sea level rise buffering capacity*" to "the sea-level rise buffering capacity of the Greenland firn layer".
Done.

l280
"*somewhat*": please quantify.
We have removed *"somewhat"* and instead name all the basins where this trend is evident. It now reads: **…is evident in the basin-averaged FAC in the northeast, southeast, and southwest…**

l282
Throughout their manuscript, the authors use the term "*seasonal breathing*" to designate the seasonal fluctuations in FAC. I know that this term has been used in some previous studies. However, I personally dislike this term. In my view, it is scientifically incorrect: firn does not breathe. I would appreciate if the

authors replace this term by another one. For example: the FAC seasonal amplitude. I thank the authors for their understanding.

Thank you for pointing this out. We had originally chosen to use the term since it appeared in other literature, but we acknowledge that it is misleading and not completely accurate. We have replaced *seasonal breathing signal* with **seasonal signal** to fix this.

l285-286

"*the seasonality was undetectable by the chosen methods*": please explain in Figure A3 why the seasonality was undetectable in some cases.

Thank you for suggesting this. The seasonality was not undetectable in Figure A3 (which shows the full ice sheet), but rather in some of the individual basins. We have added the following text to this sentence to clarify why this is: *(i.e., some basins contain too much intra-annual variability for the sine fitting function to detect a seasonal cycle)*

Caption of Figure 5

Add comma: "air temperature, all calculated".

Done.

Figure 6

Why are is there missing data from one or both models in some areas? Please explain in the main text.

We have added the following explanation in Section 3.2: *A few areas of missing data exist and are due to one or both of the firn models encountering an error in the simulation (Fig. 6). For example, if a grid cell is located in the ablation zone and does not receive enough accumulation to model a snow layer, SNOWPACK will stop the simulation and report an error.*

Caption of Figure 6

Please rephrase the first sentence of the caption to make it more intelligible grammatically.

Thank you for pointing this out. We have updated the caption as follows:
*Mean firn air content (FAC) calculated over the upper 100 m of the firn column from (a) SNOWPACK and (b) the CFM for the reference climate interval (RCI, 1980–1995). Panel (c) shows the difference between the modeled FAC values (SNOWPACK minus CFM). The values in the bottom right of each panel are the mean FAC and spatially-integrated FAC. Panel (c) also includes the percent difference. Areas where one or both of the models have missing data are shown in white. Black outlines show the six basins defined by Rignot and Mouginot (2012).*

Caption of Figure 7

Specify: "The bottom row also includes the percent difference averaged over the GrIS."

Done.

Caption of Table 1

Specify: "and the average percent difference between".

Done.

Table 1

Provide root mean squared deviation between CFM-GSFC-FDM and SNOWPACK in each basin. And please discuss these values in the main text.

Thank you for this suggestion; we have updated Table 1 and the manuscript text. See earlier comment on RMSD for the new Table 1 and text updates.

Caption of Table 2

Specify: "The average percent change between".

We have modified Table 2 (see below comment) so this no longer applies.

Table 2

Firstly, I am surprised by the magnitude of changes between 1980 and 1995 (up to 0.5% in magnitude). How is this possible given that the 1980-1995 simulation is simply a repetition of the climatic RCI loop

imposed during the spin-up. As far as I understand the spin-up process, repeating the RCI once more should cause only negligibly small changes. Can the authors please explain this?

Secondly, I think that for both periods (1980-1995 and 2005-2020), it would be more relevant to analyze the trend over the period instead of the change between two individual years. Analyzing only two individual years means that conclusions can be influenced by inter-annual variability. This may explain my first point of this comment.

It is likely that the choice of comparing only two yearly averages is the cause for the magnitude in changes seen in Table 2. When comparing the spatially-integrated FAC from the first and last day of the RCI (i.e., 1 January 1980 and 31 December 1995), the difference in both SNOWPACK and the CFM is <0.1%. The fact that we only selected two years and then averaged over them is likely the cause of these changes. We very much appreciate the suggestion to instead analyze the trends instead, so we have remade Table 2 (see below) with the 1980–1995 and 2005–2020 trends rather than single-year averages. We have also modified the text to report these results.

***Table 2. Modeled spatially-integrated firn air content (FAC) trends and standard errors of the trends for each of the six basins (Fig. 6) for the 1980–1995 reference climate interval (RCI) and the 2005–2020 period. The last row shows the trends for the full GrIS.***

| | 1980–1995 | | 2005–2020 | |
| | CFM-GSFC | SNOWPACK | CFM-GSFC | SNOWPACK |
| Basin | trend ($km^3\ y^{-1}$) | trend ($km^3\ y^{-1}$) | trend ($km^3\ y^{-1}$) | trend ($km^3\ y^{-1}$) |
|---|---|---|---|---|
| NW | -0.1±0.1 | -1.1±0.1 | -2.6±0.1 | -12.3±0.2 |
| CW | +0.2±0.0 | +1.5±0.1 | -2.1±0.0 | -10.3±0.1 |
| SW | -1.0±0.1 | -1.2±0.2 | -6.9±0.1 | -16.7±0.2 |
| NO | +0.6±0.0 | +0.4±0.1 | -5.7±0.0 | -11.4±0.1 |
| NE | -1.3±0.1 | -0.8±0.3 | +1.9±0.1 | -2.5±0.2 |
| SE | +3.1±0.1 | +3.1±0.3 | +2.4±0.2 | -4.5±0.4 |
| GrIS | +1.2±1.0 | +1.7±0.9 | -17.4±1.2 | -66.6±1.2 |

In Section 3.2, we removed the text that reports values from the original Table 2 and have instead added the following sentences:

***The 1980–1995 trends in spatially-integrated FAC are shown in Table 2 for each basin. The trends are very small (maximum = 3.1±0.3 $km^3y^{-1}$), which confirms that no substantial change in FAC occurs during the spin-up. This is a result our our choice of RCI and the design of the spin-up.***

In Section 3.3, we removed the text that reports values from the original Table 2 and have instead added the following paragraph after the discussion of ice-sheet-wide trends:

***The marginal areas of the GrIS experience the greatest amount of FAC depletion between 2005 and 2020 (Fig. 10). Both models simulate the same spatial patterns in loss, but the trends vary by basin (Table 2). SNOWPACK simulates a negative trend in spatially-integrated FAC in all basins during this time, with the strongest trend of -16.7±0.2 $km^3y^{-1}$ in the southwest. The negative trend is weakest in the northeast (-2.5±0.2 $km^3y^{-1}$) and southeast (-4.5±0.4 $km^3y^{-1}$), which are also the only two basins where the CFM-GSFC simulates positive trends (1.9±0.1 and 2.4±0.2 $km^3y^{-1}$, respectively). The CFM-GSFC also simulates the strongest negative trend in the southwest where the spatially-integrated FAC change is -6.9±0.1 $km^3y^{-1}$ (Table 2).***

l292
Change "*statistically*" to "significantly".
Done.

l297-298
"*A consistent decreasing trend is modeled from ~2002 and through ~2011.*": please keep the 2005-2020 as a baseline for analysis. Switching between different periods of analysis makes the messages more confusing.
We have removed this sentence to keep the focus on the defined periods of 1980–1995 and 2005–2020.

l305
Change "*associated water percolation processes*" to "associated water percolation and refreezing processes".
Done.

l312
"*SNOWPACK simulates greater negative changes in FAC compared to the CFM once the models diverge.*": I find this sentence unclear. Please consider rephrasing.
We have replaced this sentence with: ***Following this divergence, the magnitude of FAC change is greater in SNOWPACK compared to the CFM-GSFC.***

l314
Please specify: "followed by an increase in FAC only in the northeast and southeast".
Done.

l317-329
The differences between CFM-GSFC-FDM and SNOWPACK that are described in these two paragraphs are important. This deserves more detailed investigation into the causes of these differences (see Major comment 2).
Thank you for pointing this out. Though we have not expanded the analysis (see earlier response to Major comment 2), we have included a more thorough description and discussion of these results. In Section 3.3, we reported the values of the strongest seasonality: ***(52 and 33 km$^3$ in the CFM-GSFC and SNOWPACK, respective)***. Also to this paragraph, we added a sentence about the precipitation in the basins with low/undetectable signals: ***These three basins also have the lowest annual precipitation (Table A2).*** We have also expanded on this in the discussion (please see the response to the comment below referring to line 429-432).

l335-336
"*the ice slab nearest to the surface, which in some cases could be bare ice at the surface since there is no condition that the ice slab must be beneath a layer of snow or firn*": I apologize, but I will make another pedantic comment. The term "ice slab" has been used a lot over the last 6 years, and it is now often used inappropriately. Ice slabs are thick layers of ice that develop within a layer of porous firn. They should not be confused with the expansion of the ablation area. In this study, the algorithm of the authors makes no difference between development of ice slabs and ablation area extension. For this reason, I ask the authors to replace their use of "*ice slabs*" by "ice slabs or ablation area extension".
This is a good point; we thank the reviewer for helping us make our language more accurate. We have made the modifications to the text and Figure 11. In this section, we have added ***However, we do not distinguish between an ice slab that has formed within the firn and any solid ice exposed at the surface of the ice sheet's ablation zone.*** Elsewhere, we have replaced *"ice slabs"* with either ***solid ice surfaces*** or ***ice slabs and ablation surfaces*** when discussion the model outputs.

l340
Change "*depth to those slabs*" to "depth of those slabs".
We have changed this to ***depth to those surfaces*** to be inclusive to ice slabs and ablation surfaces per the previous comment.

l343-344
Please re-evaluate this following Major comment 1.

We have reworded the start of this section to read as follows: ***In the evaluation of the models with observations, both firn models perform well when evaluated across all SUMup core depths and within each core depth bin. Their overall high NSE coefficients (≥0.09) and low MAPEs (≤16 %) demonstrate their generally good agreement with observed FAC. However, the model performance is not uniform across all core depths (Fig. 6). In cores at least 10 m in depth, the performance is worse than in the full set of cores, and the MAPE is higher than in any subset for both SNOWPACK (27 %) and the CFM-GSFC (19 %). More deep-firn observations may be needed to better evaluate the models at depth.***

l350-351
"*The signature of model biases differs across the ice sheet as climate, topography, and the impact of firn hydrology vary.*": where is this assessed specifically?
Thank you for noticing that this is not specifically assessed in the manuscript. We have removed this sentence for accuracy.

Caption of Figure 9
Please add "note the different y-axis scales."
Done.

l353
"*Some of the highest model biases in SNOWPACK and the CFM occur in southeast Greenland*": please quantify.
We have added ***(>100 %)*** to clarify the threshold for this statement.

l354
Please clarify: "First, some of the observed density profiles are from cores that were drilled directly into a perennial firn aquifer (Miller et al., 2018)."
Done.

l361
"*Our results agree with these findings that model differences are highest where liquid water is present*": please quantify the performances of the models in conditions of high LTSR.
We have made this statement more specific to the five sites we discussed since we do not have enough high-LTSR observation sites for a meaningful evaluation of the model performance in high-LTSR conditions. This sentence and the following one have been updated as follows: ***In this study, the high model bias in the five cores in southeast Greenland supports these findings that model differences are highest where liquid water is abundant. This indicates that poor representation of meltwater percolation processes is still a substantial limiting factor in firn model performance (Verjans et al., 2021).***

l361
Please specify: "where liquid water is abundant".
Done.

l366
"*Within these five cores*": please rephrase.
Changed to ***For these five cores***

Figure 10
I am puzzled by the big differences in the northeast ice caps. I wonder how much this influences the results provided in this study. If this influence is significant, the authors should reconsider the presentation of their results. The focus of this study is "*the Greenland ice sheet*" and not "the Greenland ice sheet and its surrounding ice caps".
The northeast ice caps are indeed interesting and show substantial FAC depletion in recent years. They likely have little affect on the overall study results because they have relatively low FAC to begin with (Figure 6). The substantial FAC depletion in those grid cells could be due to the fact that these are

disconnected from the rest of the ice sheet and are more sensitive to recent climate change. However, a thorough analysis beyond the scope of this work would be necessary to fully understand the changes occurring in that area of Greenland.

l368
Change "*observed*" to "local".
Done.

l374-375
"*gives confidence in the models' abilities to simulate firn properties across the full ice sheet.*": please nuance this statement as the performances of the models in the percolation zone are limited.
We have replaced *across the full ice sheet* with **a wide spatial domain.** And we have added the following sentence: **However, most observations are constrained to the accumulation zone, which limits model validation in the hydrologically complex percolation zone.** We have also noted this limitation in a paragraph added to a new section of the paper, Section 4.4: Study limitations:

**Additionally, the SUMup observations are spatially limited and do not fully capture the variety of potential firn regimes. Few observations exist in the percolation zone where meltwater processes are complex and not often well-represented in models (Vandecrux et al., 2020b). The present study relies on the bucket scheme for vertical meltwater transport, which is a simple choice and not as sophisticated as others. In general, firn model performance is hindered by the representation of meltwater percolation (Verjans et al., 2021). The paucity of percolation zone observations limits the opportunities for evaluating model performance, and in particular, the choice of meltwater percolation scheme.**

l379
Specify: "in the firn models themselves."
Done.

l382-383
"*the LTSR is a stronger predictor of FAC in the CFM compared to SNOWPACK*": please quantify using the coefficient of determination.
Thank you for this suggestion. We have calculated these values and added **($r^2$ = 0.89)** for the CFM-GSFC and **($r^2$ = 0.77)** for SNOWPACK.

l383-384
"*The large range of possible SNOWPACK-simulated FAC values at low LTSR values is likely due to the model's sophisticated new-snow density scheme (…)*": thus, is the difference in LTSR sensitivity between CFM-GSFC- FDM and SNOWPACK mostly caused by the $\rho_0$ assumption for CFM-GSFC-FDM? (see Major comment 1)
This is an interesting point and we thank the reviewer for such an insightful comment. We have recreated Figure 5 with results from the CFM-GSFC run with $\rho_0$ from SNOWPACK to investigate this (Figure R4). There is still a substantial difference in the LTSR/FAC relationship between the two models. The CFM-GSFC spread has increased, but not exclusively at low LTSR points. Since it is not due to the initial density assignment, we attribute this to the different densification schemes in each model. As such, we have updated this sentence in the manuscript to now read: **The large range of possible SNOWPACK-simulated FAC values at low LTSR values is likely due to a combination of (1) the model's sophisticated new-snow density scheme that uses more than only air temperature and accumulation to determine near-surface density, and (2) the densification scheme. The CFM-GSFC uses a set new-snow density and its densification scheme is empirically based, which likely explains the differences between the two models.**

[Figure]

**Figure R4.** Same as Figure 5 in the manuscript but created with CFM-GSFC results that used the new-snow density from SNOWPACK rather than a fixed 350 kg m⁻³.

l386
Change "*This indicates*" to "Our results show".
Done.

l393
Typo: "*has*" should be "have".
Done. Thank you for finding this.

l396
"*which requires more detailed output from a dedicated firn model*": please mention that this is only a particularity of MERRA2 because it does not provide melt as an output.
Good point, thank you. We added *, in the case of MERRA-2,* to specify this.

l400
"*where the spread in FAC is less than in SNOWPACK*": please quantify.
Great suggestion, thank you. We have changed this to **where the spread in FAC (0 to 24 m) is less than in SNOWPACK (0 to 32 m)**.

l402
Change "*is easier to predict using LTSR*" to "shows less variability for a given LTSR".
Done.

l409-410
The total Greenland FAC values provided here do not agree with the values given in Table 2. Please correct this.
Thank you for pointing this out. The values listed here in the text are for the full GrIS (i.e., all MERRA-2 grid cells meeting the 50% ice coverage criteria described in Section 2.1). The original Table 2 reports the sum of the basins, which is why the numbers are smaller. Those values are slightly different from the ones reported here because these ones are long-term averages (1980–1995) rather than single-year averages. Per a previous suggestion from the reviewer, we have chosen to change Table 2 to report trends instead of single year averages.

Additionally, to the results (Section 3.2) we added: **The spatially-integrated FAC, which is the total air volume within the firn layer, is 34,645 km³ for SNOWPACK and 28,581 km³ for the CFM-GSFC. These values represent all modeled grid cells, meaning they include some areas outside of the six basins defined by Rignot and Mouginot (2012).**

l411-412

"*is close to a regional climate model's (HIRHAM5_MOD) estimate for this period (Vandecrux et al., 2019).*": please provide the value.

Unfortunately, Vandecrux et al. (2019) do not provide an exact value for HIRHAM5_MOD's spatially-integrated FAC. Their Figure 7 shows a time series of spatially-integrated FAC from HIRHAM5_MOD, RACMO2.3p2, and observations. From this, we estimate a value of ~34,000 km$^3$, which we have added to our manuscript. This phrase now reads as ***is close to a regional climate model's (HIRHAM5_MOD) long-term estimate of ~34,000 km$^3$ for this period (Vandecrux et al., 2019).***

l416

"*uses the constitutive relationship between stress and strain*": see comment above.

We have change "*the*" to ***a*** to clarify that we are not referencing a specific equation.

l422-423

Remove "*(and by proxy, whether the physics-based or empirical approach is recommended)*".

Done.

l427

Change "*corresponds with*" to "corresponds to".

Done.

l429-432

"*The stronger seasonality in the CFM is indicative of the model's more simple treatment of forcing data like accumulation and temperature, which have strong seasonal patterns. SNOWPACK's same sophisticated new- snow density scheme that leads to a complex relationship between LTSR and FAC also results in this smaller seasonal signal.*": it is unclear to me how the authors reach these conclusions from their results. Please explain more.

Thank you for pointing out the need for a stronger explanation here. We have rewritten these sentences: ***The temporal patterns in FAC are directly related to the atmospheric forcing. Atmospheric input like accumulation and temperature have strong seasonal patterns, which likely makes them the strongest drivers of FAC seasonality. Both models' FAC seasonal signals are primarily driven by the forcing, but strength of the signal is tied to how each model treats the forcing. SNOWPACK's FAC seasonality is weaker than that of the CFM-GSFC, which points to a more complex treatment of accumulation and temperature in SNOWPACK. In particular, these forcing variables are used in SNOWPACK's new-snow density scheme that the CFM-GSFC lacks. As such, the same new-snow density scheme that leads to the complex relationship between LTSR and FAC may also dampen SNOWPACK's seasonal signal.***

l434

"*the 2012 extreme melt season can be seen as an abrupt drop in FAC in most basins*": please be more nuanced. There is an abrupt drop only in the southwest, southeast, and to a lesser degree northeast basins.

Thank you. We have changed *"most"* to ***three*** to be more specific.

l441

Please be more nuanced: "Ice slabs, which may render deep pore space inaccessible".

Done.

l443-444

Please rephrase this sentence.

Thank you for catching our mistake of leaving out the word, "which". The sentence now reads: ***The largest percent change is in the southwest, which has the warmest temperatures and highest melt compared to other basins during this period (Table A2).***

l445-447

*"Pore space depletion can also be a sign of firn densification, which has been found to increase cold content in the firn and amplify meltwater freezing and ice slab formation in the near-surface (Vandecrux et al., 2020a)."*: this statement is a very crude simplification of a complex process with many interactions and feedbacks. I suggest rephrasing: "Pore space depletion can also be caused by firn densification, which in turn modifies the meltwater refreezing and retention capacities of the firn in a complex manner.*"*
Done.

l459
*"place our results in a context of uncertainty"*: this study does not perform any uncertainty analysis.
We have removed this sentence for accuracy.

l460-461
Change *"lets us isolate the differences in the models themselves and examine how they respond to the same forcing"* to "lets us examine how the models respond differently to the same forcing due to structural model differences and parameterization choice".
Done.

l461
Change *"metric"* to "factor".
Done.

l464
Change *"While FAC in the CFM more gradually decreases as"* to "While FAC more gradually decreases in the CFM as".
Done.

l465
*"reaches near-zero FAC values that the CFM does not capture"*: please consider rephrasing. This sentence suggests that SNOWPACK is right and that CFM is wrong.
This is a very good point; we thank the reviewer for picking up on this unintentional connotation. We have reworded this to say: **reaches near-zero FAC values not simulated by the CFM-GSFC.**

l466
Change *"even more"* to "on larger areas".
Done.

l470
The FAC values given here do not agree with values in Table 2. Please correct this.
Thank you for pointing this out. Please see the earlier comment addressing this.

l471
*"reasonable estimations when compared to other studies"*: please quantify.
We have changed this to: ***…reasonable estimations when compared to other studies (e.g., 26,800 km³ from observations and ~34,000 km³ from a model (Vandecrux et al., 2019).***

l477
Change *"the pore space depletion is more extreme"* to "the pore space is more depleted".
Done.

l480
Specify: "the firn layer in the CFM loses only an equivalent of 1 mm."
Done.

Caption of Table A1

Does the standard deviation refer to (a) the within-basin standard deviations of the RCI mean values or (b) the annual standard deviation of the basin mean values? If it is (a) specify "mean plus/minus basin standard deviation", and if it is (b) specify "mean plus/minus inter-annual standard deviation".
Caption of Table A2
Same comment as for the caption of Table A1.
Thank you for noting the ambiguity in this statistic. In both tables, the standard deviation refers to the within-basin standard deviations of the RCI mean values. We have updated the captions for both Table A1 and A2 as suggested with (a).

Caption of Figure A1
Move "*(green vertical line)*" next to "*350 kg m−3*".
Done.

Caption of Figure A2
Change "*full ice sheet*" to "Greenland ice sheet".
Changed to **GrIS**.

Section "*Code and data availability*"
I thank the authors for providing the entire model output from both models as an open-source dataset. However, I ask the authors to include a README file in the dataset that explains in details how one can read and extract information from the files.
We thank the reviewer for this suggestion that supports open science and access to data. We have created a README file and uploaded it to Zenodo. The code and data availability section has been updated to include the new Zenodo link: ***https://doi.org/10.5281/zenodo.7671892***.

References used in this review
Arnaud, L., Gay, M., Barnola, J.-M., and Duval, P.: Physical modeling of the densification of snow/firn and ice in the upper part of polar ice sheets, in: Physics of Ice Core Records, edited by: Hondoh, T., Hokkaido University Press, Sapporo, Japan, 285–305, 2000.
Arthern, R. J. and Wingham, D. J.: The Natural Fluctuations of Firn Densification and Their Effect on the Geodetic Determination of Ice Sheet Mass Balance, Clim. Change, 40, 605–624, https://doi.org/10.1023/A:1005320713306, 1998.
Arthern, R. J., Vaughan, D. G., Rankin, A. M., Mulvaney, R., and Thomas, E. R.: In situ measurements of Antarctic snow compaction compared with predictions of models, J. Geophys. Res.-Earth Surf., 115, 1– 12, https://doi.org/10.1029/2009JF001306, 2010.
Benson, C. S.: Stratigraphic Studies in the Snow and Firn of the Greenland Ice Sheet, U.S. Army Snow, Ice and Permafrost Research Establishment (SIPRE–CRREL), Research Report 70, reprinted with revisions by CRREL, 1996, 1962.
Braithwaite, R., Laternser, M., and Pfeffer, W. T.: Variation of near-surface firn density in the lower accumulation area of the Greenland ice sheet, Pâkitsoq, West Greenland, J. Glaciol., 40, 477– 485, https://doi.org/10.3189/S002214300001234X, 1994.
Fausto, R. S., Box, J. E., Vandecrux, B., van As, D., Steffen, K., MacFerrin, M., Machguth H., Colgan W., Koenig L. S., McGrath D., Charalampidis, C., and Braithwaite, R. J.: A Snow Density Dataset for Improving Surface Boundary Conditions in Greenland Ice Sheet Firn Modeling, Front. Earth Sci., 6, 51 pp., https://doi.org/10.3389/feart.2018.00051, 2018
Hanna, E., Huybrechts, P., Cappelen, J., Steffen, K., Bales, R. C., Burgess, E., McConnell, J. R., Peder Steffensen, J., Van den Broeke, M., Wake, L., Bigg, G., Griffiths, M., and Savas, D.: Greenland Ice Sheet surface mass balance 1870 to 2010 based on Twentieth Century Reanalysis, and links with global climate forcing, J. Geophys. Res.-Atmos., 116, D24121, https://doi.org/10.1029/2011JD016387, 2011.
Herron, M. and Langway, C.: Firn densification: an empirical model, J. Glaciol., 25, 373–385, https://doi.org/10.3189/S0022143000015239, 1980.
Kuipers Munneke, P., Ligtenberg, S. R. M., van den Broeke, M. R., van Angelen, J. H., and Forster, R. R.: Explaining the presence of perennial liquid water bodies in the firn of the Greenland Ice Sheet, Geophys. Res. Lett., 41, 476–483, https://doi.org/10.1002/2013GL058389, 2014.

Kuipers Munneke, P., Ligtenberg, S. R. M., Noël, B. P. Y., Howat, I. M., Box, J. E., Mosley-Thompson, E., McConnell, J. R., Steffen, K., Harper, J. T., Das, S. B., and van den Broeke, M. R.: Elevation change of the Greenland Ice Sheet due to surface mass balance and firn processes, 1960–2014, The Cryosphere, 9, 2009–2025, https://doi.org/10.5194/tc-9-2009-2015, 2015.

Maeno, N. and Ebinuma, T.: Pressure sintering of ice and its implication to the densification of snow at polar glaciers and ice sheets, J. Phys. Chem., 87, 4103–4110, https://doi.org/10.1021/j100244a023, 1983.

Medley, B., Neumann, T. A., Zwally, H. J., Smith, B. E., and Stevens, C. M.: Simulations of firn processes over the Greenland and Antarctic ice sheets: 1980–2021, The Cryosphere, 16, 3971–4011, https://doi.org/10.5194/tc-16-3971-2022, 2022.

Morris, E. M. and Wingham, D. J.: Densification of polar snow: Measurements, modeling, and implications for altimetry, J. Geophys. Res.-Earth Surf., 119, 349–365, https://doi.org/10.1002/2013JF002898, 2014.

Morris, E. M. and Wingham, D. J. : Uncertainty in mass balance trends derived from altimetry; a case study along the EGIG line, Central Greenland. Journal of Glaciology, 61(226), 345–356, https://doi.org/10.3189/2015JoG14J123, 2015.

Noël, B., van de Berg, W. J., Machguth, H., Lhermitte, S., Howat, I., Fettweis, X., and van den Broeke, M. R.: A daily, 1 km resolution data set of downscaled Greenland ice sheet surface mass balance (1958–2015), The Cryosphere, 10, 2361–2377, https://doi.org/10.5194/tc-10-2361-2016, 2016.

Simonsen, S. B., Stenseng, L., Adalgeirsdóttir, G., Fausto, R. S., Hvidberg, C. S., and Lucas-Picher, P.: Assessing a multilayered dynamic firn-compaction model for Greenland with ASIRAS radar measurements, J. Glaciol., 59, 545–558, https://doi.org/10.3189/2013JoG12J158, 2013.

Sørensen, L. S., Simonsen, S. B., Nielsen, K., Lucas-Picher, P., Spada, G., Adalgeirsdottir, G., Forsberg, R., and Hvidberg, C. S.: Mass balance of the Greenland ice sheet (2003–2008) from ICESat data – the impact of interpolation, sampling and firn density, The Cryosphere, 5, 173–186, https://doi.org/10.5194/tc-5-173- 2011, 2011.

Steger, C. R., Reijmer, C. H., van den Broeke, M. R., Wever, N., Forster, R. R., Koenig, L. S., Kuipers Munneke, P., Lehning, M., Lhermitte, S., Ligtenberg, S. R. M., Miège, C., and Noël, B. P. Y.: Firn Meltwater Retention on the Greenland Ice Sheet: A Model Comparison, Front. Earth Sci., 5, 3, https://doi.org/10.3389/feart.2017.00003, 2017

Verjans, V., Leeson, A. A., Stevens, C. M., MacFerrin, M., Noël, B., and van den Broeke, M. R.: Development of physically based liquid water schemes for Greenland firn-densification models, The Cryosphere, 13, 1819–1842, https://doi.org/10.5194/tc-13-1819-2019, 2019.

Verjans, V., Leeson, A. A., Nemeth, C., Stevens, C. M., Kuipers Munneke, P., Noël, B., and van Wessem, J. M.: Bayesian calibration of firn densification models, The Cryosphere, 14, 3017–3032, https://doi.org/10.5194/tc-14-3017-2020, 2020.

Verjans, V., Leeson, A., McMillan, M., Stevens, C., van Wessem, J. M., van de Berg, W. J., van den Broeke, M., Kittel, C., Amory, C., Fettweis, X., Hansen, N., Boberg, F., and Mottram, R.: Uncertainty in East Antarctic firn thickness constrained using a model ensemble approach, Geophys. Res. Lett., 48, e2020GL092060, https://doi.org/10.1029/2020GL092060, 2021.

References used in response

Cuffey, K. M., & Paterson, W. S. B. (2010). *The physics of glaciers*. Academic Press.

Bartelt, P., & Lehning, M. (2002). A physical SNOWPACK model for the Swiss avalanche warning: Part I: numerical model. *Cold Regions Science and Technology*, *35*(3), 123-145.

Lehning, M., Bartelt, P., Brown, B., Fierz, C., & Satyawali, P. (2002a). A physical SNOWPACK model for the Swiss avalanche warning: Part II. Snow microstructure. *Cold regions science and technology*, *35*(3), 147-167.

Lehning, M., Bartelt, P., Brown, B., & Fierz, C. (2002b). A physical SNOWPACK model for the Swiss avalanche warning: Part III: Meteorological forcing, thin layer formation and evaluation. *Cold Regions Science and Technology*, *35*(3), 169-184.

Stevens, C. M., Verjans, V., Lundin, J., Kahle, E. C., Horlings, A. N., Horlings, B. I., & Waddington, E. D. (2020). The community firn model (cfm) v1. 0. *Geoscientific Model Development*, *13*(9), 4355-4377.

Medley, B., Neumann, T. A., Zwally, H. J., Smith, B. E., & Stevens, C. M. (2022). Simulations of firn processes over the Greenland and Antarctic ice sheets: 1980–2021. *The Cryosphere*, *16*(10), 3971-4011.

---

## Referee Report (RR1)

I would like to thank the authors for the effort they put in the revision of the manuscript. Please find below some minor additional remarks (line numbers refer to the revised manuscript with marked changes):

**Point-comments**
**L105:** I would specify the precipitation types that were used from MERRA-2 – e.g. "precipitation rate (convective/large-scale rainfall and snowfall)"
**Fig. 1:** You added degree symbols to this figure in the author's response – but they are not present in the final manuscript (maybe you forgot to update this figure).
**Table 3:** You could refer to Sect. 3.2 here: "Undetectable signals (see Sect. 3.2) are..."
**L291:** You could move the liquid-to-solid ratio (LTSR) computation to Sect. 2 (Methods and data)
**L317:** Now, I'm a bit confused. Do you set all grid cells that do not receive enough accumulation to build up a firn layer to NaN? It seems so from Fig. 6. That means that spatial averages (e.g. for FAC) are actually rather computed over the accumulation zone than over the entire GrIS – right? I think it would be useful to state this somewhere explicitly (in case you apply this method...).
**L401:** Value "52" inconsistent with table 3.
**L423:** Sorry, I did not notice this during the first review iteration. There seems to be a systematic deviation between the modelled and observed ice slabs – i.e. the observed ice slabs are typically located a bit more towards the interior of the ice sheet (and thus probably buried by firn). Is this an indication that both models underestimate the formation of (such buried) ice slabs?

**Typos, phrasing and stylistic comments**
**L314:** "A few areas of missing data exist (see Fig. 6), which is caused by modelling issues in one or both firn models."
**L354:** "sine wave" → "sine function"
**L435:** "than in any other subset"
**L436:** "models at greater (>10 m) depths."
**L477:** "have higher model uncertainties" or "have a higher model uncertainty" (?)
**L552:** "atmospheric forcing and the variable densification schemes."
**L581:** "at greater (>? m) depth." (?)
**L613:** "reveal significant differences but also perform well within a reasonable range." (or similar...)
**L656:** "required conditions" (?)
**Fig. A1:** "which falls near many..." → "which agrees well with mean observed surface density." (it's probably better to mention this fact in the context of panel (b)).

---

## Referee Report (RR2)

Review of Thompson-Munson et al.: *An evaluation of a physics-based and a semi-empirical firn model across the Greenland Ice Sheet (1980–2020)*
by Vincent Verjans.

This study applies two state-of-the-art firn models at the scale of the Greenland Ice Sheet (GrIS). The Community Firn Model is used with the semi-empirical NASA GSFC-FDMv1.2 .1 densification scheme (CFM-GSFC). SNOWPACK is a more physically-detailed snow compaction model. The goal of this study is to compare results from these two different approaches to firn modeling over the GrIS. The authors also perform a comparison of model output against in-situ firn core observations. The authors have thoroughly reworked the manuscript since its first version. This includes an improved evaluation of model output with respect to firn core data, and a better interpretation of the results. This review only includes specific comments, which are still important to be addressed. I recognize the thorough work of the authors to address all my comments from the first round of reviews, and I sincerely appreciate their extensive answers in the author responses. Provided some very minor issues in the updated manuscript are addressed by the authors, I encourage the publication of this study in The Cryosphere.

Specific comments
l 45-46
Change "*in both observations (e.g., Vandecrux et al., 2019; Benson, 1996; Braithwaite et al., 1994; Sørensen et al., 2011; Kuipers Munneke et al., 2015) and models (e.g., Medley et al., 2022)*" to "in both observations (e.g., Vandecrux et al., 2019; Benson, 1996; Braithwaite et al., 1994) and models (e.g., Sørensen et al., 2011; Kuipers Munneke et al., 2015; Medley et al., 2022)".
l 69
Change "*densification based constitutive*" to "densification based on constitutive".
l 82
Change "*e.g.* " to "e.g., ".
l 124
Change "*from the MERRA-2 grids*" to "from the MERRA-2 fields".
l 131
Change "*in adjacent layers*" to "between adjacent layers".
l 149-150
Change "*it is important to specific*" to "it is important to specify".
l 156-159
I suggest removing the sentences "*The CFM-GSFC uses a layer-merging scheme at 5- and 10-m depth to reduce computational demands. The CFM-GSFC is coded so that each model time step adds a new layer. As such, daily time stepping generates many thin layers.*" And changing the next sentence to "To reduce computational demands associated with many daily accumulation events, we use the CFM-GSFC's layer merging scheme." This makes the text less repetitive.
l 227
Add a comma: "In these shallow cores, where densification".
l 248-249
Change "values comparable to" to "values comparable or improved with respect to".
l 284-285
Remove "*, which represents the averaged difference in FAC between the two models,*" because this is an incorrect definition of the RMSD.
l 355
Specify "remains relatively constant (Fig. 9a, 9b)" to make clear that this statement refers to the northwest and central west basins.
l 413
Change "*Verjans et al., 2021*" to "Verjans et al., 2019".
l 511

Change "*is likely*" to "are likely".

l 539

Change "*not often well-represented*" to "often not well-represented".

l 542

Change "*Verjans et al., 2021*" to "Verjans et al., 2019".

l 549

Change "*Model error is higher in comparisons with deeper cores than in shallower ones*" to "Model error is higher with respect to deeper cores than to shallower ones".

l 569

Change "*from a model*" to "from a regional climate model".

l 581

Change "*drawback*" to "drawbacks".

l 582

Change "*biased toward available observational data*" to "constrained toward available observational data".

l 582-583

Change "*This may result in more realistic simulations of firn properties under future climate conditions,*" to "This avoids simulations beyond the calibration range of the firn model under future climate conditions,".

---

## Author Response (AR2)

I would like to thank the authors for the effort they put in the revision of the manuscript. Please find below some minor additional remarks (line numbers refer to the revised manuscript with marked changes):

We appreciate that the reviewer has read through the manuscript again and helped us improve the clarity of the writing. We thank the reviewer for their comments and suggestions, and we have made changes to the manuscript accordingly.

**Point-comments**

**L105:** I would specify the precipitation types that were used from MERRA-2 – e.g. "precipitation rate (convective/large-scale rainfall and snowfall)"
Thank you for this suggestion. We've changed it to *rainfall and snowfall*.

**Fig. 1:** You added degree symbols to this figure in the author's response – but they are not present in the final manuscript (maybe you forgot to update this figure).
Thank you very much for pointing this out. You are correct that we forgot to update the figure in the manuscript after adding the updated one to the reviewer response. We have double-checked that the figures in the manuscript have been updated as reported in the response.

**Table 3:** You could refer to Sect. 3.2 here: "Undetectable signals (see Sect. 3.2) are..."
We appreciate this idea and have made the suggested change in the Table 3 caption.

**L291:** You could move the liquid-to-solid ratio (LTSR) computation to Sect. 2 (Methods and data)
Thank you for this suggestion. We have moved these lines to Section 2 and created a new subsection (Section 2.6) after the description of the firn air content calculation. The text and equation are the same.

**L317:** Now, I'm a bit confused. Do you set all grid cells that do not receive enough accumulation to build up a firn layer to NaN? It seems so from Fig. 6. That means that spatial averages (e.g. for FAC) are actually rather computed over the accumulation zone than over the entire GrIS – right? I think it would be useful to state this somewhere explicitly (in case you apply this method...).
We appreciate the reviewer pointing this out and we now understand how those added sentences create confusion. Many of the ablation zone grid cells do successfully build up a firn/ice layer over time. In Fig. 6, most grid cells within (and even some outside of) the black boundaries were successful. However, there were a few locations that are mostly along the southwest margin where accumulation was too low (or melt was too high) for a layer to build up over time. The 50% ice coverage threshold (Sec. 2.1) means that some grid cells are only half glaciated. This creates the chance for a snow/ice layer to not build up since the conditions for snow to accumulate and remain through the year may not be met. To clarify this, we have changed these sentences to now read as: *For example, if a grid cell does not receive enough accumulation to build up a firn/ice layer over time, we treat that grid point as missing data.*

Additionally, as to the comment on the spatial averages, we acknowledge that these values will be affected by the spatial coverage. However, the averages do not simply represent the accumulation zone (see Fig. 6 and the grid cells outside of the black boundaries). The spatial averages are for the modeled region, defined by the 50% threshold. There are grid cells both along the margin (where we expect low FAC) and in the interior (where we expect higher FAC) that returned NaN's in the model simulations. It's possible that these could even out the effect of missing data on the average, but it's difficult to say with any certainty. Moreover, the grid cells in the east and northeast outside of the Rignot and Mouginot (2012) ice sheet boundaries have generally low FAC (Fig. 6), which may bring the spatial average down. All of this is to say that the spatial average is not representative of any region in particular. To convey this to the reader, we have modified this sentence in Section 3.2: *"These values represent all modeled grid cells, meaning they include some areas outside of the six basins defined by Rignot and Mouginot (2012)."* to now say *These values represent all modeled grid cells, meaning they include some areas outside of the six basins defined by Rignot and Mouginot (2012) and may not be directly comparable to other studies.*

We hope that these modifications and explanations help clarify this point of confusion.

**L401:** Value "52" inconsistent with table 3.
Thank you for pointing this out. This is a typo in the text, so we have changed *"52"* to *53*.

**L423:** Sorry, I did not notice this during the first review iteration. There seems to be a systematic deviation between the modelled and observed ice slabs – i.e. the observed ice slabs are typically located a bit more towards the interior of the ice sheet (and thus probably buried by firn). Is this an indication that both models underestimate the formation of (such buried) ice slabs?
This is an interesting point and we thank the reviewer for noticing. There are likely at least three possibilities for the different locations: (1) as stated in this comment, the models both underestimate ice slab formation in the near surface; (2) the coarse resolution of the MERRA-2 grid cells is insufficient for the spatial heterogeneity of ice slabs; and/or (3) the IceBridge flight lines do not sample the entire ice sheet. We have added to this line: ***though the observed ice slabs are located slightly up-glacier from the modeled ones on average***. We do not speculate as to why this is, since there are several potential options that are difficult to test.

**Typos, phrasing and stylistic comments**

**L314:** "A few areas of missing data exist (see Fig. 6), which is caused by modelling issues in one or both firn models."
Done.

**L354:** "sine wave" → "sine function"
Done.

**L435:** "than in any other subset"
Done.

**L436:** "models at greater (>10 m) depths."
Done.

**L477:** "have higher model uncertainties" or "have a higher model uncertainty" (?)
Done.

**L552:** "atmospheric forcing and the variable densification schemes."
Done.

**L581:** "at greater (>? m) depth." (?)
We appreciate this suggestion, however, we have chosen to leave this as ***at depth*** because the depth in question could vary substantially based on annual accumulate rates (which also vary greatly across the ice sheet).

**L613:** "reveal significant differences but also perform well within a reasonable range." (or similar...)
Done.

**L656:** "required conditions" (?)
Done.

**Fig. A1:** "which falls near many..." →"which agrees well with mean observed surface density." (it's probably better to mention this fact in the context of panel (b)).
Thank you for this suggestion. We have updated the sentence as suggested and reorganized the caption. It now reads:

*(a) Observed surface density ($\rho_0$) from SUMup versus SNOWPACK. Since some observations begin farther below the surface, in this figure, observed $\rho_0$ is defined as the uppermost density measurement that is within 0.1 m from the surface. The SNOWPACK $\rho_0$ is calculated over the same vertical segment as the SUMup observation. (b) Histogram of observed surface density with the mean represented by the black line. The CFM-GSFC uses a prescribed surface density of 350 kg m$^{-3}$ (green vertical line in both panels), which agrees well with the mean observed surface density.*

Review of Thompson-Munson et al.: An evaluation of a physics-based and a semi-empirical firn model across the Greenland Ice Sheet (1980–2020)
by Vincent Verjans.

This study applies two state-of-the-art firn models at the scale of the Greenland Ice Sheet (GrIS). The Community Firn Model is used with the semi-empirical NASA GSFC-FDMv1.2 .1 densification scheme (CFMGSFC). SNOWPACK is a more physically-detailed snow compaction model. The goal of this study is to compare results from these two different approaches to firn modeling over the GrIS. The authors also perform a comparison of model output against in-situ firn core observations. The authors have thoroughly reworked the manuscript since its first version. This includes an improved evaluation of model output with respect to firn core data, and a better interpretation of the results. This review only includes specific comments, which are still important to be addressed. I recognize the thorough work of the authors to address all my comments from the first round of reviews, and I sincerely appreciate their extensive answers in the author responses. Provided some very minor issues in the updated manuscript are addressed by the authors, I encourage the publication of this study in The Cryosphere.

We thank the reviewer for again reading and reviewing the manuscript. We appreciate the suggestions and comments and have updated the manuscript accordingly.

Specific comments

l 45-46
Change "in both observations (e.g., Vandecrux et al., 2019; Benson, 1996; Braithwaite et al., 1994; Sørensen et al., 2011; Kuipers Munneke et al., 2015) and models (e.g., Medley et al., 2022)" to "in both observations (e.g., Vandecrux et al., 2019; Benson, 1996; Braithwaite et al., 1994) and models (e.g., Sørensen et al., 2011; Kuipers Munneke et al., 2015; Medley et al., 2022)".
Done.

l 69
Change "densification based constitutive" to "densification based on constitutive".
Done.

l 82
Change "e.g. " to "e.g., ".
Done.

l 124
Change "from the MERRA-2 grids" to "from the MERRA-2 fields".
Done.

l 131
Change "in adjacent layers" to "between adjacent layers".
Done.

l 149-150
Change "it is important to specific" to "it is important to specify".
Done.

l 156-159
I suggest removing the sentences "The CFM-GSFC uses a layer-merging scheme at 5- and 10-m depth to reduce computational demands. The CFM-GSFC is coded so that each model time step adds a new layer. As such, daily time stepping generates many thin layers." And changing the next sentence to "To reduce computational demands associated with many daily accumulation events, we use the CFM-GSFC's layer merging scheme." This makes the text less repetitive.
Done.

l 227
Add a comma: "In these shallow cores, where densification".
Done.

l 248-249
Change "values comparable to" to "values comparable or improved with respect to".
Done.

l 284-285
Remove ", which represents the averaged difference in FAC between the two models," because this is an incorrect definition of the RMSD.
Done.

l 355
Specify "remains relatively constant (Fig. 9a, 9b)" to make clear that this statement refers to the northwest and central west basins.
Done.

l 413
Change "Verjans et al., 2021" to "Verjans et al., 2019".
Done. Thank you for noting this citation error (both here and below).

l 511
Change "is likely" to "are likely".
Done.

l 539
Change "not often well-represented" to "often not well-represented".
Done.

l 542
Change "Verjans et al., 2021" to "Verjans et al., 2019".
Done.

l 549
Change "Model error is higher in comparisons with deeper cores than in shallower ones" to "Model error is higher with respect to deeper cores than to shallower ones".
Done.

l 569
Change "from a model" to "from a regional climate model".
Done.

l 581
Change "drawback" to "drawbacks".
Done.

l 582
Change "biased toward available observational data" to "constrained toward available observational data".
Thank you for this suggestion. We have changed this to **constrained by available observation data**.

l 582-583
Change "This may result in more realistic simulations of firn properties under future climate conditions," to "This avoids simulations beyond the calibration range of the firn model under future climate conditions,".
Done. Thank you for this suggested rewording that improves the meaning of the sentence.